Report No.: **FERMILAB-PUB-25-0025-ETD**

First version submitted to arXiv on **February 11, 2025**
This version submitted to arXiv on **February 11, 2025**

# A Demonstration of ARCANE Reweighting: Reducing the Sign Problem in the MC@NLO Generation of $e^+e^- \to q\bar{q} + 1\,\texttt{jet}$ Events

**Prasanth Shyamsundar[1]⋆**

**1** Fermi National Accelerator Laboratory, Batavia, Illinois 60510, USA

⋆ prasanth@fnal.gov

## Abstract

Negatively weighted events, which appear in the simulation of particle collisions, significantly increase the computational requirements of collider experiments. A new technique called ARCANE reweighting has been introduced in a companion paper to tackle this problem. This paper demonstrates the technique for the next-to-leading-order generation of $e^+e^- \to q\bar{q} + 1\,\texttt{jet}$ events. By redistributing the contributions of "standard" and "hard remainder" pathways in the generator that lead to the same final event, ARCANE reweighting almost completely eliminates the negative weights problem for this process. Some thoughts on implementing the technique in other scenarios are provided.

# 1 Introduction

Negatively weighted events, which appear in the generation of collider events at next-to-leading-order (NLO) accuracy, e.g. using the MC@NLO formalism, significantly reduce the efficiency of collider simulations [1,2]. The presence of negatively weighted events increases the total number of simulated events needed to attain specific target precisions in the Monte Carlo predictions made using the simulated data [1,2]. Since negative event weights cannot be eliminated using the standard rejection reweighting and unweighting techniques, the inefficiency is reflected not just in the event generation stage but also in the subsequent stages of the simulation pipeline, including detector simulation, electronics simulation, etc. This problem has received some attention in the last few years, leading to the invention of several new theoretical and statistical/Monte Carlo solutions to ameliorate the problem [3–11]. However, the negative weights problem continues to be a major challenge for the collider physics community [1,2].

In Ref. [12], a new technique dubbed ARCANE reweighting has been introduced in order to reduce the negative weights problem in collider event generation. The present paper serves as a companion to Ref. [12] and provides a demonstration of the technique for the NLO generation of $e^+e^- \longrightarrow q\bar{q} + 1\,\texttt{jet}$ events using the MC@NLO formalism [13–15]. Admittedly, the negative weights problem in the generation of this process using existing techniques is not particularly severe, when compared to processes like $pp \longrightarrow Z + n\,\texttt{jets}$ and $pp \longrightarrow t\bar{t} + n\,\texttt{jets}$. However, the chosen example captures the essence of the negative weights problem in the more important cases while being simpler to implement, because it avoids a few complications associated with hadron collisions (like parton distribution functions and initial state radiation).

The ARCANE reweighting technique can be briefly summarized as follows.[1] Consider a generation pipeline that produces events parameterized as $(V, L, W^{(\text{ORIG})})$, where

- $V$ represents all the attributes of an event that are "visible" to (i.e., will be used by) the subsequent stages of the simulation and analysis pipeline,

- $L$ represents all the latent attributes of an event that are "hidden" from the subsequent stages of the simulation pipeline, and

- $W^{(\text{ORIG})}$ is a special weight attribute used to modify the distribution represented by the simulated data.

---

[1]The presentation here has some minor changes in notation compared to Ref. [12]. For example, $L$ is used here for hidden information instead of $H$ to avoid confusion with the notation $\mathbb{H}$ for hard remainder events in the MC@NLO formalism.

Here, $V$ and $L$ are assumed to capture all sources of randomness in the generation of the event. So, $W^{(\text{ORIG})}$ is fully determined by $V$ and $L$, and will be interchangeably written as $W^{(\text{ORIG})}(V, L)$. ARCANE reweighting involves additively modifying the event weights as

$$W^{(\text{ARCANE})}(V, L) \equiv W^{(\text{ORIG})}(V, L) + \frac{G(V, L)}{P(V, L)}, \tag{1}$$

where $P(V, L)$ is the sampling probability density of $(V, L)$ under the give simulation pipeline and $G(V, L)$ is a special function called the ARCANE redistribution function that satisfies the following condition:

$$\int_{\Omega_L} d\ell \; G(v, \ell) \equiv 0, \qquad \forall v \in \Omega_V, \tag{2}$$

where $\Omega_L$ and $\Omega_V$ are the domains of $L$ and $V$, respectively, and the integration with respect to $\ell$ is performed using an appropriate reference measure. The reweighting in (1) can be thought of as redistributing the contributions from different Monte Carlo "histories" or pathways, denoted by $(V, L)$, in the event generator that lead to the exact same value of $V$. The weighted densities of $(V, L)$, also referred to as the quasi densities here and in Ref. [12], under the "ORIG" and "ARCANE" weighting schemes are given by

$$F^{(\text{superscript})}(V, L) \equiv P(V, L) \, \mathbb{E}\left[ W^{(\text{superscript})}(V, L) \,\Big|\, V, L \right] \equiv P(V, L) \, W^{(\text{superscript})}(V, L). \tag{3}$$

Likewise the quasi densities of $V$, under the different weighting schemes, are given by

$$F^{(\text{superscript})}(V) \equiv P(V) \, \mathbb{E}\left[ W^{(\text{superscript})}(V, L) \,\Big|\, V \right]. \tag{4}$$

It can be seen that the quasi densities are related as

$$F^{(\text{ARCANE})}(V, L) \equiv F^{(\text{ORIG})}(V, L) + G(V, L), \tag{5}$$

$$F^{(\text{ARCANE})}(V) \equiv F^{(\text{ORIG})}(V). \tag{6}$$

In this way, ARCANE reweighting modifies the joint quasi density of $(V, L)$ without affecting the quasi density of $V$. The original event-weights could be negative if $F^{(\text{ORIG})}$ is negative for certain values of $(V, L)$. In such situations, redistributing the contributions of the different pathways, as in (5), can reduce the negative weights problem. In addition to (2), the ARCANE redistribution function $G$ needs to satisfy certain conditions to ensure proper coverage and finiteness of weights; this will be reviewed later, as needed, in this paper.

From the simple description above of ARCANE reweighting, it may not be clear how or if the technique can be used to tackle the negative weights problem in realistic event generation scenarios; this paper is intended to bridge this gap. The rest of the paper is organized as follows. Section 2 provides a detailed overview of the event generator pipeline, constructed using previously existing techniques, for generating $e^+ e^- \longrightarrow q\bar{q} + 1\,\texttt{jet}$ events under the MC@NLO formalism. Some groundwork for the subsequent implementation of ARCANE reweighting is also described in this section. Section 3 describes the implementation of ARCANE reweighting in detail, for the example at hand. Some results are described and discussed in Section 4. Some concluding remarks, including thoughts on implementing the technique for other processes of interest, are provided in Section 6.

All functions used in this paper are assumed to (or will be ensured to) have finite values in the relevant domains—the physics formalisms used to tackle divergences, like subtraction schemes, are assumed to already be incorporated into the event generation pipeline prior to modifying the pipeline with ARCANE reweighting. All functions that are used as integrands in this paper are assumed to satisfy appropriate notions of integrability. All integrals in this paper are assumed to be performed with respect to appropriate reference measures. For simplicity, throughout this paper, the phrase "$f(x)$ is a function" will mean "$f$ is a function of the variable $x$," unless explicitly stated otherwise.

## 2 Base Event Generator, Groundwork for ARCANE Reweighting

The base (i.e., prior to applying ARCANE reweighting) event-generation pipeline used is this paper is taken directly from Ref. [16], which is a tutorial (with an accompanying codebase) on matching NLO calculations with parton showers and a few other related topics. In this section, the mechanics of the event generation pipeline will be described in sufficient detail so as to understand the subsequent implementation of ARCANE reweighting. Monte Carlo collider-event-generation experts may be able to skip Section 2.1 and Sections 2.3-2.6 and still follow the rest of the paper. Additional details needed to reproduce the present work can be found in Ref. [16] and/or the codebase associated with the present work. The physics motivations and formalisms behind the generation pipeline can be found in Refs. [16,17].

### 2.1 General Overview of the Generation Pipeline

The events of interest in this paper contain $e^+e^-$ in the initial state and contain either $q\bar{q}g$ or $q\bar{q}$ in the final state. The event-attributes of interest, ultimately, are the momenta, flavors, and colors of the final state particles, as well as the starting scale for the subsequent parton showering, which will be discussed later. For each event, the flavor of the quark $q$ will be one of $d$, $u$, $s$, $c$, and $b$. The generator uses the large-number-of-colors approximation. Under this, without loss of generality, the colors of $q$, $\bar{q}$, and $g$ can be set to $(1,0)$ and $(0,2)$, and $(2,1)$, respectively, in all events with the $q\bar{q}g$ final state.[2] Likewise, the colors of $q$ and $\bar{q}$ can be set to $(1,0)$ and $(0,1)$, respectively, in all events with the $q\bar{q}$ final state. All initial and final state particles are taken to be massless in the event kinematics. The flowchart in Figure 1 provides a general overview of the pipeline for generating a single event, which roughly proceeds as follows:

- First the "event-type" attribute is chosen randomly from the set $\{\mathbb{H},\mathbb{S}\}$. Here, $\mathbb{H}$-events and $\mathbb{S}$-events stand for "hard remainder" and "standard" events, respectively.

- If the event is chosen to be of the $\mathbb{H}$-type, a "resolved event" $e^+e^- \longrightarrow q\bar{q}g$ is sampled. On the other hand, if the event is chosen to be of the $\mathbb{S}$-type, an "unresolved event" $e^+e^- \longrightarrow q\bar{q}$ is sampled. Here, sampling an event means to sample the particles' momenta as well as the quark flavor.

- Subsequently, the events undergo parton showers, which produce additional quarks and/or gluons, until a pre-chosen infra-red cutoff scale, specified by the parameter $\ell_{\text{cutoff}}$, is reached. The parton shower will begin from an event-specific starting scale $\ell_{\text{start}}$. The outcome of this sampling procedure is one parton-level event.

The events are sampled and weighted by the generator in such a way that they correspond to a specific theoretical/phenomenological/heuristic model (with specific values for the model parameters). The ingredients of this model include the Standard Model of particle physics and various aspects of perturbative Quantum Chromodynamics (QCD) [18], including the Catani–Seymour dipole factorization [19], the MC@NLO formalism for next-to-leading-order event generation, running of coupling constants, etc. The exact formulas (involving matrix elements, subtraction schemes, dipole splitting functions, etc.) used to compute the weight of an event (with a completely-specified MC-history) are *mostly* irrelevant to the present work, and will only be discussed as needed. Due to the nature of the NLO matching performed by MC@NLO, some $\mathbb{H}$-type events have negative weights for this collision process; this is the problem being tackled in this paper.

---

[2]In a color tuple $(x,y)$, $x$ is the color charge and $y$ is the anticolor charge. The different colors are indexed as $1,2,\ldots$, and a value of 0 indicates no color charge.

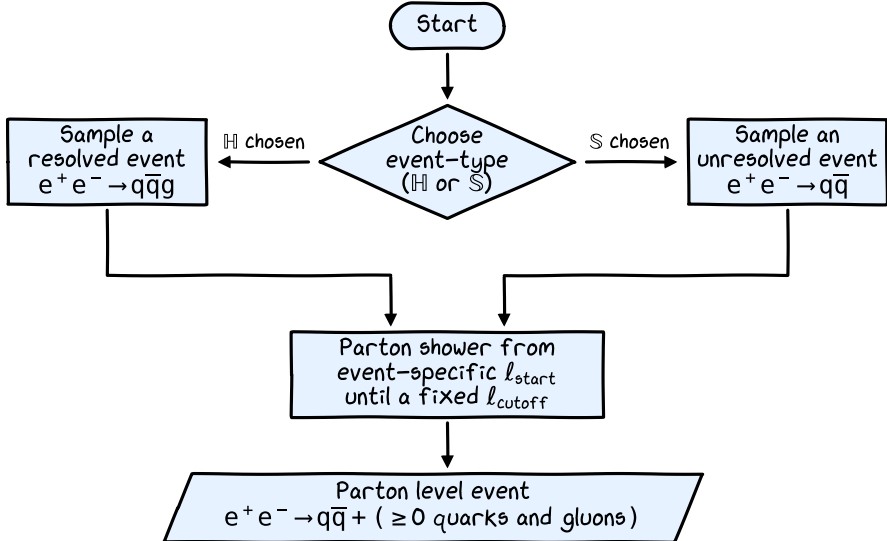

Figure 1: Flowchart depicting a general overview of the event-generation pipeline for producing a parton-level event, for the process under consideration, using the MC@NLO formalism.

The event-type attribute is not relevant beyond the parton-level event-generator, i.e., will not be used in subsequent processing of the event, e.g., in fragmentation and hadronization, detector and electronics simulations, object reconstructions, and physics analyses. One can have two functionally identical parton-level events (ignoring differences in weights) be of different event-types. The idea in this paper is to redistribute the "contributions" of the $\mathbb{H}$- and $\mathbb{S}$-pathways in the event generator to a given parton-level event, in order to reduce the negative weights problem.

## 2.2 Choosing a "Stopping Point" for Performing ARCANE Reweighting

In order to do the redistribution of contributions correctly, one needs to pick a stopping point (or criterion), for all possible pathways within the event generator and identify the visible and hidden event-attributes, denoted by $V$ and $L$ respectively, at this point of the generator. While a convenient choice of stopping point for the example under consideration may be obvious to the reader, it could be instructive to discuss a few bad stopping points first. Consider the following stopping point: immediately after sampling a resolved event for the $\mathbb{H}$-pathway and immediately after sampling an unresolved event for $\mathbb{S}$-pathway. The event-type can be treated as hidden at this point, so moving contributions from $\mathbb{S}$- to $\mathbb{H}$-pathways is acceptable. However, there is no overlap in the distributions of $\mathbb{H}$- and $\mathbb{S}$-type events at this point, since their final states are different. So it is not possible to perform the desired redistribution, making this an ineffective choice of stopping point.

Another possible stopping point is immediately after the complete parton showering process. Beyond this, the event-type can be treated as hidden. Furthermore, there is significant overlap between the distributions of $\mathbb{H}$- and $\mathbb{S}$-type events at this point. So this is a viable point in the generator for performing ARCANE reweighting. However, there could be a lot of emissions in the parton showering stage. This increases the complexity of the Monte Carlo event-history to be tracked and handled; it is preferable to reduce this complexity if possible.

The stopping point used in this paper is the following: a) immediately after sampling a resolved event for the $\mathbb{H}$-pathway and b) immediately after a single parton-shower emission step, following the sampling of an unresolved event, for the $\mathbb{S}$-pathway. The emission step may

or may not result in a successful emission; this will be discussed later. A successful emission leads to a q$\bar{\text{q}}$g final state in the $\mathbb{S}$-pathway (with equivalent color structure as in the resolved $\mathbb{H}$-type events). As before, the event-type can be treated as hidden and there is an overlap between the distributions of $\mathbb{H}$- and $\mathbb{S}$-events. So this is a viable (and a relatively simple) point in the generation pipeline for redistributing the contributions of $\mathbb{H}$- and $\mathbb{S}$-pathways. The different components of the generation pipeline, up to the chosen stopping point, will be discussed next.

## 2.3   Unresolved Event Kinematics

This subsection describes the kinematics of an unresolved (or leading-order) event $\text{e}^+\text{e}^- \longrightarrow \text{q}\bar{\text{q}}$. The four-momenta of the incident colliding particles $\text{e}^-$ and $\text{e}^+$ are given by

$$p_{\text{e}^-} \equiv \frac{E_{\text{cms}}}{2\,c}\left(1,0,0,1\right) \qquad \text{and} \qquad p_{\text{e}^+} \equiv \frac{E_{\text{cms}}}{2\,c}\left(1,0,0,-1\right), \tag{7}$$

respectively, where $c$ is the speed of light and $E_{\text{cms}}$ is the total energy of colliding particles in their center-of-momentum frame, which is also the lab frame; $E_{\text{cms}}$ is a fixed parameter of the event generator. The four momenta of the daughter particles q and $\bar{\text{q}}$ are given by

$$p_{\text{q}} \equiv \frac{E_{\text{cms}}}{2\,c}\left(1\,,\ \sin(\theta_{\text{lo}})\cos(\varphi_{\text{lo}})\,,\ \sin(\theta_{\text{lo}})\sin(\varphi_{\text{lo}})\,,\ \cos\theta_{\text{lo}}\right) \qquad \text{and} \tag{8}$$

$$p_{\bar{\text{q}}} \equiv \frac{E_{\text{cms}}}{2\,c}\left(1\,,\ -\sin(\theta_{\text{lo}})\cos(\varphi_{\text{lo}})\,,\ -\sin(\theta_{\text{lo}})\sin(\varphi_{\text{lo}})\,,\ -\cos\theta_{\text{lo}}\right), \tag{9}$$

respectively, where $\theta_{\text{lo}} \in [0,\pi]$ and $\varphi_{\text{lo}} \in [0,\pi)$ are angles parameterizing the collision kinematics. Note that given $p_{\text{q}}$ and $p_{\bar{\text{q}}}$, the angles $\theta_{\text{lo}}$ and $\varphi_{\text{lo}}$ are uniquely determined.

**Phase-space element.**    The following fact is noted here for later use. For two massless particles i and j constrained to satisfy $p_{\text{i}} + p_{\text{j}} = p_{\text{tot}}$, where $p_{\text{tot}}$ is a timelike four-momentum, the Lorentz-invariant phase space elements (for two different parametrizations) are related by[3]

$$\frac{\text{d}^3\vec{p}_{\text{i}}}{(2\,\pi)^3\,2\,E_{\text{i}}}\,\frac{\text{d}^3\vec{p}_{\text{j}}}{(2\,\pi)^3\,2\,E_{\text{j}}}\,\delta^{(4)}\big(p_{\text{i}} + p_{\text{j}} - p_{\text{tot}}\big) = \frac{\text{d}\cos(\theta_{\text{cms}})\ \text{d}\varphi_{\text{cms}}}{8\,(2\,\pi)^6}, \tag{10}$$

where $\theta_{\text{cms}}$ and $\varphi_{\text{cms}}$ are the spherical angles (in some orientation of axes) of the three-momentum of, say, particle i in the center-of-momentum frame of $p_{\text{tot}}$. Here, (i) $E_{\langle\text{subscript}\rangle}$ and $\vec{p}_{\langle\text{subscript}\rangle}$ are the energy and three-momentum components, respectively, of $p_{\langle\text{subscript}\rangle}$ and (ii) $\delta^{(4)}$ is the four-dimensional Dirac delta function.

## 2.4   Emission Kinematics

Let us a consider a generic emission process $\{\text{ij}\}\,\text{k} \longrightarrow \text{i}\,\text{j}\,\text{k}$. Here $\{\text{ij}\}$, i, j, and k represent the emitter particle before the emission, the emitter particle after the emission, the emitted "daughter" particle, and the "spectator" particle, respectively; all these particles are taken to be massless. Let $\tilde{p}_{\{\text{ij}\}}$ and $\tilde{p}_{\text{k}}$ be the four-momenta of the particles $\{\text{ij}\}$ and k, respectively, before the emission. Let $p_{\text{i}}$, $p_{\text{j}}$, and $p_{\text{k}}$ be the four-momenta of the particles i, j, and k, respectively, after the emission. Let $M$ be the total energy of the emission system in its center-of-momentum frame:

$$M^2 \equiv \big(\tilde{p}_{\{\text{ij}\}} + \tilde{p}_{\text{k}}\big)^2 = \big(p_{\text{i}} + p_{\text{j}} + p_{\text{k}}\big)^2 = 2\,\tilde{p}_{\{\text{ij}\}}\,\tilde{p}_{\text{k}} = 2\big(p_{\text{i}}\,p_{\text{j}} + p_{\text{j}}\,p_{\text{k}} + p_{\text{k}}\,p_{\text{i}}\big) \tag{11}$$

---

[3]Using an equality sign in (10) is a convenient abuse of notation, albeit a benign one.

The emission process is parameterized by the variables $y \in [0, 1]$, $z \in [0, 1]$, and $\varphi \in [0, 2\pi]$. The final momenta are given, in terms of the initial momenta and the emission-parameters, by

$$p_\mathtt{i} \equiv z \, \tilde{p}_{\{\mathtt{ij}\}} + (1 - z) \, y \, \tilde{p}_\mathtt{k} + k_T \, , \tag{12}$$

$$p_\mathtt{j} \equiv (1 - z) \tilde{p}_{\{\mathtt{ij}\}} + z \, y \, \tilde{p}_\mathtt{k} - k_T \, , \tag{13}$$

$$p_\mathtt{k} \equiv (1 - y) \tilde{p}_\mathtt{k} \, . \tag{14}$$

Here $k_T$ is a four-momentum determined by the initial momenta and emission-parameters, and satisfies the following conditions:

$$k_T^2 = - M^2 \, y \, z \, (1 - z) \, , \qquad \text{and} \qquad \tilde{p}_{\{\mathtt{ij}\}} \, k_T = \tilde{p}_\mathtt{k} \, k_T = 0 \, . \tag{15}$$

These conditions ensure that $p_\mathtt{i}^2 = p_\mathtt{j}^2 = 0$. In the center-of-momentum frame of the emission process, $k_T$ is given by

$$k_T^{\text{cms}} \equiv (0, \vec{k}_T^{\text{cms}}) \, , \tag{16}$$

where $\vec{k}_T^{\text{cms}}$ is a three-vector that lies in the plane $\mathbb{P}_\perp$ perpendicular to the emitter's direction of travel (a) before emission (b) in the center-of-momentum frame of the emission process.[4] The magnitude and direction of $\vec{k}_T^{\text{cms}}$ are given, respectively, by

$$\left| \vec{k}_T^{\text{cms}} \right| \equiv \sqrt{M^2 \, y \, z \, (1 - z)} \qquad \text{and} \qquad \hat{k}_T^{\text{cms}} \equiv \cos(\varphi) \, \hat{\kappa}_1 + \sin(\varphi) \, \hat{\kappa}_2 \, , \tag{17}$$

where $\hat{\kappa}_1$ and $\hat{\kappa}_2$ are unit-vectors that (a) lie in $\mathbb{P}_\perp$, (b) are perpendicular to each other, and (c) are completely determined by $\tilde{p}_{\{\mathtt{ij}\}}$ and $\tilde{p}_\mathtt{k}$. Note that given a valid choice of $(p_\mathtt{i}, p_\mathtt{j}, p_\mathtt{k})$, the value of $(\tilde{p}_{\{\mathtt{ij}\}}, \tilde{p}_\mathtt{k}, y, z, \varphi)$ is uniquely determined, and can be computed, e.g., as follows:[5]

$$\tilde{p}_{\{\mathtt{ij}\}} \equiv p_\mathtt{i} + p_\mathtt{j} - \left( \frac{p_\mathtt{i} \, p_\mathtt{j}}{p_\mathtt{j} \, p_\mathtt{k} + p_\mathtt{k} \, p_\mathtt{i}} \right) p_\mathtt{k} \, , \qquad \tilde{p}_\mathtt{k} \equiv \left( \frac{p_\mathtt{i} \, p_\mathtt{j} + p_\mathtt{j} \, p_\mathtt{k} + p_\mathtt{k} \, p_\mathtt{i}}{p_\mathtt{j} \, p_\mathtt{k} + p_\mathtt{k} \, p_\mathtt{i}} \right) p_\mathtt{k} \, , \tag{18a}$$

$$y \equiv \frac{p_\mathtt{i} \, p_\mathtt{j}}{p_\mathtt{i} \, p_\mathtt{j} + p_\mathtt{j} \, p_\mathtt{k} + p_\mathtt{k} \, p_\mathtt{i}} \, , \qquad z \equiv \frac{p_\mathtt{k} \, p_\mathtt{i}}{p_\mathtt{j} \, p_\mathtt{k} + p_\mathtt{k} \, p_\mathtt{i}} \, , \tag{18b}$$

$$k_T \equiv \left( \frac{p_\mathtt{j} \, p_\mathtt{k}}{p_\mathtt{j} \, p_\mathtt{k} + p_\mathtt{k} \, p_\mathtt{i}} \right) p_\mathtt{i} - \left( \frac{p_\mathtt{k} \, p_\mathtt{i}}{p_\mathtt{j} \, p_\mathtt{k} + p_\mathtt{k} \, p_\mathtt{i}} \right) p_\mathtt{j} - \left( \frac{(p_\mathtt{i} \, p_\mathtt{j})(p_\mathtt{j} \, p_\mathtt{k} - p_\mathtt{k} \, p_\mathtt{i})}{(p_\mathtt{j} \, p_\mathtt{k} + p_\mathtt{k} \, p_\mathtt{i})^2} \right) p_\mathtt{k} \tag{18c}$$

Some other quantities of interest are the energy scale parameters $t$ and $\ell$ defined as

$$t \equiv - k_T^2 = M^2 \, y \, z \, (1 - z) = \frac{2 (p_\mathtt{i} \, p_\mathtt{j})(p_\mathtt{j} \, p_\mathtt{k})(p_\mathtt{k} \, p_\mathtt{i})}{(p_\mathtt{j} \, p_\mathtt{k} + p_\mathtt{k} \, p_\mathtt{i})^2} \qquad \text{and} \tag{19a}$$

$$\ell \equiv \ln \left( \frac{t}{1 \, \text{GeV}^2/c^2} \right) \, , \tag{19b}$$

respectively. The emission-kinematics can be parameterized by $(t, z, \varphi)$ (or, equivalently, by $(\ell, z, \varphi)$) instead of $(y, z, \varphi)$, with the domain of $(t, z, \varphi)$ given by

$$0 \leq t \leq \frac{M^2}{4} \, , \qquad z_- \leq z \leq z_+ \, , \qquad 0 \leq \varphi < 2\pi \, , \tag{20a}$$

$$\text{where} \qquad z_\pm \equiv \frac{1}{2} \left( 1 \pm \sqrt{1 - \frac{4 \, t}{M^2}} \, \right) \, . \tag{20b}$$

---

[4]Equivalently, $\vec{k}_T^{\text{cms}}$ lies in the plane perpendicular to the spectator's direction of travel (a) before or after the emission (b) in the center-of-momentum frame of the emission process.

[5]If needed, $\varphi$ can be computed using $(k_T, \tilde{p}_{\{\mathtt{ij}\}}, \tilde{p}_\mathtt{k})$, by computing $(\vec{k}_T^{\text{cms}}, \hat{\kappa}_1, \hat{\kappa}_2)$ as an intermediate step. However, typically, and in this paper, the relevant conditional quasi (i.e., weighted) densities and sampling densities are all independent of $\varphi$, so it will not need to be computed for the purposes of this paper.

**Jacobian Factors.**   The Jacobian factors for transforming from $y$ to $t$ and $\ell$ are given, respectively, by

$$\left|\frac{\partial t}{\partial y}\right| = \frac{t}{y} = \frac{M^2 z (1-z)}{y}, \qquad\qquad \left|\frac{\partial \ell}{\partial y}\right| = \frac{1}{y}. \tag{21}$$

**Phase-space element.**   For the case where the total four-momentum of the emission-system is constrained to equal a timelike $p_{\text{tot}}$, the Lorentz-invariant phase space elements in terms of $\left(\vec{p}_{\text{i}}, \vec{p}_{\text{j}}, \vec{p}_{\text{k}}\right)$ and $\left(\vec{\tilde{p}}_{\{\text{ij}\}}, \vec{\tilde{p}}_{\{\text{k}\}}, y, z, \varphi\right)$ are related as follows [19]:

$$
\begin{aligned}
&\frac{\mathrm{d}^3\vec{p}_{\text{i}}}{(2\pi)^3\,2\,E_{\text{i}}} \frac{\mathrm{d}^3\vec{p}_{\text{j}}}{(2\pi)^3\,2\,E_{\text{j}}} \frac{\mathrm{d}^3\vec{p}_{\text{k}}}{(2\pi)^3\,2\,E_{\text{k}}}\, \delta^{(4)}\!\left(p_{\text{i}} + p_{\text{j}} + p_{\text{k}} - p_{\text{tot}}\right) \\
&= \left[\frac{\mathrm{d}^3\vec{\tilde{p}}_{\{\text{ij}\}}}{(2\pi)^3\,2\,\tilde{E}_{\{\text{ij}\}}} \frac{\mathrm{d}^3\vec{\tilde{p}}_{\text{k}}}{(2\pi)^3\,2\,\tilde{E}_{\text{k}}}\, \delta^{(4)}\!\left(\tilde{p}_{\{\text{ij}\}} + \tilde{p}_{\text{k}} - p_{\text{tot}}\right)\right]\left[\frac{M^2(1-y)}{16\,\pi^2}\,\mathrm{d}y\,\mathrm{d}z\,\frac{\mathrm{d}\varphi}{2\pi}\right].
\end{aligned}
\tag{22}
$$

## 2.5   Overview of the Parton Showering Procedure

This work uses transverse-momentum-ordered parton showers, following Ref. [16]. The parton showering of an event involves a sequence of zero or more emissions, each of which increases the number of particles in the final state by 1. The input to the parton showering process are (a) the event prior to parton showering, (b) a global emission scale cutoff $\ell_{\text{cutoff}}$, and (c) an event-specific starting scale $\ell_{\text{start}} > \ell_{\text{cutoff}}$ for the first emission of the shower. At each step, one either (a) performs an emission at a scale $\ell$ satisfying $\ell_{\text{start}} > \ell > \ell_{\text{cutoff}}$, or (b) terminates the emission process and reports that the cutoff scale $\ell_{\text{cutoff}}$ has been reached. If an emission occurs in a given step, then the scale of the emission, namely $\ell$, is set as the starting scale $\ell_{\text{start}}$ for the next emission. The showering process continues until one of the emission attempts results in the cutoff scale being reached.

At a given step, based on the flavors and colors of the particles in the final state (prior to the attempted emission), different emission "channels" indexed by $\{1, \ldots, C\}$ may be available. Each emission channel is characterized by an emitter particle, a spectator particle, and a daughter particle (as defined in Section 2.4). To perform an emission, one needs to pick a channel $c \in \{1, \ldots, C\}$ and valid emission parameters $(\ell, z, \varphi)$ which satisfy (a) $\ell_{\text{start}} > \ell > \ell_{\text{cutoff}}$ and (b) the constraint (20) for the chosen channel $c$. Then the emission corresponding to $(c, \ell, z, \varphi)$ is performed; the kinematics of the emission are as discussed in Section 2.4. In addition to adding a new particle to final state and changing the particles' momenta, the emission will also change the color and possibly the flavor of the emitter particle. Note that after each emission, the set of available channels for the succeeding emission changes.

As mentioned in Section 2.2, this paper is only interested in the first parton shower emission step, for $\mathbb{S}$-type events. There are two available channels for this emission. $c = 1$ corresponds to the emitter and spectator being q and q̄, respectively. Likewise, $c = 2$ corresponds to the emitter and spectator being q̄ and q, respectively. In both cases, the daughter particle is g. The flavor of the emitter is not changed by an emission from these channels. The mechanics of sampling a single parton shower emission is described next.

## 2.6   Single Parton Shower Emission With the Veto Algorithm

This section discusses the weighted veto algorithm for performing a single parton shower emission step, with multiple "competing" emission channels [17, 20–23]. Understanding the mechanics of this algorithm is crucial for the subsequent implementation of ARCANE reweighting in this paper. Given $(\ell_{\text{start}}, \ell_{\text{cutoff}})$ and a set of competing emission channels, the goal of this

algorithm is to sample the emission channel $c$ and the emission parameters $\ell$ and $z$. The emission parameter $\varphi$ will be sampled separately, independent of all else;[6] this is not considered as part of the veto algorithm here. The outcome of the emission scale cutoff being reached (and no emission occurring) will be indicated by returning $\ell_{\text{cutoff}}$ as $\ell$, with $(c,z)$ sampled from some arbitrary distribution. The value $(c,z)$ will not be used subsequently if the returned $\ell$ equals $\ell_{\text{cutoff}}$.[7] The sampled $(\ell,c,z)$ should correspond to a valid emission if $\ell \neq \ell_{\text{cutoff}}$.

Each emission channel $c$ has an associated "$z$-dependent splitting kernel function", say $f_c(\ell,z)$. The parton shower step is characterized by the functions $f_1,\ldots,f_C$, which will be collectively denoted as $\{f\}$. The function $f_c$ implicitly depends on the properties of the emitter, spectator, and daughter particles of channel $c$, specifically their flavors and their total energy in the center-of-momentum frame. The choice of $f_c$-s used in this work will be described in Section 3.5. The value of $f_c(\ell,z)$ can roughly be thought of as the rate of the emission corresponding to $(\ell,c,z)$ occurring, provided an emission has not already occurred at a higher energy scale. Concretely, the target quasi density of $(\ell,c,z)$ for this sampling procedure is $Q_{\{f\}}(\,\cdot\,;\ell_{\text{start}},\ell_{\text{cutoff}})$ defined as follows:

$$\kappa_{f_c}(\ell) \equiv \int_{\mathbb{R}} \mathrm{d}z\, f_c(\ell,z), \tag{23a}$$

$$\Delta_{f_c}(\ell,\ell_{\text{start}}) \equiv \exp\!\left(-\int_{\ell}^{\ell_{\text{start}}} \mathrm{d}\ell'\left[\kappa_{f_c}(\ell')\,\Theta(\ell_{\text{start}}-\ell')\right]\right), \tag{23b}$$

$$\kappa_{\{f\}}(\ell) \equiv \sum_{c=1}^{C} \int_{\mathbb{R}} \mathrm{d}z\, f_c(\ell,z) = \sum_{c=1}^{C} \kappa_{f_c}(\ell), \tag{23c}$$

$$\Delta_{\{f\}}(\ell,\ell_{\text{start}}) \equiv \exp\!\left(-\int_{\ell}^{\ell_{\text{start}}} \mathrm{d}\ell'\left[\kappa_{\{f\}}(\ell')\,\Theta(\ell_{\text{start}}-\ell')\right]\right) \tag{23d}$$

$$= \prod_{c=1}^{C} \Delta_{f_c}(\ell,\ell_{\text{start}}), \tag{23e}$$

$$q_{\{f\}}^{\text{no-cutoff}}(\ell\,;\ell_{\text{start}}) \equiv \Delta_{\{f\}}(\ell,\ell_{\text{start}})\,\kappa_{\{f\}}(\ell)\,\Theta(\ell_{\text{start}}-\ell), \tag{23f}$$

$$Q_{\{f\}}^{\text{no-cutoff}}(\ell,c,z\,;\ell_{\text{start}}) \equiv q_{\{f\}}^{\text{no-cutoff}}(\ell\,;\ell_{\text{start}})\,\frac{f_c(\ell,z)}{\kappa_{\{f\}}(\ell)} \tag{23g}$$

$$= \Delta_{\{f\}}(\ell,\ell_{\text{start}})\,f_c(\ell,z)\,\Theta(\ell_{\text{start}}-\ell), \tag{23h}$$

$$q_{\{f\}}(\ell\,;\ell_{\text{start}},\ell_{\text{cutoff}}) \equiv \Theta(\ell-\ell_{\text{cutoff}})\,q_{\{f\}}^{\text{no-cutoff}}(\ell\,;\ell_{\text{start}}) \\ + \Delta_{\{f\}}(\ell_{\text{cutoff}}\,;\ell_{\text{start}})\,\delta(\ell-\ell_{\text{cutoff}}), \tag{23i}$$

$$Q_{\{f\}}(\ell,c,z\,;\ell_{\text{start}},\ell_{\text{cutoff}}) \equiv \Theta(\ell-\ell_{\text{cutoff}})\,Q_{\{f\}}^{\text{no-cutoff}}(\ell,c,z\,;\ell_{\text{start}}) \\ + \Delta_{\{f\}}(\ell_{\text{cutoff}}\,;\ell_{\text{start}})\,\delta(\ell-\ell_{\text{cutoff}})\,\mathtt{arbit\_unitnorm}(c,z), \tag{23j}$$

where $\mathbb{R}$ is the set of real numbers, $\delta$ is the Dirac delta function and $\Theta$ is the Heaviside step function defined as

$$\Theta(x) \equiv \begin{cases} 1 & \text{if } x > 0, \\ 0 & \text{if } x \leq 0, \end{cases} \tag{24}$$

---

[6]This is true for parton showers based on spin-averaged splitting kernels, like the ones used in this paper. In general, splitting functions could have an azimuthal dependence [18, 24]; the veto algorithm description in this paper can be modified straightforwardly to accommodate such cases.

[7]In practice, $c$ and $z$ need not even be sampled in this case. They are assumed to be sampled here for notational simplicity; this way, one can associate a probability density and a quasi density with $(\ell,c,z)$ under the overall emission-or-lack-there-of process.

and `arbit_unitnorm` is an arbitrary unit-normalized quasi density of $(c, z)$. The value of $(c, z)$ sampled from `arbit_unitnorm` will not actually be used, since it corresponds to the case of the cutoff scale being reached. The $f_c$-s are known to satisfy $\Delta_{\{f\}}(\ell, \ell_{\text{start}}) > 0$ for all $\ell_{\text{start}} \geq \ell > \ell_{\text{cutoff}}$.

$Q_{\{f\}}$ can be thought of as a generalization of the exponential distribution with a non-constant rate (among other added intricacies). From (23c), it can be seen that $c$ and $z$ play somewhat similar roles in this setup—they both parameterize the various emission options available at a given scale, with $c$ being a discrete index and $z$ being a continuous parameter.[8]

In the definition of $Q_{\{f\}}$ above, each $f_c$ is assumed to be 0 for $(\ell, z)$ outside the corresponding kinematically allowed region defined by (20); this way the integrals with respect to $z$ in (23a) and (23c) can be simply written as being over the domain $\mathbb{R}$. The $f_c$-s used in this paper may differ from $z$-dependent kernels used elsewhere in the literature by some prefactors, including Jacobian factors and coupling constants, as well as Heaviside step functions needed to set $f_c$ to 0 in kinematically disallowed regions. It can be seen from (23) that

$$\frac{\partial \Delta_{\{f\}}(\ell, \ell_{\text{start}})}{\partial \ell} = \Delta_{\{f\}}(\ell, \ell_{\text{start}}) \, \kappa_{\{f\}}(\ell) \, \Theta(\ell_{\text{start}} - \ell) = q_{\{f\}}^{\text{no-cutoff}}(\ell \, ; \ell_{\text{start}}), \quad (25)$$

$$\implies \quad \int_{\ell}^{\ell_{\text{start}}} \mathrm{d}\ell \sum_{c=1}^{C} \int_{\mathbb{R}} \mathrm{d}z \, Q_{\{f\}}^{\text{no-cutoff}}(\ell, c, z \, ; \ell_{\text{start}}) = \int_{\ell}^{\ell_{\text{start}}} \mathrm{d}\ell \, q_{\{f\}}^{\text{no-cutoff}}(\ell \, ; \ell_{\text{start}})$$

$$= 1 - \Delta_{\{f\}}(\ell, \ell_{\text{start}}). \quad (26)$$

This shows that $\Delta_{\{f\}}(\ell, \ell_{\text{start}})$ can be roughly interpreted as the probability of no emission occurring until the scale $\ell$, starting from $\ell_{\text{start}}$. The remaining functions in (23) can be roughly interpreted as follows:

- $\kappa_{f_c}(\ell)$ can be thought of as the rate of the emission corresponding to $(\ell, c)$ occurring, provided an emission has not already occurred at a higher energy scale. It will be referred to as the "$z$-independent splitting kernel function".

- $\Delta_{f_c}(\ell, \ell_{\text{start}})$ can be thought of as the probability of no emission occurring until the scale $\ell$, starting from $\ell_{\text{start}}$, assuming that only the channel $c$ is available for emission.

- $\kappa_{\{f\}}(\ell)$ can be thought of as the rate of emission at the scale $\ell$, provided an emission has not already occurred at a higher energy scale.

- $q_{\{f\}}^{\text{no-cutoff}}(\ell, \ell_{\text{start}})$ can be thought of as the quasi density of $\ell$, the scale at which the first emission occurs, starting from $\ell_{\text{start}}$, without enforcing any cutoffs on $\ell$.

- $Q_{\{f\}}^{\text{no-cutoff}}(\ell, \ell_{\text{start}})$ can be thought of as the quasi density of $(\ell, c, z)$, starting from $\ell_{\text{start}}$, without enforcing any cutoffs on $\ell$.

- $q_{\{f\}}$ and $Q_{\{f\}}$ modify $q_{\{f\}}^{\text{no-cutoff}}$ and $Q_{\{f\}}^{\text{no-cutoff}}$, respectively, to enforce the cutoff scale $\ell_{\text{cutoff}}$.

$Q_{\{f\}}(\, \cdot \, ; \ell_{\text{start}}, \ell_{\text{cutoff}})$ is unit-normalized[9] (but not necessarily non-negative) for any choice of functions $f_1, \dots, f_C$ and any $(\ell_{\text{start}}, \ell_{\text{cutoff}})$; this can be seen from (23j) and (26). Furthermore, if $\lim_{\ell \to -\infty} \Delta_{\{f\}}(\ell, \ell_{\text{start}}) = 0$, then $Q_{\{f\}}^{\text{no-cutoff}}(\, \cdot \, ; \ell_{\text{start}})$ is also unit-normalized; this can be seen from (26).

---

[8]However, $c$ and $z$ are often handled differently in the proposal-sampling step of the veto algorithm.

[9]One consequence of this property is that incorporating a parton shower will not change the total cross-section of a process.

**Sampling Emission Parameters as per** $Q_{\{f\}}$**.**    In some special cases, depending on the properties of $f_c$ (e.g., non-negativity) and the ability to compute certain functions in (23) (and/or their inverses), one can employ relatively straightforward techniques like inverse transform sampling to sample emission parameters as per $Q_{\{f\}}$. However, in practice, the kernel functions $f_c$ may be too complicated to employ such techniques. In such cases, one can use the weighted Sudakov veto algorithm, which covers the general case, where

- $f_c(\ell, z)$ can be positive, negative, or zero for different values of $(\ell, c, z)$, and

- $f_c(\ell, z)$ is assumed be directly computable, e.g., using a closed-form expression, but other functions in (23), like $\kappa_{f_c}, \Delta_{f_c}, \kappa_{\{f\}}, \Delta_{\{f\}}, q_{\{f\}}^{\text{no-cut}}, Q_{\{f\}}^{\text{no-cut}}$, or their inverses (where applicable) are not assumed to be directly computable.

The working of the veto algorithm can be understood as follows. Let $g_1, \ldots, g_C$ be functions of $(\ell, z)$ such that $g_c(\ell, z) \neq 0$ for all $(\ell, c, z)$ satisfying $f_c(\ell, z) \neq 0$. Furthermore, let (i) $\lim_{\ell \to -\infty} \Delta_{\{g\}}(\ell, \ell_{\text{start}}) = 0$ and (ii) $\Delta_{\{g\}}(\ell, \ell_{\text{start}}) > 0$ for all $\ell_{\text{start}} \geq \ell > -\infty$.[10] Now, $Q_{\{f\}}$ can be expressed as a recursive chain of functions as follows [23]:[11]

$$
\begin{aligned}
Q_{\{f\}}(\ell, c, z \,; \ell_{\text{start}}, \ell_{\text{cutoff}}) = \int_\ell^{\ell_{\text{start}}} \mathrm{d}\ell' \sum_{c'=1}^C \int_{\mathbb{R}} \mathrm{d}z' \, Q_{\{g\}}^{\text{no-cutoff}}(\ell', c', z' \,; \ell_{\text{start}}) \, \Bigg[ \\
\Big[ 1 - \Theta(\ell' - \ell_{\text{cutoff}}) \Big] \qquad\qquad \times \delta(\ell - \ell_{\text{cutoff}}) \, \texttt{arbit\_unitnorm}(c, z) \\
+ \ \Theta(\ell' - \ell_{\text{cutoff}}) \frac{f_{c'}(\ell', z')}{g_{c'}(\ell', z')} \qquad\quad \times \delta(\ell - \ell') \, \delta(z - z') \, \delta_{\mathrm{K}}(c, c') \\
+ \ \Theta(\ell' - \ell_{\text{cutoff}}) \left( 1 - \frac{f_{c'}(\ell', z')}{g_{c'}(\ell', z')} \right) \times Q_{\{f\}}(\ell, c, z \,; \ell', \ell_{\text{cutoff}}) \\
\Bigg],
\end{aligned}
\tag{27}
$$

where $\delta_{\mathrm{K}}$ is the Kronecker delta function and $Q_{\{g\}}^{\text{no-cutoff}}$ is defined similarly to $Q_{\{f\}}^{\text{no-cutoff}}$ in (23), with $f_c$-s replaced by $g_c$-s. One can choose the functions $g_1, \ldots, g_C$ conveniently so that $Q_{\{g\}}^{\text{no-cutoff}}(\,\cdot\,; \ell_{\text{start}})$ can be straightforwardly sampled using an alternative technique, cf. Refs. [21–23]; the details of this alternative technique are irrelevant for the purposes of this paper. Now, the Forward Chain Monte Carlo procedure represented by (27) can be used to sample $(\ell, c, z)$ as per $Q_{\{f\}}(\,\cdot\,; \ell_{\text{start}}, \ell_{\text{cutoff}})$—this is the weighted Veto algorithm and is described explicitly in Algorithm 1.

Line 1 of (27) corresponds to sampling a "proposal", namely $(\ell', c', z')$, for the emission parameters as per $Q_{\{g\}}^{\text{no-cutoff}}(\,\cdot\,; \ell_{\text{start}})$.[12] Line 2 involves handling the case where the proposed $\ell'$ is not above $\ell_{\text{cutoff}}$, by appropriately terminating the sampling process. Lines 3 and 4 can be implemented using an accept-reject-filter on the proposal. Let the proposal be accepted with a probability $\alpha_{c'}(\ell', z')$.[13] If the proposal is accepted, the event-weight is multiplied by a factor of $\dfrac{f_{c'}(\ell', z')}{g_{c'}(\ell', z')} \times \dfrac{1}{\alpha_{c'}(\ell', z')}$ and proposal $(\ell', c', z')$ is returned as $(\ell, c, z)$; this captures line 3. If the proposal is rejected, the event-weight is multiplied by a factor of

---

[10]Condition (i) ensures that $Q_{\{g\}}^{\text{no-cutoff}}(\,\cdot\,; \ell_{\text{start}})$ is unit-normalized and condition (ii) ensures that the veto algorithm will terminate and not get stuck at a scale above $\ell_{\text{cutoff}}$.

[11]In line 1 of (27), $Q_{\{g\}}^{\text{no-cutoff}}(\,\cdot\,; \ell_{\text{start}})$ can also be replaced with $Q_{\{g\}}(\,\cdot\,; \ell_{\text{start}}, \ell_{\text{cutoff}})$ (the convention that $\Theta(0) = 0$ helps here). In that case, the requirement that $Q_{\{g\}}^{\text{no-cutoff}}(\,\cdot\,; \ell_{\text{start}})$ be unit-normalized is not necessary.

[12]Somewhat interestingly, the algorithm will work even if one uses different choices of $g_c$-s in different proposal steps for the same emission.

[13]In principle, $g_c$-s and $\alpha_c$-s can be dependent on $\ell_{\text{start}}$, or any other previously sampled event-attribute.

---

**Algorithm 1** Veto algorithm for sampling $(\ell, c, z)$ as per $Q_{\{f\}}(\,\cdot\,; \ell_{\text{start}}, \ell_{\text{cutoff}})$

---

1: **prerequisite** A function SAMPLEWEIGHTEDPROPOSAL that takes $(\ell_{\text{start}}, \ell_{\text{cutoff}}, g_1, \ldots, g_C)$ as inputs and returns a pseudorandom weighted datapoint $(W, \ell, c, z)$. This sampling procedure models the quasi density $Q_{\{g\}}^{\text{no-cutoff}}(\,\cdot\,; \ell_{\text{start}}, \ell_{\text{cutoff}})$ for $(\ell, c, z)$. It can only work for certain choices of $(g_1, \ldots, g_C)$, and in particular is known to not work for $g_c \equiv f_c$.

2: **function** WEIGHTEDVETOALGORITHMWITHCUTOFF($\ell_{\text{start}}, \ell_{\text{cutoff}}, W_{\text{start}}$)   *# $W_{\text{start}} = 1$ by default*
3:     **choose**   Convenient $z$-dependent proposal kernel functions $g_1, \ldots, g_C$.
4:             *# Notes:*
5:             *# – $g_c(\ell, z)$ must be non-zero for all $(\ell, c, z)$ such that $f_c(\ell, z) \neq 0$.*
6:             *# – The $g_c$-s should be compatible with the SAMPLEWEIGHTEDPROPOSAL function.*
7:     **choose**   Acceptance probability functions $\alpha_1, \alpha_C$ that take $(\ell, z)$ as input.
8:             *# Notes:*
9:             *# – $\alpha$-s should satisfy $0 \leq \alpha_c(\ell, z) \leq 1$ for all possible values of $(\ell, c, z)$.*
10:             *# – $\alpha_c(\ell, z)$ must be non-zero for all $(\ell, c, z)$ such that $f_c(\ell, z) \neq 0$.*
11:             *# – In principle, $\alpha$ can also depend on other aspects of the MC history of the event.*

12:     **assign**   $W \longleftarrow W_{\text{start}}$

13:     *# The next two lines involve sampling an emission-proposal. They capture line 1 of* (27)*.*
14:     **assign**   $(W_{\text{proposal}}, \ell', c', z') \longleftarrow$ SAMPLEWEIGHTEDPROPOSAL($\ell_{\text{start}}, \ell_{\text{cutoff}}, g_1, \ldots, g_C$)
15:     **assign**   $W \longleftarrow W \times W_{\text{proposal}}$

16:     **if** $\ell' \leq \ell_{\text{cutoff}}$ **then**       *# Emission scale cutoff reached. This block captures line 2 of* (27)*.*
17:         **assign**   $\ell \longleftarrow \ell_{\text{cutoff}}$
18:         **sample**   $(c, z)$ from `arbit_unitnorm` distribution   *# These will not be used subsequently.*
19:         **return**   $(W, \ell, c, z)$

20:     **else**     *# Proposed emission scale is above the cutoff. Proceeding to the proposal accept-reject step.*
21:         **assign**   `accept_prob` $\longleftarrow \alpha_{c'}(\ell', z')$
22:         **sample**   `rand` uniformly from the interval $[0, 1)$

23:         **if** `rand` < `accept_prob` **then**         *# Proposal accepted. This block captures line 3 of* (27)*.*
24:             **assign**   $(\ell, c, z) \longleftarrow (\ell', c', z')$
25:             **assign**   $W \longleftarrow W \times \dfrac{f_{c'}(\ell', z')}{g_{c'}(\ell', z')} \times \dfrac{1}{\texttt{accept\_prob}}$
26:             **return**   $(W, \ell, c, z)$

27:         **else**     *# Proposal rejected. This block captures line 4 of* (27)*.*
28:             **assign**   $W \longleftarrow W \times \left(1 - \dfrac{f_{c'}(\ell', z')}{g_{c'}(\ell', z')}\right) \times \dfrac{1}{1 - \texttt{accept\_prob}}$
29:             **assign**   $(\ell_{\text{next-start}}, W_{\text{next-start}}) \longleftarrow (\ell', W)$
30:             **return**   WEIGHTEDVETOALGORITHMWITHCUTOFF($\ell_{\text{next-start}}, \ell_{\text{cutoff}}, W_{\text{next-start}}$)
31:         **end if**
32:     **end if**
33: **end function**

34: **return**   WEIGHTEDVETOALGORITHMWITHCUTOFF($\ell_{\text{start}}, \ell_{\text{cutoff}}, 1$)

---

$\left(1 - \frac{f_{c'}(\ell', z')}{g_{c'}(\ell', z')}\right) \times \frac{1}{1 - \alpha_{c'}(\ell', z')}$ and one restarts the sampling under $Q_{\{f\}}$, with $\ell'$ set as the new starting scale; this captures line 4. The functions $g_1, \ldots, g_C$ will henceforth be referred to as "$z$-dependent proposal kernel functions" or "proposal kernels" for short.

Let $N_{\text{rej}}$ be the number of proposals rejected by the veto algorithm before a proposal either reached the scale cutoff or was accepted. Let the sequence of proposals considered in the algorithm (including the last one) be indexed as $(\ell_i', c_i', z_i')$ with $i \in \{1, \ldots, N_{\text{rej}} + 1\}$. For the case where the last proposal is accepted, the weight-factor from the veto algorithm is given by

$$
W_{\text{yes-emission}} \equiv \underbrace{\left[\prod_{i=1}^{N_{\text{rej}}+1} W_{\text{proposal},i}\right]}_{\equiv 1} \times \left[\prod_{i=1}^{N_{\text{rej}}}\left[\left(1 - \frac{f_{c_i'}(\ell_i', z_i')}{g_{c_i'}(\ell_i', z_i')}\right)\frac{1}{1 - \alpha_{c_i'}(\ell_i', z_i')}\right]\right]
$$
$$
\times \left[\frac{f_{c_j'}(\ell_j', z_j')}{g_{c_j'}(\ell_j', z_j')}\frac{1}{\alpha_{c_j'}(\ell_j', z_j')}\right]\Bigg|_{j=N_{\text{rej}}+1},
\tag{28}
$$

where $W_{\text{proposal},i}$ is a weight-factor from the procedure used to sample the $i$-th proposal as per $Q_{\{g\}}^{\text{no-cutoff}}$. Likewise, the weight-factor from the veto algorithm for the case where the scale cutoff is reached is given by

$$
W_{\text{no-emission}} \equiv \underbrace{\left[\prod_{i=1}^{N_{\text{rej}}+1} W_{\text{proposal},i}\right]}_{\equiv 1} \times \left[\prod_{i=1}^{N_{\text{rej}}}\left[\left(1 - \frac{f_{c_i'}(\ell_i', z_i')}{g_{c_i'}(\ell_i', z_i')}\right)\frac{1}{1 - \alpha_{c_i'}(\ell_i', z_i')}\right]\right].
\tag{29}
$$

Typically, and in this paper, the proposal kernels $g_c$-s are chosen to be strictly non-negative and the proposal sampling algorithm is designed so that $W_{\text{proposal},i}$-s are identically 1. Furthermore, the functions $g_c$, $\kappa_{\{g\}}$, $\Delta_{\{g\}}$, $Q_{\{g\}}^{\text{no-cutoff}}$, etc. will henceforth be assumed to be computable straightforwardly, e.g., using closed-form expressions.

**Overall event-weight.** The weight-factor $W_{\text{yes/no-emission}}$ from the veto algorithm constitutes a multiplicative factor to the overall event-weight, which also depends on other parts of the generator pipeline. Concretely, the overall event-weight after a successful parton shower emission step is given by

$$
W_{\text{overall}} = W_{\text{pre-emission}} \times W_{\text{yes-emission}} \times \frac{1}{2\pi P_{\text{emission}}(\varphi)},
\tag{30}
$$

where (a) $W_{\text{pre-emission}}$ only depends on the aspects of the MC-history of the event before the emission is attempted using the veto algorithm, (b) $\varphi \in [0, 2\pi)$ is the emission angle (sampled independent of all else) to perform the emission, and (c) $P_{\text{emission}}(\varphi)$ is the corresponding probability density, chosen to be uniform over $[0, 2\pi)$ typically, and in this work. The fact that the event-weight factorizes into two parts, one which depends only on the pre-emission event and one which depends only on the emission proposals, simplifies the implementation of ARCANE reweighting in Section 3.4.

**Latent generator pathways and their probabilities.** There are many possible sequences of rejected proposals, for the same value of $(\ell, c, z)$ sampled using the veto algorithm. The specific sequence of rejected proposals is latent information that is typically not used during the subsequent processing of the event, except for purposes like reweighting as in Refs. [25–

28]. In this sense, the different possible sequences of rejected proposals constitute different pathways in the event generator that lead to the same "visible" event. The implementation of ARCANE reweighting in this paper involves distributing some reweighting terms over these different pathways, as will be explained in Section 3. The conditional probability density of these pathways is provided next, in order to facilitate such a redistribution.

Let us restrict ourselves to the case where an emission occurs, i.e., the $\ell$ returned by the veto algorithm is greater than $\ell_{\text{cutoff}}$. The sampling (i.e., unweighted) probability distribution of the sequence of rejected proposals, given that $(\ell, c, z)$ is returned by the algorithm, is given by

$$P\left(\left[(\ell'_1, c'_1, z'_1), \dots, (\ell'_{N_{\text{rej}}}, c'_{N_{\text{rej}}}, z'_{N_{\text{rej}}})\right] \; \middle| \; \ell, c, z \; ; \; \ell_{\text{start}}, \ell_{\text{cutoff}}\right) \tag{31}$$
$$\equiv \frac{P\left(\left[(\ell'_1, c'_1, z'_1), \dots, (\ell'_{N_{\text{rej}}}, c'_{N_{\text{rej}}}, z'_{N_{\text{rej}}})\right], (\ell, c, z) \; ; \; \ell_{\text{start}}, \ell_{\text{cutoff}}\right)}{P(\ell, c, z \; ; \; \ell_{\text{start}}, \ell_{\text{cutoff}})}$$

$$= \frac{\overbrace{Q^{\text{no-cutoff}}_{\{g\}}(\ell, c, z \; ; \; \ell'_{N_{\text{rej}}}) \, \alpha_c(\ell, z)}^{\text{sampling the last proposal and accepting it}} \prod_{i=1}^{N_{\text{rej}}} \left[\overbrace{Q^{\text{no-cutoff}}_{\{g\}}(\ell'_i, c'_i, z'_i \; ; \; \ell'_{i-1}) \left(1 - \alpha_{c'_i}(\ell'_i, z'_i)\right)}^{\text{sampling the } i\text{-th proposal and rejecting it}}\right]}{\underbrace{Q^{\text{no-cutoff}}_{\{\alpha g\}}(\ell, c, z \; ; \; \ell_{\text{start}})}_{\text{overall, sampling probability density of } (\ell, c, z) \text{ under the veto algorithm.}^{14}}}, \tag{32}$$

$$= \left[\frac{\Theta(\ell'_{N_{\text{rej}}} - \ell) \, \Delta_{\{g\}}(\ell, \ell'_{N_{\text{rej}}})}{\Theta(\ell_{\text{start}} - \ell) \, \Delta_{\{\alpha g\}}(\ell, \ell_{\text{start}})}\right]$$
$$\times \prod_{i=1}^{N_{\text{rej}}}\left[\Theta(\ell'_{i-1} - \ell'_i) \, \Delta_{\{g\}}(\ell'_i, \ell'_{i-1}) \left(g_{c'_i}(\ell'_i, z'_i) - \alpha_{c'_i}(\ell'_i, z'_i) \, g_{c'_i}(\ell'_i, z'_i)\right)\right]. \tag{33}$$

Here $\ell'_0 \equiv \ell_{\text{start}}$ and $\{\alpha g\}$ denotes the set of functions $\{(\alpha g)_1, \dots, (\alpha g)_C\}$ defined as

$$(\alpha g)_c(\ell, z) \equiv \alpha_c(\ell, z) \, g_c(\ell, z). \tag{34}$$

**Choice of acceptance probability functions.** The choice of proposal kernels $g_c$-s and acceptance probability functions $\alpha_c$-s influence (i) the efficiency of the veto algorithm, in terms of the expected number of rejected proposals and (ii) the distribution of weights from the veto algorithm. The splitting kernel functions $f_c$-s used in this paper are strictly non-negative. For this case, typically, and in this paper, the proposal kernels $g_c$-s are chosen to satisfy $g_c(\ell, z) \geq f_c(\ell, z)$ for all $(\ell, c, z)$, and the acceptance probability functions are chosen as

$$\alpha_c(\ell, z) \equiv \frac{f_c(\ell, z)}{g_c(\ell, z)}. \tag{35}$$

Under these choices, the weights from the veto algorithm, i.e., $W_{\text{yes-emission}}$ and $W_{\text{no-emission}}$, are identically 1; this special case corresponds to the unweighted veto algorithm. Furthermore, the conditional probability in (33) can be written as follows

$$P\left(N_{\text{rej}}, \left[(\ell'_1, c'_1, z'_1), \dots, (\ell'_{N_{\text{rej}}}, c'_{N_{\text{rej}}}, z'_{N_{\text{rej}}})\right] \; \middle| \; \ell, c, z \; ; \; \ell_{\text{start}}, \ell_{\text{cutoff}}\right)$$
$$= \frac{\Theta(\ell'_{N_{\text{rej}}} - \ell) \, \Delta_{\{g\}}(\ell, \ell'_{N_{\text{rej}}})}{\Theta(\ell_{\text{start}} - \ell) \, \Delta_{\{f\}}(\ell, \ell_{\text{start}})} \prod_{i=1}^{N_{\text{rej}}}\left[\Theta(\ell'_{i-1} - \ell'_i) \, \Delta_{\{g\}}(\ell'_i, \ell'_{i-1}) \left(g_{c'_i}(\ell'_i, z'_i) - f_{c'_i}(\ell'_i, z'_i)\right)\right]. \tag{36}$$

---

[14]This follows from the fact that setting $\{f\} \equiv \{\alpha g\}$ in (28) leads to $W_{\text{yes-emission}} \equiv 1$. Note that we are interested in the unweighted probability density here.

## 2.7 Generation Pipeline Until the Chosen Stopping Point

For brevity, henceforth, the qualifier "until the chosen stopping point" will apply to all discussions of the event generation pipeline, unless otherwise specified. Likewise the term "event", unless otherwise specified, will refer to an event generated until the chosen stopping point.

**Generation Pipeline for $\mathbb{H}$-Events.** The generation pipeline for $\mathbb{H}$-events is depicted as a flowchart in Figure 2. This pipeline, which is just one way to generate the kinematics of a resolved event (and choose the emission starting scale), proceeds as follows. First, an unresolved event $e^+e^- \longrightarrow q\bar{q}$ is sampled. Then an emission channel $c_{gen}$ and emission parameters $(y_{gen}, z_{gen}, \varphi_{gen})$ are sampled. The corresponding emission is performed to generate a resolved event. Next, assuming this resolved event was produced by an emission via channel 1, the corresponding emission parameters $(\ell_1, y_1)$ are computed, using (18) and (19). Likewise, assuming the same event was produced by an emission via channel 2, the corresponding emission parameters $(\ell_2, y_2)$ are computed. Next, a channel $c_{for\text{-}scale} \in \{1, 2\}$ is sampled. $\ell_{c_{for\text{-}scale}}$ is set as the start scale $\ell_{start}$ for the next parton shower emission. As an edge case, if either one of $y_1$ and $y_2$ is less than $a_{min}$ (a parameter of the event generator), then the event-weight is set to 0 and the original unresolved event is returned.[15] Note that this procedure involves the sampling of two channel-variables, $c_{gen}$ and $c_{for\text{-}scale}$—the former for performing an emission and **gen**erating a resolved event and the latter **for** setting the starting **scale** for the next emission.

**Generation Pipeline for $\mathbb{S}$-Events.** The generation pipeline for $\mathbb{S}$-events is depicted as a flowchart in Figure 3, and proceeds as follows. First an unresolved event $e^+e^- \longrightarrow q\bar{q}$ is sampled. The value of $\ell_{start}$ is set to $\ln\left(\frac{E_{cms}^2}{1\,\text{GeV}^2}\right)$.[16] A single parton shower emission is attempted using the veto algorithm. The outcome of this process is either an unresolved event (if no emission occurs), or a resolved event (with a specific $\ell_{start}$ for the next emission attempt).

## 2.8 List of Relevant Event-Attributes

The following is the list of all randomly sampled event-attributes, both visible and hidden, for $\mathbb{H}$-events with $\min(y_1, y_2) \geq a_{min}$ (such events belong in classes 1 or 2, as defined in Section 3):

- The event-type attribute, `type`, which equals $\mathbb{H}$.

- $\cos(\theta_{lo,gen})$ and $\varphi_{lo,gen}$ parameterizing the leading-order event kinematics.

- The flavor `fl` of the quark.

- The emission channel $c_{gen}$ and parameters $(y_{gen}, z_{gen}, \varphi_{gen})$.

- The channel $c_{for\text{-}scale}$.

The following is the list of all randomly sampled event-attributes, both visible and hidden, for $\mathbb{S}$-events which undergo a successful first emission (such events belong in classes 4 or 5, as defined in Section 3):

- The event-type attribute, `type`, which equals $\mathbb{S}$.

- $\cos(\theta_{lo})$ and $\varphi_{lo}$ parameterizing the leading-order event kinematics.

---

[15]The event being returned is irrelevant if the event-weight is 0. This choice of returning the original unresolved event is made to conform with Ref. [16].

[16]From (20) it can be seen that no emission will occur with $\ell > \ln(E_{cms}^2/(4\,\text{GeV}^2))$. So, one can reduce $\ell_{start}$ to this threshold value, in principle. But this is not done in the present work, conforming with Ref. [16].

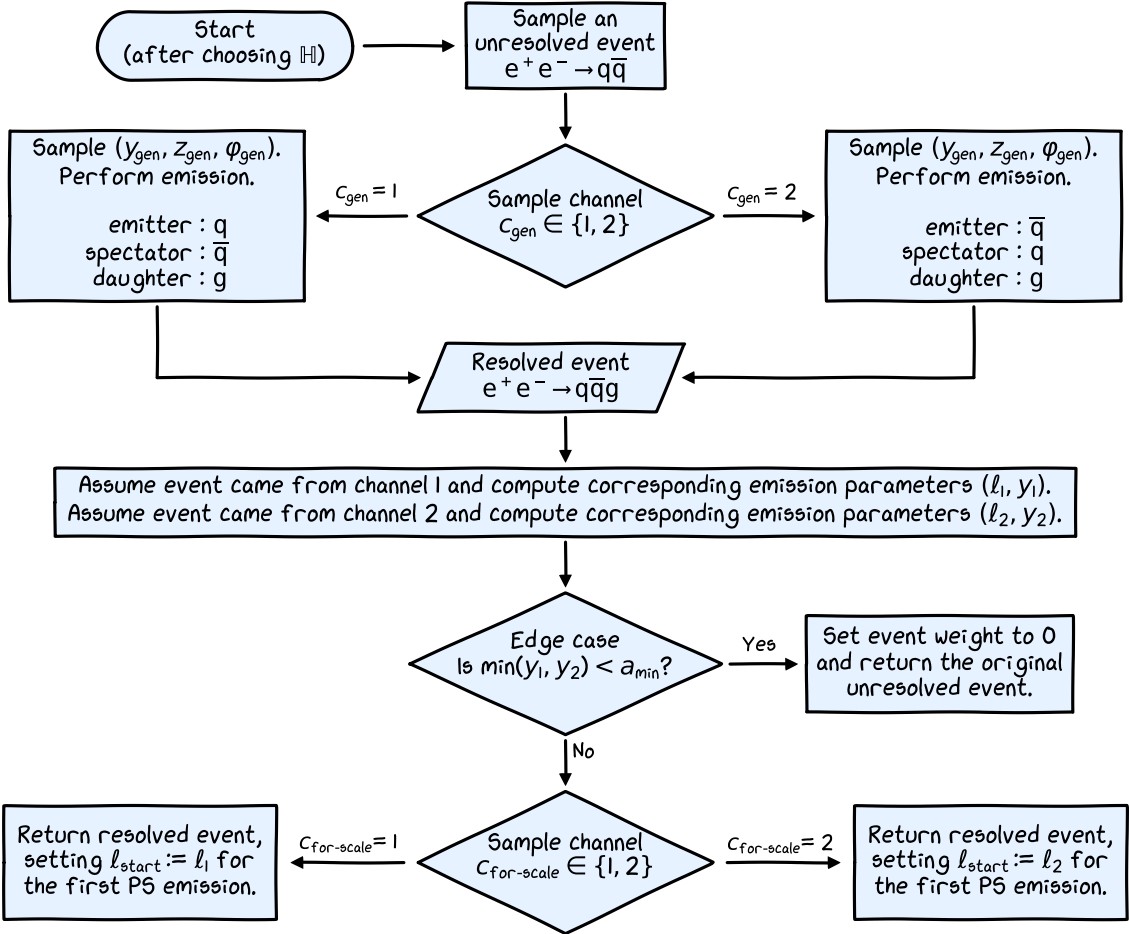

Figure 2: Flowchart depicting the generation of $\mathbb{H}$-events, until the generation of a resolved event and the choosing of $\ell_{\text{start}}$ for the first parton shower (PS) emission.

- The flavor fl of the quark.

- The specific sequence of $N_{\text{rej}} \in \{0, 1, \dots\}$ rejected proposals

$$\texttt{rej-list} \equiv \left[ (\ell'_1, c'_1, z'_1), \dots, (\ell'_{N_{\text{rej}}}, c'_{N_{\text{rej}}}, z'_{N_{\text{rej}}}) \right] \tag{37}$$

- Accepted proposal $(\ell, c, z)$ and the emission parameter $\varphi$.

Note that the leading-order-kinematics angles and emission parameters have a subscript "gen" for $\mathbb{H}$-events. This is to denote that those angles and parameters will lead to the momenta in the final state of the event, if $c_{\text{gen}}$ is used as the emission channel, as opposed to $c_{\text{for-scale}}$ (which may or may not be the same as $c_{\text{gen}}$).

The attributes listed here uniquely determine all the visible and hidden aspects of a resolved event, including its weight. While the color structure of the particles is generally relevant, it is equivalent for all events with the $q\bar{q}g$ final state. So the color structure is treated as a constant and left out here. The probability density function of the attributes listed above can be computed, for any valid choice of values for the attributes, as a product of the conditional probability densities of the individual random decisions and draws. The visible attributes of these events are the momenta in the $q\bar{q}g$ final state, the flavor fl, and the starting scale $\ell_{\text{start}}$ for the following parton shower emission. With this groundwork, we are now in a position to perform ARCANE reweighting.

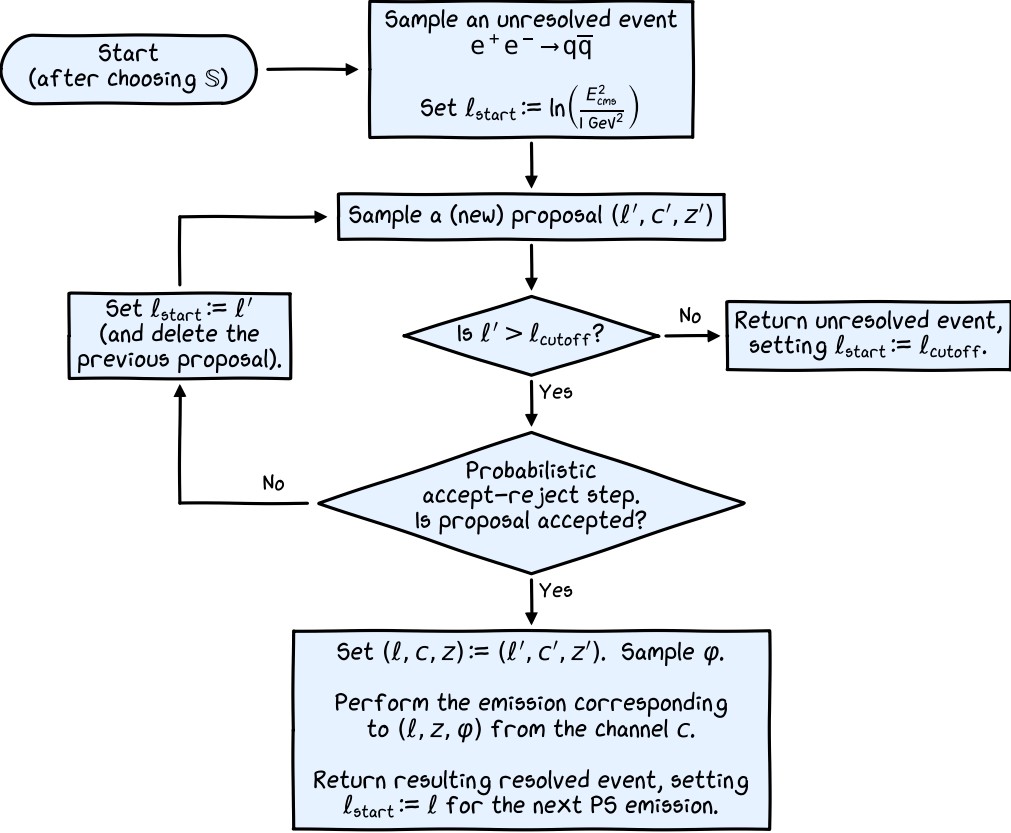

Figure 3: Flowchart depicting the generation of $\mathbb{S}$-events, until performing the first parton shower (PS) emission and choosing $\ell_{\text{start}}$ for the next emission.

## 3 Walkthrough of the ARCANE Reweighting Implementation

A non-technical overview of the implementation of ARCANE reweighting will be provided next, before walking through the technical details of the implementation.

### 3.1 Overview of the Redistribution Strategy

The events produced by the generation pipeline, up to the chosen stopping point, can be partitioned into the following six classes:

Class 1. $\mathbb{H}$-events with $\min(y_1, y_2) > a_{\min}$ and $\ell_{\text{start}} > \ell_{\text{cutoff}}$ for the next emission.

Class 2. $\mathbb{H}$-events with $\min(y_1, y_2) > a_{\min}$ and $\ell_{\text{start}} \leq \ell_{\text{cutoff}}$ for the next emission (this can happen since $\ell_{\text{cutoff}}$ is not used in the generation pipeline for $\mathbb{H}$-type resolved events).

Class 3. $\mathbb{H}$-events that fall under the $\min(y_1, y_2) \leq a_{\min}$ edge case (they all have 0 weight).

Class 4. $\mathbb{S}$-events that undergo a successful first PS emission and satisfy $\min(y_1, y_2) > a_{\min}$ (with $y_1$ and $y_2$ defined similarly as in $\mathbb{H}$-events). These events will satisfy $\ell_{\text{start}} > \ell_{\text{cutoff}}$ for the next emission.

Class 5. $\mathbb{S}$-events that undergo a successful first PS emission and satisfy $\min(y_1, y_2) \leq a_{\min}$ (this can occur, in principle, even for a really small $a_{\min}$, despite the emission scale cutoff, since the cutoff only applies to one of the channels and not the other one).

Class 6. $\mathbb{S}$-event that do not undergo a successful first PS emission.

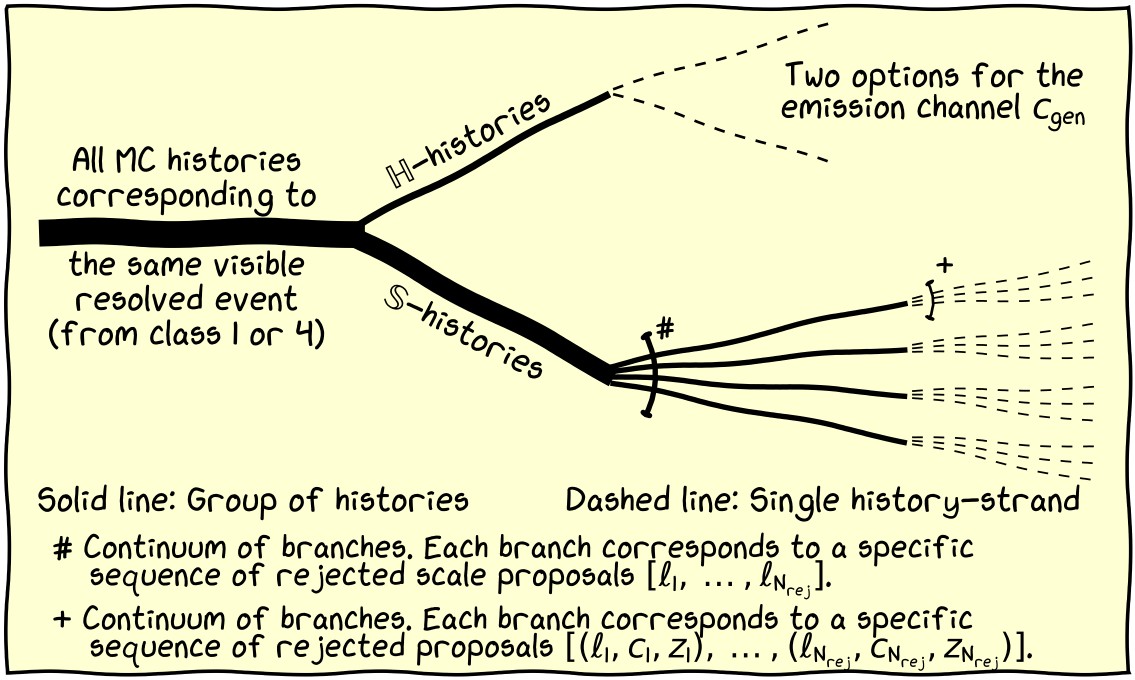

Figure 4: A diagram depicting the different Monte Carlo (MC) histories that lead to the same visible resolved event with a $q\bar{q}g$ final state, satisfying $\min(y_1, y_2) > a_{\min}$ and $\ell_{\text{start}} > \ell_{\text{cutoff}}$ for the next emission. The solid lines of varying thicknesses represent different groups of histories. Going from left-to-right, each split or fork represents a partition (i.e., a non-overlapping and exhaustive grouping) of a set of histories before the split into multiple subsets after the split. The dashed lines represent individual completely-specified histories, each of which has (i) a unique weight associated with it and (ii) a straightforwardly computatable value for the probability density function.

Considering only the visible attributes of the events, $\mathbb{H}$-events of class 2 have no overlap, in distribution, with any of the $\mathbb{S}$-event-classes. Likewise, classes 3, 5, and 6 either (a) only have 0 weights, or (b) have no overlap, in distribution, with any other class containing events with non-zero weights.[17] This leaves classes 1 and 4, which do overlap in distribution with each other, considering only visible attributes. Let us restrict the redistribution of contributions via ARCANE reweighting to these two classes of events (stated differently, the ARCANE additive reweighting term will be set to 0 for events from the remaining classes).

In the rest of this paper, unless otherwise stated, "event" will refer to an event belonging in either class 1 or class 4. Figure 4 depicts the different Monte Carlo histories that lead to the same visible event. The histories can be partitioned as $\mathbb{H}$- and $\mathbb{S}$-histories. There are two individual $\mathbb{H}$-histories, corresponding to $c_{\text{gen}} = 1$ and $c_{\text{gen}} = 2$, respectively. There are a continuum of $\mathbb{S}$-histories, each corresponding to a specific sequence of rejected proposals in the veto algorithm.

The redistribution of contributions of these histories will be done in two steps. First, the total "contribution" of the two $\mathbb{H}$-histories will be computed and redistributed, proportional to their relative frequency of occurrence. This is similar to how events are weighted in the standard or SHERPA-style multichannel importance sampling [29, 30]. After this, the two different $\mathbb{H}$-histories will be treated as "merged" into one. The sampling probability density of

---

[17]Depending on how the events are handled beyond the chosen stopping point, classes 2, 5, and 6 may overlap with each other at a later point in the generation pipeline, e.g., during or after fragmentation.

this merged $\mathbb{H}$-history can be computed as the sum of the probabilities of the two contributing branches (with appropriate Jacobian factors).

Next, an appropriate contribution will be moved from the merged $\mathbb{H}$-history to the $\mathbb{S}$-histories-group. This contribution will be distributed among the various $\mathbb{S}$-histories. This redistribution of contributions will be accomplished via ARCANE reweighting. The rest of this section describes the technical details of performing this redistribution.

### 3.2 Notation and Setup

Redistribution of the kind performed in this paper can be simplified by first choosing a consistent parametrization for the visible event $V$, across all the different histories. A simple choice of parametrization is as follows:

$$V \equiv \Big( \cos(\theta_{\text{lo}}), \varphi_{\text{lo}}, \texttt{fl}, \ell, c, z, \varphi \Big), \tag{38}$$

where the parameters are defined exactly as they are in the $\mathbb{S}$-histories. For an $\mathbb{H}$-event, $V$ is computed as follows:

$$c \equiv c_{\text{for-scale}}, \qquad \cos(\theta_{\text{lo}}) \equiv \cos(\theta_{\text{lo,for-scale}}), \qquad \varphi_{\text{lo}} \equiv \varphi_{\text{lo,for-scale}}, \tag{39}$$

$$\ell \equiv \ell_{\text{for-scale}}, \qquad z \equiv z_{\text{for-scale}}, \qquad \varphi \equiv \varphi_{\text{for-scale}}. \tag{40}$$

where the parameters $\cos(\theta_{\text{lo,for-scale}})$, $\varphi_{\text{lo,for-scale}}$, $\ell_{\text{for-scale}}$, $z_{\text{for-scale}}$, and $\varphi_{\text{for-scale}}$ are computed, as per the event kinematics described in Section 2.3 and Section 2.4, assuming that the emission channel $c_{\text{for-scale}}$ led to the final state momenta in the event. This parametrization of visible events determines the final state momenta as well as $\ell_{\text{start}}$ for the next emission, without any redundancies. The full event information, including the visible and hidden attributes, for $\mathbb{H}$-events can be parameterized as

$$\Big( V, \texttt{type} = \mathbb{H}, c_{\text{gen}} \Big), \quad \text{which will be written simply as} \quad \Big( V, \mathbb{H}, c_{\text{gen}} \Big). \tag{41}$$

Likewise the full event information for $\mathbb{S}$-events can be parameterized as

$$\Big( V, \texttt{type} = \mathbb{S}, \texttt{rej-list} \Big), \quad \text{which will be written simply as} \quad \Big( V, \mathbb{S}, \texttt{rej-list} \Big). \tag{42}$$

Let $P\big(V, \mathbb{H}, c_{\text{gen}}\big)$ and $P\big(V, \mathbb{S}, \texttt{rej-list}\big)$ be the sampling (i.e., unweighted) probability densities of the full event information, defined in terms of appropriate reference measures, for $\mathbb{H}$- and $\mathbb{S}$-events, respectively. These functions can be extracted from the internals of the base event generator. Note that these probability density functions are not individually unit-normalized. Rather, they are normalized as

$$\int_{\Omega_V^{\text{cls-1-or-4}}} \mathrm{d}V \sum_{c_{\text{gen}}=1}^{2} P(V, \mathbb{H}, c_{\text{gen}}) = \text{Prob. of a generic, random event belonging to class 1}, \tag{43}$$

$$\int_{\Omega_V^{\text{cls-1-or-4}}} \mathrm{d}V \int_{\Omega_{\texttt{rej-list}}} \mathrm{d}(\texttt{rej-list}) \; P(V, \mathbb{S}, \texttt{rej-list}) = \text{Same, but for class 4}, \tag{44}$$

where the integrations with respect to $V$ and $\texttt{rej-list}$ use appropriate reference measures and integral domains.

Due to the change in the parametrization of $\mathbb{H}$-events, one needs to incorporate appropriate Jacobian factors to compute the probability density under the adopted parametrization correctly. From (10) and (22), it can be seen that

$$\frac{\mathrm{d}^3 \vec{p}_{\text{em}}}{(2\pi)^3 \, 2 E_{\text{em}}} \frac{\mathrm{d}^3 \vec{p}_{\text{g}}}{(2\pi)^3 \, 2 E_{\text{g}}} \frac{\mathrm{d}^3 \vec{p}_{\text{sp}}}{(2\pi)^3 \, 2 E_{\text{sp}}} \delta^{(4)}\big(p_{\text{em}} + p_{\text{g}} + p_{\text{sp}} - p_{\text{tot}}\big)$$

$$= \left[ \frac{\mathrm{d}\cos(\theta_{\text{lo}}) \; \mathrm{d}\varphi_{\text{lo}}}{8 \, (2\pi)^6} \right] \left[ \frac{2\big(\tilde{p}_{\text{em}} \, \tilde{p}_{\text{sp}}\big)(1-y)}{16 \, \pi^2} \, \mathrm{d}y \, \mathrm{d}z \, \frac{\mathrm{d}\varphi}{2\pi} \right], \tag{45}$$

where em and sp are q and $\bar{\text{q}}$, respectively, for channel 1, and vice versa for channel 2. Here, the momenta in the left-hand-side correspond to the final state particles, while $\tilde{p}_{\text{em}}$ and $\tilde{p}_{\text{sp}}$ are momenta in the intermediate unresolved state. Note that the left-hand-side is invariant under swapping q and $\bar{\text{q}}$, so the relevant Jacobian factor for switching channels can be extracted from this expression. Since $\tilde{p}_{\text{em}}\tilde{p}_{\text{sp}}$ is the same for both channels, only the $(1-y)$ factor in (45) is relevant for switching channels, for the process under consideration in this paper. Concretely,

$$
\begin{aligned}
P\big(V, \mathbb{H}, c_{\text{gen}}\big) = P\big(\mathbb{H}, \cos(\theta_{\text{lo,gen}}), \varphi_{\text{lo,gen}}, \texttt{fl}, c_{\text{gen}}, y_{\text{gen}}, z_{\text{gen}}, \varphi_{\text{gen}}, c_{\text{for-scale}}\big) \\
\times \underbrace{\frac{1 - y_{\text{for-scale}}}{1 - y_{\text{gen}}}}_{\substack{\text{Jacobian factor for switching channels} \\ \text{in the } (\cos(\theta_{\text{lo}}), \varphi_{\text{lo}}, y, z, \varphi)\text{-parametrization}}} \times \underbrace{y_{\text{for-scale}}}_{\substack{\text{Jacobian factor for} \\ \text{switching from } y_{\text{for-scale}} \\ \text{to } \ell_{\text{for-scale}}, \text{ using (21)}}} ,
\end{aligned}
\tag{46}
$$

where $P\big(\mathbb{H}, \cos(\theta_{\text{lo,gen}}), \varphi_{\text{lo,gen}}, \texttt{fl}, c_{\text{gen}}, y_{\text{gen}}, z_{\text{gen}}, \varphi_{\text{gen}}, c_{\text{for-scale}}\big)$ can be computed simply as a product of the relevant conditional sampling probability functions. Making the parametrization of $V$ consistent across histories and accounting for the Jacobian factors upfront has simplified the pedagogy here, since the need for tracking Jacobian factors has been eliminated for the subsequent steps.

Let $W^{(\text{ORIG})}\big(V, \mathbb{H}, c_{\text{gen}}\big)$ and $W^{(\text{ORIG})}\big(V, \mathbb{S}, \texttt{rej-list}\big)$ be the weights of $\mathbb{H}$- and $\mathbb{S}$-events, respectively, under the base (i.e., original) event generator. These functions can also be extracted from the internals of the base generator. Note that event-weights are invariant under changes in event-parametrization. Recall that the quasi density of the full event information, under the weighting scheme denoted by "(superscript)" be given by

$$
F^{(\text{superscript})}(V, L) \equiv P(V, L) \times W^{(\text{superscript})}(V, L),
\tag{47}
$$

where $L$ represents all the relevant latent attributes of a given event ("superscript" is "ORIG" for the base event generator). By integrating over the latent information, the marginal quasi density of a visible event $V$ (belonging to class 1 or 4) under the weighting scheme "(superscript)" can be written as

$$
\begin{aligned}
F^{(\text{superscript})}(V) \equiv & \left[ \sum_{c_{\text{gen}}=1}^{2} F^{(\text{superscript})}(V, \mathbb{H}, c_{\text{gen}}) \right] \\
& + \left[ \int_{\Omega_{\texttt{rej-list}}} \text{d}(\texttt{rej-list}) \quad F^{(\text{superscript})}(V, \mathbb{S}, \texttt{rej-list}) \right].
\end{aligned}
\tag{48}
$$

As a reminder, this marginal quasi density will be preserved under ARCANE reweighting.

Let $P(\mathbb{H})$ and $P(\mathbb{S}) \equiv 1 - P(\mathbb{H})$ be the probabilities of choosing the event-type to be $\mathbb{H}$ and $\mathbb{S}$, respectively, in the MC@NLO pipeline, with $0 < P(\mathbb{H}), P(\mathbb{S}) < 1$. The following auxiliary weight functions and conditional probability functions are introduced for simplifying some subsequent discussions:

$$
P_{\mathbb{H}}(V, c_{\text{gen}}) \equiv \frac{P(V, \mathbb{H}, c_{\text{gen}})}{P(\mathbb{H})},
\tag{49}
$$

$$
P_{\mathbb{S}}(V, \texttt{rej-list}) \equiv \frac{P(V, \mathbb{S}, \texttt{rej-list})}{P(\mathbb{S})},
\tag{50}
$$

$$
W_{\mathbb{H}}^{(\text{superscript})}(V, c_{\text{gen}}) \equiv P(\mathbb{H}) \times W^{(\text{superscript})}(V, \mathbb{H}, c_{\text{gen}}),
\tag{51}
$$

$$
W_{\mathbb{S}}^{(\text{superscript})}(V, \texttt{rej-list}) \equiv P(\mathbb{S}) \times W^{(\text{superscript})}(V, \mathbb{S}, \texttt{rej-list}).
\tag{52}
$$

If one keeps the functions $P_{\mathbb{H}}$, $P_{\mathbb{S}}$, $W_{\mathbb{H}}^{(\text{superscript})}$, and $W_{\mathbb{S}}^{(\text{superscript})}$ fixed in an event-generator, and changes $P(\mathbb{H})$ and $P(\mathbb{S})$, the quasi density of events will remain unchanged.

## 3.3  Step 1: Merging the two $\mathbb{H}$-Histories

This step involves redistributing the total $\mathbb{H}$-contribution to the quasi density of a given visible event across the two $\mathbb{H}$-channels, so that the event-weight is independent of $c_{\text{gen}}$. After such a redistribution, it is convenient to treat the two $\mathbb{H}$-history-branches as merged, and just drop $c_{\text{gen}}$ from the attributes list. Concretely, the event-weight for the merged $\mathbb{H}$-branch is given by

$$W^{(\mathbb{H}\text{-merged})}(V,\mathbb{H}) \equiv \frac{\displaystyle\sum_{c'_{\text{gen}}=1}^{2} F^{(\text{ORIG})}\Big(V,\mathbb{H},c'_{\text{gen}}\Big)}{\displaystyle\sum_{c'_{\text{gen}}=1}^{2} P\Big(V,\mathbb{H},c'_{\text{gen}}\Big)} = \frac{\displaystyle\sum_{c'_{\text{gen}}=1}^{2} P(V,\mathbb{H},c'_{\text{gen}}) \times W^{(\text{ORIG})}\Big(V,\mathbb{H},c'_{\text{gen}}\Big)}{\displaystyle\sum_{c'_{\text{gen}}=1}^{2} P\Big(V,\mathbb{H},c'_{\text{gen}}\Big)} \, . \tag{53}$$

The auxiliary weight function $W_{\mathbb{H}}^{(\mathbb{H}\text{-merged})}$ is given by

$$W_{\mathbb{H}}^{(\mathbb{H}\text{-merged})}(V) \equiv P(\mathbb{H}) \, \times \, W^{(\mathbb{H}\text{-merged})}(V,\mathbb{H}) = \frac{\displaystyle\sum_{c'_{\text{gen}}=1}^{2} P_{\mathbb{H}}(V,c'_{\text{gen}}) \, \times \, W_{\mathbb{H}}^{(\text{ORIG})}\Big(V,c'_{\text{gen}}\Big)}{\displaystyle\sum_{c'_{\text{gen}}=1}^{2} P_{\mathbb{H}}\Big(V,c'_{\text{gen}}\Big)} \, , \tag{54}$$

and is independent of $P(\mathbb{H})$, assuming the functions $P_{\mathbb{H}}$ and $W_{\mathbb{H}}^{(\text{ORIG})}$ are kept fixed. This is understandable, since the reweighting is happening entirely among the $\mathbb{H}$-histories. For completeness, the sampling probability density and its conditional version for the merged $\mathbb{H}$-history are given by

$$P(V,\mathbb{H}) = \sum_{c'_{\text{gen}}=1}^{2} P\Big(V,\mathbb{H},c'_{\text{gen}}\Big), \qquad P_{\mathbb{H}}(V) \equiv \frac{P(V,\mathbb{H})}{P(\mathbb{H})} = \sum_{c'_{\text{gen}}=1}^{2} P_{\mathbb{H}}\Big(V,c'_{\text{gen}}\Big). \tag{55}$$

The weights and probabilities of the merged $\mathbb{H}$-history can be computed from the internals of the base generator, by backtracking the event-kinematics corresponding to the different values of $c'_{\text{gen}}$ in the summations above.

It is important to point out that this form of redistribution is not a novel contribution of this paper, although it does fit within the framework of ARCANE redistribution.[18] As mentioned before, standard or SHERPA-style multichannel importance sampling also weights events in a similar manner. Recall that the base event generator used here is taken from a tutorial. Production-quality event generator programs typically either (a) already perform such distributions as needed or (b) sample resolved $\mathbb{H}$-events via a different parametrization of the event-kinematics without needing multiple channels in the first place.

Let the event generator that incorporates this $\mathbb{H}$-branches-merging provide the new baseline ("NBL") weighting scheme. For notational convenience, the function $W^{(\text{NBL})}$ is defined as follows, noting that $\mathbb{S}$-events are unaffected by the reweighting procedure in this step:

$$W^{(\text{NBL})}(V,\mathbb{H}) \equiv W^{(\mathbb{H}\text{-merged})}(V,\mathbb{H}), \tag{56}$$

$$W^{(\text{NBL})}\big(V,\mathbb{S},\texttt{rej-list}\big) \equiv W^{(\text{ORIG})}\big(V,\mathbb{S},\texttt{rej-list}\big), \tag{57}$$

with $F^{(\text{NBL})}$, $W_{\mathbb{H}}^{(\text{NBL})}$, and $W_{\mathbb{S}}^{(\text{NBL})}$ defined in terms of $W^{(\text{NBL})}$, as per (47), (51), and (52). This way the next redistribution step can be described in terms of only the "NBL" weights and distributions. As an aside, the $\mathbb{H}$-merging step can also be performed for events of class 2, even though they are not focused on in this paper.

---

[18]$W^{(\mathbb{H}\text{-merged})} - W^{(\text{ORIG})}$ can be thought of as an additive reweighting factor.

### 3.4 Step 2: Redistribution Between $\mathbb{H}$- and $\mathbb{S}$-Histories

Similar to step 1, one can *attempt* to redistribute contributions between the $\mathbb{H}$- and $\mathbb{S}$-histories by replacing the weight of each event by the average weight over all possible events with the same $V$. Let $W^*(V)$ represent this ideal weight function, given by

$$W^*(V) \equiv \frac{F^{(\mathrm{NBL})}(V, \mathbb{H}) + F^{(\mathrm{NBL})}(V, \mathbb{S})}{P(V, \mathbb{H}) + P(V, \mathbb{S})}, \tag{58}$$

where $P(V, \mathbb{S})$ and $F^{(\mathrm{NBL})}(V, \mathbb{S})$ are the marginal unweighted and weighted distributions of $(V, \mathbb{S})$, respectively, given by

$$P(V, \mathbb{S}) \equiv \int_{\Omega_{\mathtt{rej-list}}} \mathrm{d}(\mathtt{rej-list}) \quad P(V, \mathbb{S}, \mathtt{rej-list}), \tag{59}$$

$$F^{(\mathrm{NBL})}(V, \mathbb{S}) \equiv \int_{\Omega_{\mathtt{rej-list}}} \mathrm{d}(\mathtt{rej-list}) \quad F^{(\mathrm{NBL})}(V, \mathbb{S}, \mathtt{rej-list}) \tag{60}$$

$$= P(V, \mathbb{S}) \times \mathbb{E}\Big[ W^{(\mathrm{NBL})}(V, \mathbb{S}, \mathtt{rej-list}) \,\Big|\, V, \mathbb{S} \Big]. \tag{61}$$

The problem is, $F^{(\mathrm{NBL})}(V, \mathbb{S})$ is not directly computable for typical choices of kernel splitting functions $\{f\}$ used in the veto algorithm, including the ones used in this paper. Likewise, $P(V, \mathbb{S})$ is also not be directly computable for the unweighted veto algorithm used in this paper.[19] Consequently, the "ARCANE" weight function $W^{(\mathrm{ARCANE})}$ computed in this paper will only approximate $W^*$ and not exactly match it. Furthermore, unlike $W^*$, the ARCANE weight $W^{(\mathrm{ARCANE})}$ will depend on the hidden attributes `type` and `rej-list`.

ARCANE reweighting can be performed by first defining the quasi density $F^{(\mathrm{ARCANE})}(V, L)$. The strategy of moving a contribution from the merged $\mathbb{H}$-branch and redistributing it over the $\mathbb{S}$-branches can be realized by modeling $F^{(\mathrm{ARCANE})}$ as

$$F^{(\mathrm{ARCANE})}(V, \mathbb{H}) \equiv F^{(\mathrm{NBL})}(V, \mathbb{H}) \qquad\qquad - \Lambda(V), \tag{62a}$$

$$F^{(\mathrm{ARCANE})}(V, \mathbb{S}, \mathtt{rej-list}) \equiv F^{(\mathrm{NBL})}(V, \mathbb{S}, \mathtt{rej-list}) + \Lambda(V)\, \Phi_{\mathrm{un}}(\mathtt{rej-list}\,;\, V). \tag{62b}$$

Here $\Lambda$ can be any function of $V$ and $\Phi_{\mathrm{un}}(\cdot\,;\, V)$ can be any unit-normalized (not necessarily non-negative) function of `rej-list`, both subject to certain conditions to ensure proper coverage and finiteness of weights, which will be discussed later. For completeness, the ARCANE redistribution function $G$, which is used is the presentation of the technique in Ref. [12] and Section 1, corresponding to this redistribution is given by

$$G(V, \mathbb{H}) \equiv -\Lambda(V), \qquad\qquad G(V, \mathbb{S}, \mathtt{rej-list}) \equiv +\Lambda(V)\, \Phi_{\mathrm{un}}(\mathtt{rej-list}\,;\, V). \tag{63}$$

It is easy to see that

$$F^{(\mathrm{ARCANE})}(V) \equiv F^{(\mathrm{NBL})}(V) \equiv F^{(\mathrm{ORIG})}(V). \tag{64}$$

Having ensured this condition with the specific form of $F^{(\mathrm{ARCANE})}(V, L)$ in (62), the next task is to either learn (from generated data) or engineer (using the internals of the generator) good choices for the functions $\Lambda$ and $\Phi_{\mathrm{un}}$; this paper takes the engineering approach.

---

[19]$P(V, \mathbb{S})$ could be directly computable if the acceptance probability functions $\alpha_c(\ell, z)$ are chosen conveniently, but that would lead to the weighted veto algorithm.

**Engineering the redistribution function.**   The strategy is to first write down expressions for the ideal functions $\Lambda^*$ and $\Phi^*_{\text{un}}$ for which the corresponding $W^{(\text{ARCANE})}(V,L)$ will match $W^*(V)$. Next, these expressions will be inspected to see which parts of them can be directly computed using the internals of the generator and which ones cannot. The parts which cannot be directly computed exactly will then be replaced with approximations, taking care to make sure that the resulting $\Phi_{\text{un}}$ is unit-normalized and that the conditions to ensure proper coverage are satisfied. The better the quality of the approximations, the closer $W^{(\text{ARCANE})}(V,L)$ will be to $W^*(V)$. But regardless of the quality of the approximations, no bias is introduced.

For simplicity, this paper only performs the redistribution for the situation where the unweighted veto algorithm is used. However, the procedure here can be extended to the weighted veto algorithm case as well; this will be briefly discussed in Appendix A.2. Using (58) and by setting

$$W^{*,(\text{ARCANE})}(V,\mathbb{H}) \equiv \frac{F^{*,(\text{ARCANE})}(V,\mathbb{H})}{P(V,\mathbb{H})} \equiv \frac{F^{(\text{NBL})}(V,\mathbb{H}) - \Lambda^*(V)}{P(V,\mathbb{H})} = W^*(V), \tag{65}$$

it can be shown that

$$\Lambda^*(V) \equiv \frac{F^{(\text{NBL})}(V,\mathbb{H})\, P(V,\mathbb{S}) - F^{(\text{NBL})}(V,\mathbb{S})\, P(V,\mathbb{H})}{P(V,\mathbb{H}) + P(V,\mathbb{S})}. \tag{66}$$

All the functions on the right-hand-side, except $P(V,\mathbb{S})$ and $F^{(\text{NBL})}(V,\mathbb{S})$, can be directly computed. From the description of the generator pipeline for $\mathbb{S}$-events in Section 2, the structure of the unweighted veto algorithm, and the fact that $\varphi$ for the emission is sampled independent of the veto algorithm, it can be seen that

$$P(V,\mathbb{S}) = P(\mathbb{S}, V_{\text{lo}}, \varphi)\, Q^{\text{no-cutoff}}_{\{f\}}(\ell, c, z\,;\ell_{\text{start}}) \equiv P(\mathbb{S})\, P_{\mathbb{S}}(V_{\text{lo}}, \varphi)\, Q^{\text{no-cutoff}}_{\{f\}}(\ell, c, z\,;\ell_{\text{start}}), \tag{67}$$

where $V_{\text{lo}} \equiv (\cos(\theta_{\text{lo}}), \varphi_{\text{lo}}, \texttt{fl})$. Here $Q^{\text{no-cutoff}}_{\{f\}}$ is equivalent to $Q_{\{f\}}$, since we are restricting ourselves to events from classes 1 and 4, which satisfy $\ell > \ell_{\text{cutoff}}$. Furthermore, from (61) and (30), which states that $W^{(\text{NBL})}_{\mathbb{S}}(V, \texttt{rej-list})$ only depends on $(V_{\text{lo}}, \varphi)$ when using the unweighted veto algorithm, we can write

$$F^{(\text{NBL})}(V,\mathbb{S}) = W^{(\text{NBL})}_{\mathbb{S}}(V_{\text{lo}}, \varphi)\, P_{\mathbb{S}}(V_{\text{lo}}, \varphi)\, Q^{\text{no-cutoff}}_{\{f\}}(\ell, c, z\,;\ell_{\text{start}}) \tag{68}$$

$$\equiv F^{(\text{NBL})}(\mathbb{S}, V_{\text{lo}}, \varphi)\, Q^{\text{no-cutoff}}_{\{f\}}(\ell, c, z\,;\ell_{\text{start}}), \tag{69}$$

with $W^{(\text{NBL})}_{\mathbb{S}}(V_{\text{lo}}, \varphi)$ and $F^{(\text{NBL})}(\mathbb{S}, V_{\text{lo}}, \varphi)$, appropriately defined.[20] Plugging (23h), (67) and (69) into (66), and using the fact that $\ell > \ell_{\text{start}}$ for events of class 1 and 4, we have

$$\Lambda^*(V) \equiv \left[ \frac{F^{(\text{NBL})}(V,\mathbb{H})\, P(\mathbb{S}, V_{\text{lo}}, \varphi) - F^{(\text{NBL})}(\mathbb{S}, V_{\text{lo}}, \varphi)\, P(V,\mathbb{H})}{P(V,\mathbb{H}) + \left[ P(\mathbb{S}, V_{\text{lo}}, \varphi)\, \Theta(\ell_{\text{start}} - \ell)\, \Delta_{\{f\}}(\ell, \ell_{\text{start}})\, f_c(\ell, z) \right]} \right] \tag{70}$$

$$\times \left[ \Theta(\ell_{\text{start}} - \ell)\, \Delta_{\{f\}}(\ell, \ell_{\text{start}})\, f_c(\ell, z) \right].$$

This can be rewritten as

$$\Lambda^*(V) \equiv \left[ \frac{W^{(\text{NBL})}_{\mathbb{H}}(V)\, P_{\mathbb{H}}(V)\quad P(\mathbb{S})\, P_{\mathbb{S}}(V) - W^{(\text{NBL})}_{\mathbb{S}}(V_{\text{lo}}, \varphi)\, P_{\mathbb{S}}(V)\quad P(\mathbb{H})\, P_{\mathbb{H}}(V)}{P(\mathbb{H})\, P_{\mathbb{H}}(V) + P(\mathbb{S})\, P_{\mathbb{S}}(V)} \right], \tag{71}$$

where $P_{\mathbb{S}}(V)$ is given by

$$P_{\mathbb{S}}(V) \equiv P_{\mathbb{S}}(V_{\text{lo}}, \varphi)\, \Theta(\ell_{\text{start}} - \ell)\, \Delta_{\{f\}}(\ell, \ell_{\text{start}})\, f_c(\ell, z). \tag{72}$$

---

[20]In the present example, $F(\mathbb{S}, V_{\text{lo}}, \varphi)$, $P_{\mathbb{S}}(V_{\text{lo}}, \varphi)$, and $W^{(\text{NBL})}_{\mathbb{S}}(V_{\text{lo}}, \varphi)$ are all independent of $\varphi$.

$P_{\mathbb{S}}(V)$ is the probability density of the visible event $V$, given that the event was chosen to be of the $\mathbb{S}$-type.

For the unweighted veto algorithm case, the optimal way to redistribute this $\Lambda^*(V)$, across the different $\mathbb{S}$-histories, is simply proportional to the sampling probabilities of the individual $\mathbb{S}$-histories. So, from (36), we have

$$
\begin{aligned}
&\Phi_{\mathrm{un}}^*(\texttt{rej-list}\,;V)\\
&= \frac{\Theta(\ell'_{N_{\mathrm{rej}}}-\ell)\,\Delta_{\{g\}}(\ell,\ell'_{N_{\mathrm{rej}}})}{\Theta(\ell_{\mathrm{start}}-\ell)\,\Delta_{\{f\}}(\ell,\ell_{\mathrm{start}})}\ \prod_{i=1}^{N_{\mathrm{rej}}}\Big[\Theta(\ell'_{i-1}-\ell'_i)\,\Delta_{\{g\}}(\ell'_i,\ell'_{i-1})\,\big(g_{c'_i}(\ell'_i,z'_i)-f_{c'_i}(\ell'_i,z'_i)\big)\Big],
\end{aligned}
\tag{73}
$$

All functions on the right-hand-sides of (70) and (73) are directly computable, except $\Delta_{\{f\}}$, which will have to be approximated.

The inability to compute $\Delta_{\{f\}}$ easily seems to be at the heart of the negative weights problem for the specific example in this paper, as well as for MC@NLO event generation, in general. This is not surprising and can be understood as follows. If $\Delta_{\{f\}}(\ell,\ell_{\mathrm{start}})$ was easy to compute, then one can sample the parton shower emission parameters directly without using the veto algorithm.[21] In this case, there would only be a single $\mathbb{S}$-history (since there are no rejected proposals), and moving contributions across the $\mathbb{H}$- and $\mathbb{S}$-branches could be done in a more straightforward manner using multichannel (re)weighting, similar to how the two $\mathbb{H}$-branches were merged in Section 3.3.

In $\Phi_{\mathrm{un}}^*$, if we simply replace $\Delta_{\{f\}}$ with an approximation, say $\widehat{\Delta}_{\{f\}}$, then the resulting $\Phi$-function is not guaranteed to be unit-normalized. On the other hand, replacing the functions $\{f\}$ in (73), with approximations for them, say $\{h\}$, will lead to a unit-normalized $\Phi_{\mathrm{un}}$.[22] This leads to the following choice of $\Phi_{\mathrm{un}}$, implicitly parameterized by the functions $\{h\}$:

$$
\begin{aligned}
&\Phi_{\mathrm{un}}(\texttt{rej-list}\,;V)\\
&= \frac{\Theta(\ell'_{N_{\mathrm{rej}}}-\ell)\,\Delta_{\{g\}}(\ell,\ell'_{N_{\mathrm{rej}}})}{\Theta(\ell_{\mathrm{start}}-\ell)\,\Delta_{\{h\}}(\ell,\ell_{\mathrm{start}})}\ \prod_{i=1}^{N_{\mathrm{rej}}}\Big[\Theta(\ell'_{i-1}-\ell'_i)\,\Delta_{\{g\}}(\ell'_i,\ell'_{i-1})\,\big(g_{c_i}(\ell'_i,z'_i)-h_{c'_i}(\ell'_i,z'_i)\big)\Big],
\end{aligned}
\tag{74}
$$

The functions $\{h\}\equiv\{h_1,\dots,h_C\}$ should be chosen so that $\Delta_{\{h\}}$ can be computed easily. These functions wil be referred to as "$z$-dependent **redistribution** kernel functions" or redistribution kernels for short. Unlike with $\Phi_{\mathrm{un}}$, there are no major restrictions on how $\Delta_{\{f\}}$ should be approximated in the expression for $\Lambda^*(V)$ in (70). Let $\Lambda(V)$ be chosen as[23]

$$
\Lambda(V) \equiv \left[\frac{F^{(\mathrm{NBL})}(V,\mathbb{H})\,P(\mathbb{S},V_{\mathrm{lo}},\varphi)-F^{(\mathrm{NBL})}(\mathbb{S},V_{\mathrm{lo}},\varphi)\,P(V,\mathbb{H})}{P(V,\mathbb{H})+\Big[P(\mathbb{S},V_{\mathrm{lo}},\varphi)\,\Theta(\ell_{\mathrm{start}}-\ell)\,\Delta_{\{h\}}(\ell,\ell_{\mathrm{start}})\,f_c(\ell,z)\Big]}\right]
\tag{75}
$$
$$
\times\ \Big[\Theta(\ell_{\mathrm{start}}-\ell)\,\Delta_{\{h\}}(\ell,\ell_{\mathrm{start}})\,f_c(\ell,z)\Big],
$$

which can be re-written as

$$
\Lambda(V) \equiv \left[\frac{W_{\mathbb{H}}^{(\mathrm{NBL})}(V)\,P_{\mathbb{H}}(V)\quad P(\mathbb{S})\,\widehat{P}_{\mathbb{S}}^{\{h\}}(V)-W_{\mathbb{S}}^{(\mathrm{NBL})}(V_{\mathrm{lo}},\varphi)\,\widehat{P}_{\mathbb{S}}^{\{h\}}(V)\quad P(\mathbb{H})\,P_{\mathbb{H}}(V)}{P(\mathbb{H})\,P_{\mathbb{H}}(V)+P(\mathbb{S})\,\widehat{P}_{\mathbb{S}}^{\{h\}}(V)}\right],
\tag{76}
$$

where

$$
\boxed{\widehat{P}_{\mathbb{S}}^{\{h\}}(V)\equiv P_{\mathbb{S}}(V_{\mathrm{lo}},\varphi)\,\Theta(\ell_{\mathrm{start}}-\ell)\,\Delta_{\{h\}}(\ell,\ell_{\mathrm{start}})\,f_c(\ell,z).}
\tag{77}
$$

---

[21]The inverse function $\Delta_{\{f\}}^{-1}(\ \cdot\ ,\ell_{\mathrm{start}})$ can be computed easily, e.g., using a binary search, if the monotonic function $\Delta_{\{f\}}(\ \cdot\ ,\ell_{\mathrm{start}})$ is easy to compute.

[22]This is true even if the functions $h_c$ are not guaranteed to be non-negative.

[23]Here, only the $\Delta_{\{f\}}$-s have been replaced by $\Delta_{\{h\}}$, and the direct appearances of $f_c(\ell,z)$ have been left intact. While this choice is not necessary for performing ARCANE, it offers some conveniences; see footnotes 26 and 28.

Note that

$$\widehat{P}_{\mathbb{S}}^{\{h\}}(V) = P_{\mathbb{S}}(V)\,\frac{\Delta_{\{h\}}(\ell, \ell_{\text{start}})}{\Delta_{\{f\}}(\ell, \ell_{\text{start}})}, \tag{78}$$

If $\{h\}$ exactly matches $\{f\}$, then $\widehat{P}_{\mathbb{S}}^{\{h\}}(V)$ would match $P_{\mathbb{S}}(V)$. With these choices of $\Lambda$ and $\Psi_{\text{un}}$, the ARCANE weight for $\mathbb{H}$-events of class 1, namely

$$W^{(\text{ARCANE})}(V, \mathbb{H}) \equiv \frac{F^{(\text{NBL})}(V, \mathbb{H}) - \Lambda(V)}{P(V, \mathbb{H})}, \tag{79}$$

is given by[24]

$$\boxed{W^{(\text{ARCANE})}(V, \mathbb{H}) = \frac{W_{\mathbb{H}}^{(\text{NBL})}(V)\,P_{\mathbb{H}}(V) + W_{\mathbb{S}}^{(\text{NBL})}(V_{\text{lo}}, \varphi)\,\widehat{P}_{\mathbb{S}}^{\{h\}}(V)}{P(\mathbb{H})\,P_{\mathbb{H}}(V) + P(\mathbb{S})\,\widehat{P}_{\mathbb{S}}^{\{h\}}(V)}.} \tag{80}$$

Similarly, the ARCANE weight for $\mathbb{S}$-events from class 4 is

$$W^{(\text{ARCANE})}(V, \mathbb{S}, \texttt{rej-list}) \equiv \frac{F^{(\text{NBL})}(V, \mathbb{S}, \texttt{rej-list}) + \Lambda(V)\,\Phi_{\text{un}}(\texttt{rej-list}\,;\,V)}{P(V, \mathbb{S}, \texttt{rej-list})}, \tag{81}$$

where $P(V, \mathbb{S}, \texttt{rej-list})$ is given by

$$\begin{aligned}
P(V, \mathbb{S}, \texttt{rej-list}) = {}& P(\mathbb{S}, V_{\text{lo}}, \varphi)\,\Theta(\ell'_{N_{\text{rej}}} - \ell)\,\Delta_{\{g\}}(\ell, \ell'_{N_{\text{rej}}})\,f_c(\ell, z) \\
& \times \prod_{i=1}^{N_{\text{rej}}}\left[\Theta(\ell'_{i-1} - \ell'_i)\,\Delta_{\{g\}}(\ell'_i, \ell'_{i-1})\left(g_{c'_i}(\ell'_i, z'_i) - f_{c'_i}(\ell'_i, z'_i)\right)\right].
\end{aligned} \tag{82}$$

Plugging (74), (75), and (82) into (81) we get[25]

$$\boxed{\begin{aligned}
W^{(\text{ARCANE})}(V, \mathbb{S}, \texttt{rej-list}) = {}& \frac{W_{\mathbb{S}}^{(\text{NBL})}(V_{\text{lo}}, \varphi)}{P(\mathbb{S})} \\
& + \left[\frac{P_{\mathbb{H}}(V)\left(W_{\mathbb{H}}^{(\text{NBL})}(V) - \dfrac{P(\mathbb{H})}{P(\mathbb{S})}\,W_{\mathbb{S}}^{(\text{NBL})}(V_{\text{lo}}, \varphi)\right)}{P(\mathbb{H})\,P_{\mathbb{H}}(V) + P(\mathbb{S})\,\widehat{P}_{\mathbb{S}}^{\{h\}}(V)}\right]\left[\prod_{i=1}^{N_{\text{rej}}}\left(\frac{g_{c'_i}(\ell'_i, z'_i) - h_{c'_i}(\ell'_i, z'_i)}{g_{c'_i}(\ell'_i, z'_i) - f_{c'_i}(\ell'_i, z'_i)}\right)\right],
\end{aligned}} \tag{83}$$

for resolved $\mathbb{S}$-events satisfying $\ell_{\text{start}} > \ell'_0 > \ell'_1 > \cdots > \ell'_{N_{\text{rej}}} > \ell > \ell_{\text{cutoff}}$. For a resolved $\mathbb{S}$-event that does not satisfy this condition (and so will not be encountered in practice), $W^{(\text{ARCANE})}$ is undefined. Provided one has access to suitable redistribution kernels $\{h\}$, (77), (80), and (83) can be used to compute the ARCANE weights for events from classes 1 and 4. Note that in order to merge the two $\mathbb{H}$-branches and subsequently reweight events using these formulas, one needs access to the hidden event-attributes `type` and `rej-list`, if applicable. Furthermore, one needs to be able to compute quantities like (a) probability of different histories and (b) weights under different histories for a given visible event, regardless of the actual MC-history of the event. This necessitates a strong interdependence between the software implementations of the event-generation and ARCANE reweighting.

---

[24]This formula works for class 2 events as well, for which $\widehat{P}_{\mathbb{S}}^{\{h\}}(V) = 0$ and $W^{(\text{ARCANE})}(V, \mathbb{H}) \equiv W^{(\text{NBL})}(V, \mathbb{H})$.

[25]This formula works for class 5 events as well, for which $P_{\mathbb{H}}(V) = 0$.

**Conditions for proper coverage and finiteness of weights.**   Some caution must be taken to avoid the situation where $F^{(\text{ARCANE})}(V, L)$ is non-zero for MC-histories with zero probability densities, which would amount to the event-generator pipeline not covering the support of $F^{(\text{ARCANE})}(V, L)$. If we assume that the original event-generator has proper coverage over $F^{(\text{NBL})}$, then it suffices to ensure that the ARCANE redistribution function $G(V, L)$ in (63) is 0 for impossible MC-histories $(V, L)$. Note that the value of $G(V, L)$ is irrelevant if $P(V)$ itself is 0 under the original generator. For the following discussion, recall that (a) $G(V, L)$ has been set to 0 for events not in classes 1 or 4, and (b) $P(\mathbb{H})$ and $P(\mathbb{S})$ satisfy $0 < P(\mathbb{H}), P(\mathbb{S}) < 1$.

*Condition on* $\Lambda$. If a given $V$ satisfies $P_{\mathbb{H}}(V) = 0$ and $P_{\mathbb{S}}(V) > 0$, then $\Lambda(V)$ must be 0. Likewise, if $P_{\mathbb{S}}(V) = 0$ and $P_{\mathbb{H}}(V) > 0$, then $\Lambda(V)$ must be 0. From (75), (78), and the fact that $\Delta_{\{f\}}(\ell, \ell_{\text{start}})$ is guaranteed to be non-zero, it can be seen that the conditions on $\Lambda$ are automatically satisfied.[26]

*Condition on* $\Phi_{\text{un}}$. For an event satisfying $P_{\mathbb{H}}(V) > 0$ and $P_{\mathbb{S}}(V) > 0$, $\Phi_{\text{un}}(\text{rej-list}; V)$ should be 0 for impossible values of `rej-list`. The Heaviside step functions $\Theta$ in (74) ensure this condition, for the most part. The only additional requirements are that[27]

$$h_c(\ell, z) = 0, \qquad\qquad \forall(\ell, c, z) : g_c(\ell, z) = 0, \tag{84}$$

$$h_c(\ell, z) = f_c(\ell, z), \qquad\qquad \forall(\ell, c, z) : g_c(\ell, z) = f_c(\ell, z). \tag{85}$$

These ensure that if a proposal $(\ell', c', z')$ would either (a) never be sampled or (b) never be rejected, then MC-histories with that proposal in the rejected-proposals-list would have $\Phi_{\text{un}}(\text{rej-list}; V) = 0$. The condition in (85) can be ensured by simply choosing the proposal kernels $\{g\}$ to be strictly greater than the splitting kernels $\{f\}$; this is the case in the base event-generator from [16] used in this paper.

*A note on the base event generator.* As a side note, a necessary condition for the original event generator to have proper coverage over $F^{(\text{NBL})}$ is that

$$g_c(\ell, z) \neq 0, \qquad\qquad \forall(\ell, c, z) : f_c(\ell, z) \neq 0, \tag{86}$$

or, equivalently,

$$f_c(\ell, z) = 0, \qquad\qquad \forall(\ell, c, z) : g_c(\ell, z) = 0. \tag{87}$$

While this condition is satisfied by the base event generator used in this paper, the converse is not satisfied. In other words, there are values of $(\ell, c, z)$ such that $f_c(\ell, z) = 0$ and $g_c(\ell, z) > 0$.

**One more simplification.**   The natural next step would be to model (i.e., parameterize) the functions $h_c$-s in such a way that their indefinite integrals with respect to $(\ell, z)$ have closed-form expressions (so that $\Delta_{\{h\}}$ can be computed directly), and fit the model to the target functions $f_c$-s. However, there is another simplification that can be performed at this stage. Note, from (83), that different $\mathbb{S}$-histories leading to the same visible-event $V$, with the same sequence of rejected scales $[\ell'_1, \ldots, \ell'_{N_{\text{rej}}}]$, can have different ARCANE weights depending on the values of $(c'_i, z'_i)$-s. This particular weight-variability would be eliminated if

$$\frac{g_c(\ell, z) - h_c(\ell, z)}{g_c(\ell, z) - f_c(\ell, z)} \equiv \Psi(\ell), \qquad\qquad \forall(\ell, c, z) : g_c(\ell, z) \neq 0 \text{ and } g_c(\ell, z) \neq f_c(\ell, z) \tag{88}$$

---

[26]Had $\widehat{P}_{\mathbb{S}}^{\{h\}}(V)$ in $\Lambda(V)$ been chosen differently, with the $f_c(\ell, z)$ in the right-hand-side of (77) replaced with $h_c(\ell, z)$, then one would have needed an additional condition that $h_c(\ell, z)$ must equal 0 for all $(\ell, c, z)$ satisfying $f_c(\ell, z) = 0$, in order to satisfy the condition on $\Lambda$. This additional criterion will be violated in this paper; see footnote 28. The expression for $W^{(\text{ARCANE})}(V, \mathbb{S}, \text{rej-list})$ would also be slightly different for this choice of $\widehat{P}_{\mathbb{S}}^{\{h\}}(V)$.

[27]The condition in (84) is just a special case of (85), since $g_c(\ell, z) = 0$ implies $f_c(\ell, z) = 0$.

for some function $\Psi$ of $\ell$. Under this assumption, using (84), (85), (87), and (23c), it follows that

$$\sum_{c=1}^{C} \int_{\mathbb{R}} \mathrm{d}z \left[ g_c(\ell, z) - h_c(\ell, z) \right] \equiv \Psi(\ell) \sum_{c=1}^{C} \int_{\mathbb{R}} \mathrm{d}z \left[ g_c(\ell, z) - h_c(\ell, z) \right], \tag{89}$$

$$\implies \quad \Psi(\ell) \equiv \frac{\kappa_{\{g\}}(\ell) - \kappa_{\{h\}}(\ell)}{\kappa_{\{g\}}(\ell) - \kappa_{\{f\}}(\ell)}. \tag{90}$$

Plugging this into (83), we get

$$\boxed{\begin{aligned} W^{\text{(ARCANE)}}(V, \mathbb{S}, \texttt{rej-list}) &= \frac{W_{\mathbb{S}}^{\text{(NBL)}}(V_{\text{lo}}, \varphi)}{P(\mathbb{S})} \\ &+ \left[ \frac{P_{\mathbb{H}}(V) \left( W_{\mathbb{H}}^{\text{(NBL)}}(V) - \frac{P(\mathbb{H})}{P(\mathbb{S})} W_{\mathbb{S}}^{\text{(NBL)}}(V_{\text{lo}}, \varphi) \right)}{P(\mathbb{H}) P_{\mathbb{H}}(V) + P(\mathbb{S}) \widehat{P}_{\mathbb{S}}^{\{h\}}(V)} \right] \left[ \prod_{i=1}^{N_{\text{rej}}} \left( \frac{\kappa_{\{g\}}(\ell_i') - \kappa_{\{h\}}(\ell_i')}{\kappa_{\{g\}}(\ell_i') - \kappa_{\{f\}}(\ell_i')} \right) \right]. \end{aligned}} \tag{91}$$

This is the final expression that will be used to compute the ARCANE weights for $\mathbb{S}$-events of class 4 in this paper. Note that in order to use this expression, one needs to be able to compute the functions $\kappa_{\{g\}}$ and, more importantly, $\kappa_{\{f\}}$. It so happens that there exist closed-form expressions for these functions, for the choice of splitting and proposal kernels used in this paper. In addition to eliminating the weight-variance associated with the variability in $(c_i', z_i')$-s, this approach offers another advantage. To use this expression, one does not need to model and fit the functions $h_1, \ldots, h_C$. Instead, one only needs to model $\kappa_{h_1}, \ldots, \kappa_{h_C}$ (which are functions of only one variable), since $\Delta_{\{h\}}$ depends on $\{h\}$ only via $\kappa_{\{h\}}$. Similar to (84) and (85), the conditions on $\kappa_{\{h\}}$ to ensure proper coverage are

$$\boxed{\begin{aligned} \kappa_{\{h\}}(\ell) &= 0, &\forall \ell &: \kappa_{\{g\}}(\ell) = 0, \tag{92} \\ \kappa_{\{h\}}(\ell) &= \kappa_{\{g\}}(\ell), &\forall \ell &: \kappa_{\{f\}}(\ell) = \kappa_{\{g\}}(\ell). \tag{93} \end{aligned}}$$

For completeness, the choice made in (88) corresponds to setting $h_c(\ell, z)$ as follows:

$$h_c(\ell, z) = \begin{cases} g_c(\ell, z) - \left( \dfrac{\kappa_{\{g\}}(\ell) - \kappa_{\{h\}}(\ell)}{\kappa_{\{g\}}(\ell) - \kappa_{\{f\}}(\ell)} \right) \left( g_c(\ell, z) - f_c(\ell, z) \right), \\ \qquad\qquad\qquad\qquad \text{if } \kappa_{\{g\}}(\ell) \left[ \kappa_{\{g\}}(\ell) - \kappa_{\{f\}}(\ell) \right] \neq 0, \\[2mm] 0, \qquad\qquad\qquad\quad \text{if } \kappa_{\{g\}}(\ell) = 0, \\[2mm] f_c(\ell, z), \qquad\qquad\quad \text{if } \kappa_{\{g\}}(\ell) = \kappa_{\{f\}}(\ell), \end{cases} \tag{94}$$

It can be seen from (87), (92), and (93) that this choice of $h_c(\ell, z)$ satisfies (84) and (85).[28]

The expression in (91) can be alternatively derived as follows: Since all events with the same value of $V$ have the same weight under different $\mathbb{S}$-histories, it is convenient to merge these histories where possible. In this context, merging histories means to first (a) compute the overall probability density of the group of histories being merged, and the (b) eliminate the hidden-attributes which identify the individual histories within the merged group. Figure 4 depicts one way to group $\mathbb{S}$-histories: All histories with the same value of $[\ell_1', \ldots, \ell_{N_{\text{rej}}}']$ but

---

[28]For $(\ell, c, z)$ such that $f_c(\ell, z) = 0$ and $g_c(\ell, z) \neq 0$, $h_c(\ell, c)$ does not necessarily equal 0. If one wants to satisfy this condition as well, while keeping the form of $h_c(\ell, z)$ in (94), then one would have to modify the proposal kernels $\{g\}$ to be zero where the splitting kernels $\{f\}$ are zero, e.g., for $z \notin [z_-, z_+]$.

different values for $(c_i', z_i')$-s will be under one group. From (23f) and (23h), it can be seen that the functions $\kappa_{\{f\}}$ and $\kappa_{\{g\}}$ are the $(c, z)$-independent counterparts (to be used in a veto algorithm without explicit competing channels or the $z$ parameter) to the kernels $\{f\}$ and $\{g\}$. This correspondence works at all the relevant levels. For example, in the unweighted veto algorithm, the average acceptance probability of a proposal $(\ell', c', z')$, for a fixed value of $\ell'$ is given by

$$
\frac{\sum_{c'=1}^{C} \int_{\mathbb{R}} \mathrm{d}z \; g_{c'}(\ell', z') \, \alpha_c(\ell', z')}{\sum_{c'=1}^{C} \int_{\mathbb{R}} \mathrm{d}z \; g_{c'}(\ell', z')} = \frac{\sum_{c'=1}^{C} \int_{\mathbb{R}} \mathrm{d}z \; f_{c'}(\ell', z')}{\sum_{c'=1}^{C} \int_{\mathbb{R}} \mathrm{d}z \; g_{c'}(\ell', z')} = \frac{\kappa_{\{f\}}(\ell')}{\kappa_{\{g\}}(\ell')}, \tag{95}
$$

which is the $(c, z)$-independent analog of $\alpha_c(\ell, z)$ being $f_c(\ell, z)/g_c(\ell, z)$. By performing the merging of $\mathbb{S}$-histories described here upfront and proceeding with the derivation of the AR-CANE weight functions as before, one can derive (91) directly. However, the expression in (83) is provided for a general case when one may not have closed-form expressions for $\kappa_{\{f\}}$.

### 3.5 Constructing the $z$-Independent Redistribution Kernel Function

The parton showers in this work use the spin-averaged versions of the Catani–Seymour splitting functions [19]. Both emission channels relevant in this paper have the same $z$-dependent splitting kernel function $f_1 \equiv f_2 \equiv f_{qq}$, given by

$$
f_{qq}(\ell, z ; M^2) \equiv \frac{\alpha_s(t)}{2\pi} C_F \; (1-y) \left( \frac{2}{1-z(1-y)} - (1+z) \right) \Theta(z - z_-) \, \Theta(z_+ - z) \\
\times \, \Theta\left( \frac{M^2}{4} - t \right), \tag{96}
$$

where the parameter $M$ is the total center-of-momentum frame energy of the emission system, $\alpha_s(t)$ is the strong coupling at scale $t$, $C_F$ is a QCD color-factor equal to 4/3, and, as defined before in Section 2.4,

$$
t \equiv e^\ell \; \mathrm{GeV}^2, \qquad y \equiv \frac{t}{M^2 z(1-z)}, \qquad z_\pm \equiv \frac{1}{2} \left( 1 \pm \sqrt{1 - \frac{4t}{M^2}} \right). \tag{97}
$$

The specific functional form of $\alpha_s(t)$ used in the present work can be found in the codebase associated with it, as well as Ref. [16]. The function $f_{qq}$ was symbolically integrated with respect to $z$ using SageMath [31], with Maxima [32] as the backend, to get[29]

$$
\kappa_{f_1}(\ell ; M^2) \equiv \kappa_{f_2}(\ell ; M^2) \equiv \kappa_{f_{qq}}(\ell ; M^2) \equiv \int_{\mathbb{R}} \mathrm{d}z \, f_{qq}(\ell, z ; M^2) \tag{98}
$$

$$
= \frac{\alpha_s(t)}{2\pi} C_F \; \Theta\left( \frac{M^2}{4} - t \right) \times \left( - z - \frac{z^2}{2} - 2\rho \ln(1-z) - \frac{(1-\rho)\rho \ln(z)}{1+\rho} \right.
$$

$$
\left. - \frac{\ln\left(\rho + (1-z)^2\right)}{1+\rho} + \frac{2\sqrt{\rho} \arctan\left( \frac{1-z}{\sqrt{\rho}} \right)}{1+\rho} \right) \Bigg|_{z=z_-}^{z=z_+} , \tag{99}
$$

---

[29]The result was subsequently verified by symbolic differentiation using SymPy [33].

where $\rho \equiv t/M^2$. Since $z_+$ and $z_-$ are fully determined by $\rho$, one can write

$$\kappa_{f_{qq}}(\ell \; ; \; \tilde{p}_{\{ij\}}, \tilde{p}_k) \equiv \alpha_s(t) \times \widetilde{\kappa}_{f_{qq}}(\rho) \times \Theta\left(\frac{M^2}{4} - t\right), \tag{100}$$

with $\widetilde{\kappa}_{f_{qq}}(\rho)$ appropriately defined for $0 \le \rho \le M^2/4$. The $\alpha_s(t)$ function used in the base event generator and $\widetilde{\kappa}(\rho)$ both have a closed-form expressions, so $\kappa_{f_{qq}}(\ell \; ; \; M^2)$ can be directly computed.

For completeness, the proposal kernel function used in this paper, for the relevant emission channels, is $g_1 \equiv g_2 \equiv g_{qq}$, given by

$$g_{qq}(\ell, z) \equiv \frac{\alpha_s(t_{\text{cutoff}})}{2\pi} C_F \frac{2}{1-z} \Theta\left(z - z_{\min}\right) \Theta\left(z_{\max} - z\right), \tag{101}$$

where $t_{\text{cutoff}}$, $z_{\min}$, and $z_{\max}$ are $t$, $z_-$, and $z_+$, respectively, computed for $\ell = \ell_{\text{cutoff}}$. using the fact that $\alpha_s$ increases with decreasing $t$, it can be seen that $g_{qq}(\ell, z) > f_{qq}(\ell, z \; ; \; M^2)$ for all $(\ell, z)$. Integrating this with respect to $z$ leads to

$$\kappa_{g_{qq}}(\ell) \equiv \int_{\mathbb{R}} dz \; g_{qq}(\ell, z) = \frac{\alpha_s(t_{\text{cutoff}})}{2\pi} (2 C_F) \ln\left(\frac{1 - z_{\min}}{1 - z_{\max}}\right). \tag{102}$$

**Modeling the $z$-independent redistribution kernel.** The task at hand is to find an approximation for $\kappa_{f_{qq}}$, namely $\kappa_{h_{qq}}$, whose indefinite integral with respect to $\ell$ can be computed exactly. Note that different values for $M^2$ lead to different $\kappa_{f_{qq}}(\cdot \; ; \; M^2)$. In principle, for a fixed value of $M^2$, one can directly perform a one-dimensional fit to get $\kappa_{h_{qq}}(\ell)$. However the fitting procedure would have to be repeated for different values of $M^2$. For lepton collisions at a fixed total collision energy, this would mean repeating the fitting procedure when changing the global $E_{\text{cms}}$ parameter, which might not be too difficult. On the other hand, when using the technique for hadron collisions (after handling other complications associated with them), the fitting procedure would need to be repeated for each event,[30] which is cumbersome and possibly infeasible. A more practical approach would be to perform a two-dimensional fit for $\kappa_{h_{qq}}(\ell \; ; \; M^2)$. Alternatively, in this paper $\kappa_{h_{qq}}$ is modeled as follows:

$$\boxed{\kappa_{h_{qq}}(\ell \; ; \; M^2) \equiv \hat{A}(\ell) \times \hat{B}\left(\ln\left(\frac{M^2}{4 \, \text{GeV}^2}\right) - \ell\right) \times \Theta\left(\ln\left(\frac{M^2}{4 \, \text{GeV}^2}\right) - \ell\right),} \tag{103}$$

where $\hat{A}$ and $\hat{B}$ approximate $\alpha_s$ and $\widetilde{\kappa}_{f_{qq}}$, respectively, as follows

$$\hat{A}(\ell) \approx A(\ell) \equiv \alpha_s\left(e^\ell \, \text{GeV}^2\right), \qquad \hat{B}(u) \approx B(u) \equiv \widetilde{\kappa}_{f_{qq}}\left(e^{-u}/4\right). \tag{104}$$

The shift in the argument of the function $\hat{B}$ in (103) has been chosen so that $B(0) = 0$. Now, $\kappa_{h_{qq}}$ can be constructed for a range of values of $M^2$ with just two one-dimensional fits (in appropriate domains) for the functions $\hat{A}$ and $\hat{B}$, as long as their product in (103) can be integrated with respect to $\ell$, for any choice of $M^2$. There are several candidate-models for the functions $\hat{A}$ and $\hat{B}$, e.g., polynomials, step-functions, polynomial splines, as well as variants of these. Polynomial fitting (with a slight modification) is the approach used in the work. Polynomial fitting can pose several difficulties, e.g., precision of coefficients needing to increase as the degree of the polynomial increases, poor extrapolation properties beyond the fit

---

[30]It is acceptable to use different choices for $\{h\}$ to reweight different events, as long as the procedure used to pick $\{h\}$ is independent of the hidden attributes of the event being reweighted, i.e., `type`, $c_{\text{gen}}$, and/or `rej-list`.

region (note that extrapolation is not required in this work), etc. Furthermore, integrating the product of two polynomials (whose arguments $u$ and $\ell$ differ by an additive shift) may be computationally too expensive for one's purposes. So, a production-quality implementation of ARCANE reweighting could consider polynomial splines or other more sophisticated and/or computationally cheaper alternatives. Polynomial fitting is chosen here for its simplicity to implement and describe, and its adequacy for the purposes of this work.

For reasons described in footnote 34, only tangentially related to the actual quality of the ARCANE implementation, this work tries to reduce the maximum (over $\ell$-values) unsigned relative error in $\kappa_{h_{qq}}$ with respect to $\kappa_{f_{qq}}$. To this end, it is important to approximate $B(u)$ well near $u = 0$. It can be shown that[31]

$$\lim_{u \to 0^+} \frac{B(u)}{u^{3/2}} = \frac{C_F}{6\pi} . \tag{105}$$

Based on this, $\hat{A}$ and $\hat{B}$ are modeled as follows:[32]

$$\hat{A}(\ell) = \sum_{i=0}^{D_A} a_i \, \ell^i , \qquad\qquad \hat{B}(u) = u^{3/2} \sum_{i=0}^{D_B} b_i \, u^i . \tag{106}$$

Given a value of $M^2$, one can integrate the resulting $\kappa_{h_{qq}}$ with respect to $\ell$, e.g., by first transforming $\hat{A}$ into a polynomial in $u$. This way $\Delta_{h_c}(\ell, \ell_{\text{start}})$ can be computed exactly for both channels, and hence so can $\Delta_{\{h\}}(\ell, \ell_{\text{start}})$.

**Polynomial-fitting.** Given the Heaviside step function in (103), the function $\hat{A}$ needs to be fit at least from $\ell = \ell_{\text{cutoff}}$ to $\ell = \ln(M^2/(4\,\text{GeV}^2))$. Likewise, $\hat{B}$ needs to be fit at least from $u = 0$ to $u = \ln(M^2/(4\,\text{GeV}^2)) - \ell_{\text{cutoff}}$.

In the Monte Carlo experiments performed in this work, $E_{\text{cms}} \equiv M$ is set to 91.2 GeV. Let us say the maximum value of interest for $M$ (across different runs of the generator, with different parameter-choices) is 110 GeV. Likewise, let us say the minimum value of interest for $\ell_{\text{start}}$ is 0 (which corresponds to $t_{\text{cutoff}} = 1\,\text{GeV}^2$). For good measure, $\hat{A}$ was fit over the interval $\ell \in [\ln(1/2), \ln(120^2/4)]$. Likewise, $\hat{B}$ was fit over the interval $u \in [0, \ln(120^2/4) - \ln(1/2)]$. The polynomial degrees $D_A$ and $D_B$ were chosen to be 11. The coefficients $\{a_i\}_{i=0}^{D_A}$ were fit by (i) sampling 1001 equidistant $\ell$-values from the relevant range, including the endpoints, (b) computing the target function $A(\ell)$ at the chosen $\ell$-values using the known closed-form expression, (c) performing a least-square fit, and (d) rounding the coefficients to 6 significant figures. The same procedure was used to fit the coefficients $\{b_i\}_{i=0}^{D_B}$, with $B(u)\,u^{-3/2}$ as the target function. The resulting coefficient-values are listed in Table 1.

**A bad approximation.** To demonstrate that the quality of the approximation in this step does not affect the validity of the resulting event-weighting, a second "bad" approximation of $\kappa_{f_{qq}}$, namely $\kappa_{h_{qq}^{\text{bad}}}$, is chosen as follows:

$$\kappa_{h_{qq}^{\text{bad}}}(\ell \; ; \; M^2 = 91.2^2\,\text{GeV}^2) \quad \equiv \quad 0.3 . \tag{107}$$

---

[31]The exponent 3/2 was first guesstimated using numerical differentiation: If $B(u) \approx c\,u^\gamma$ for $0 \le u \ll 1$, then $\gamma \approx [u/B(u)]\,[B(u+\epsilon) - B(u-\epsilon)]/(2\,\epsilon)$ for $0 < \epsilon \ll b \ll 1$. The limit was subsequently derived using SymPy.

[32]In principle, one can also explicitly factor an approximate behavior of $A(\ell)$ into $\hat{A}(\ell)$ to help with the approximation. But integrating the product of $\hat{A}$ and $\hat{B}$ would be more complicated in that case, involving, e.g., hypergeometric functions. Furthermore, for the specific form of $\alpha_s(t)$ [16] used in this project, there exist closed-form expressions (involving generalized hypergeometric functions) for the indefinite integral of $A(\ell)\,\ell^\gamma$ with respect to $a$. For small, non-negative, integral values of $\gamma$, these expressions are relative simple to evaluate, so one may not have to approximate $A(\ell)$ at all. However, these sophistications are deemed unnecessary here.

Table 1: Fitted values for the coefficients $\{a_i\}_{i=1}^{D_A}$ and $\{b_i\}_{i=1}^{D_B}$, which model $\kappa_{h_{qq}}$ as per (103) and (106). As part of the fitting procedure, the coefficients were rounded to 6 significant figures.

| | | |
|---|---|---|
| $a_0 = +4.39740 \times 10^{-1}$ | $a_1 = -1.70355 \times 10^{-1}$ | $a_2 = +7.44694 \times 10^{-2}$ |
| $a_3 = -2.25743 \times 10^{-2}$ | $a_4 = +1.40163 \times 10^{-3}$ | $a_5 = +1.22857 \times 10^{-3}$ |
| $a_6 = -2.33848 \times 10^{-4}$ | $a_7 = -5.37434 \times 10^{-5}$ | $a_8 = +2.53591 \times 10^{-5}$ |
| $a_9 = -3.83532 \times 10^{-6}$ | $a_{10} = +2.68811 \times 10^{-7}$ | $a_{11} = -7.38850 \times 10^{-9}$ |
| $b_0 = +7.07322 \times 10^{-2}$ | $b_1 = +7.43341 \times 10^{-2}$ | $b_2 = -4.82073 \times 10^{-2}$ |
| $b_3 = +1.55139 \times 10^{-2}$ | $b_4 = -2.74400 \times 10^{-3}$ | $b_5 = +1.12259 \times 10^{-4}$ |
| $b_6 = +7.28707 \times 10^{-5}$ | $b_7 = -2.07329 \times 10^{-5}$ | $b_8 = +2.85740 \times 10^{-6}$ |
| $b_9 = -2.28593 \times 10^{-7}$ | $b_{10} = +1.01511 \times 10^{-8}$ | $b_{11} = -1.94279 \times 10^{-10}$ |

**Approximation results.**   The $z$-independent splitting kernel $\kappa_{f_{qq}}$, proposal kernel $\kappa_{g_{qq}}$, redistribution kernel $\kappa_{h_{qq}}$, and the bad redistribution kernel $\kappa_{h_{qq}^{\mathrm{bad}}}$ are all depicted in Figure 5, for $M = 91.2\,\mathrm{GeV}$. The top-left- and top-right-panels of Figure 5 depict the kernels on a linear and log-scale, respectively, as a function of $\ell$ from $\ell_{\mathrm{cutoff}}$ to $\ell_{\mathrm{start}}$. The bottom-left- and bottom-right-panels of Figure 5 depict the signed absolute and relative errors, respectively, in $\kappa_{h_{qq}}$ with respect to $\kappa_{f_{qq}}$. The left- and middle-panels of Figure 6 show $\kappa_{f_{qq}}$ and $\kappa_{h_{qq}}$, respectively, as a function of both $M$ and $\ell$, for $M \in [1\,\mathrm{GeV}, 110\,\mathrm{GeV}]$ and $\ell \in [0, \ln(110^2)]$. The right-panel of Figure 6 shows the corresponding signed relative error in $\kappa_{h_{qq}}$ with respect to $\kappa_{f_{qq}}$. It can be seen from these figures that $\kappa_{h_{qq}}$ is a good approximation for $\kappa_{f_{qq}}$, while $\kappa_{h_{qq}^{\mathrm{bad}}}$ is not.

**Bound on the ideal weight function $W^*(V)$.**   Note from Figure 5 that the (unsigned) relative error is upper-bounded by 0.005, i.e., for $M = 91.2\,\mathrm{GeV}$ and $0 \le \ell < \ln(91.2^2/4)$,

$$(1 - \epsilon_{\mathrm{rel}}^{\mathrm{ub}})\,\kappa_{f_{qq}}(\ell \; ; \; M^2) \le \kappa_{h_{qq}}(\ell \; ; \; M^2) \le (1 + \epsilon_{\mathrm{rel}}^{\mathrm{ub}})\,\kappa_{f_{qq}}(\ell \; ; \; M^2), \qquad (108)$$

where $\epsilon_{\mathrm{rel}}^{\mathrm{ub}} = 0.005$. From this, it can be shown that

$$\left[\Delta_{\{h\}}(\ell, \ell_{\mathrm{start}})\right]^{1/(1 - \epsilon_{\mathrm{rel}}^{\mathrm{ub}})} \le \Delta_{\{f\}}(\ell, \ell_{\mathrm{start}}) \le \left[\Delta_{\{h\}}(\ell, \ell_{\mathrm{start}})\right]^{1/(1 + \epsilon_{\mathrm{rel}}^{\mathrm{ub}})}. \qquad (109)$$

Since $W^{(\mathrm{ARCANE})}(V, H)$ in (80) varies monotonically (either non-decreasing or non-increasing) with $\widehat{P}_{\mathbb{S}}^{\{h\}}(V)$, (80) and (109) can be used to get upper- and lower-bounds on the ideal value of $W^{(\mathrm{ARCANE})}(V, H)$, namely $W^*(V)$.[34] Let $W^{*,\mathrm{ub}}(V)$ and $W^{*,\mathrm{lb}}(V)$ be the corresponding upper- and lower-bounds on $W^*(V)$.

## 4   Experiments and Results

As mentioned before, the event generator was run with $E_{\mathrm{cms}} = 91.2\,\mathrm{GeV}$ and $\ell_{\mathrm{cutoff}} = 0$. The value of $P(\mathbb{H})$ and $P(\mathbb{S})$ were set to 0.25 and 0.75, respectively. These values were chosen to conform with the defaults in Ref. [16]. Three independent datasets labeled "small", "medium", and "large" were generated, with a total of $10^4$, $10^5$, and $10^6$ MC@NLO events, respectively.

---

[33]As mentioned previously, the quarks are taken to be massless in the event kinematics.

[34]One can also use an upper-bound on the (unsigned) **absolute** error in $\kappa_{h_{qq}}$, say $\epsilon_{\mathrm{abs}}^{\mathrm{ub}}$, to derive bounds on $W^*(V)$. In this case, the corresponding bounds on $\Delta_{\{f\}}$ will depend on $\ell$, in addition to $\Delta_{\{h\}}$ and $\epsilon_{\mathrm{abs}}^{\mathrm{ub}}$. One reason for trying to reduce the relative error in $\kappa_{h_{qq}}$ is to avoid this minor inconvenience (i.e., the $\ell$-dependence).

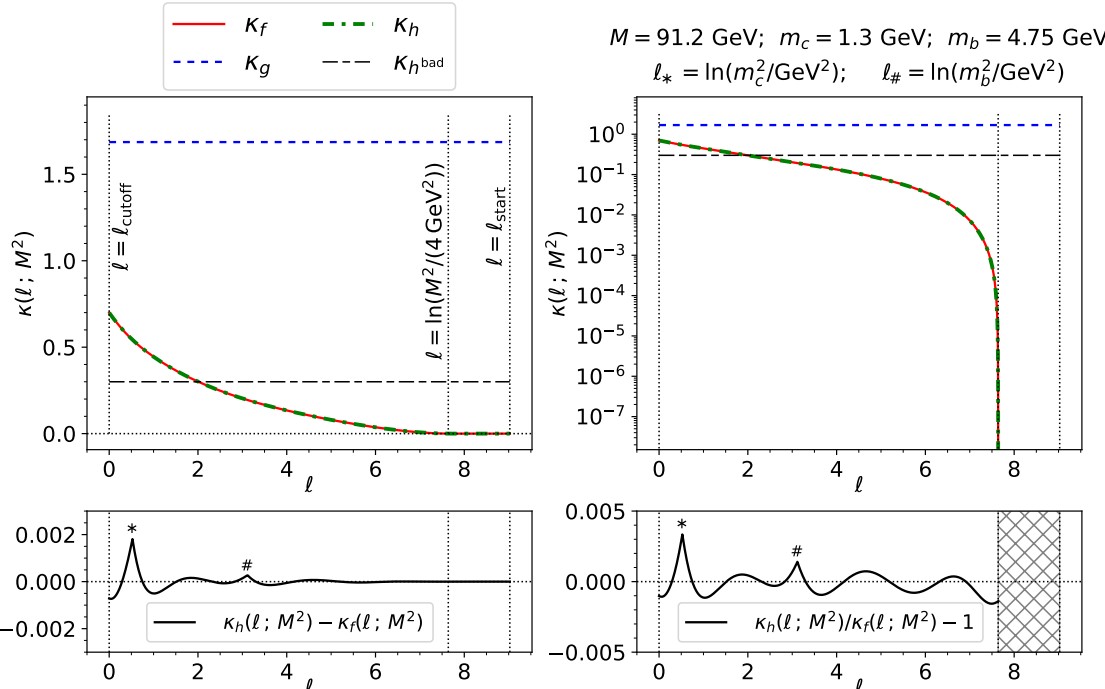

Figure 5: Plots depicting the $z$-independent splitting kernel $\kappa_{f_{qq}}$, proposal kernel $\kappa_{g_{qq}}$, and redistribution kernels $\kappa_{h_{qq}}$ and $\kappa_{h_{qq}^{\text{bad}}}$ used in this paper. For convenience, the subscript $qq$ has been dropped from the plot labels and legends. The top-left and top-right panels depict the kernels on a linear-scale and a log-scale, respectively. The bottom-left and bottom-right plot depict signed the absolute and relative errors, respectively, in $\kappa_{h_{qq}}$ with respect to $\kappa_{f_{qq}}$. All plots in this figure are for $M = 91.2\,\text{GeV}$. The quark mass parameters $m_c$ and $m_b$ are used in the functional form of the running strong coupling $\alpha_s(t)$.[33] The sharp kinks in the bottom two panels occur at $t = m_b^2 = (4.75\,\text{GeV})^2$ and $t = m_c^2 = (1.3\,\text{GeV})^2$, where the $\alpha_s(t)$ is non-differentiable. The kernels $\kappa_{f_{qq}}$ and $\kappa_{h_{qq}}$ are both zero in the hatched region of the botton-right panel.

Most of the analyses were performed using the medium dataset, with the small and large datasets used for specific purposes, as described later.

Around 95% of the events in each dataset belonged to either class 1 or class 4. For each of these events, in addition to the original event-weight $W^{(\text{ORIG})}$, the modified event weights $W^{(\text{NBL})}$ and $W^{(\text{ARCANE})}$ were computed using (53), (77), (80), (91), (103), and (106). Furthermore, using $\kappa_{h_{qq}^{\text{bad}}}$ in (107) as the redistribution kernel, the ARCANE weight $W^{(\text{ARCANE,BAD})}$ was computed for the events. Finally, using the bounds on $\Delta_{\{f\}}(\ell, \ell_{\text{start}})$ in (109), upper- and lower-bounds on $W^*(V)$, namely $W^{*,\text{ub}}(V)$ and $W^{*,\text{lb}}(V)$, respectively, were also computed for all events in class 1 or class 4.

**Histograms of weights.** Figure 7 shows two-dimensional histograms of a few different pairs of weight functions, as heatmaps, for the medium dataset. The top-left panel of Figure 7 shows that merging the different $\mathbb{H}$-channels (to get $W^{(\text{NBL})}$ from $W^{(\text{ORIG})}$) leads to a substantial reduction in the variance of the event-weights. The weights of $\mathbb{S}$-events are unaffected in this step; these events feature along the diagonal on this panel. The overall negative weights fraction (assuming an unweighting step is performed after the event-generation) reduces from around 2.78% under the "ORIG" weighting scheme to around 2.25% under the "NBL" weighting

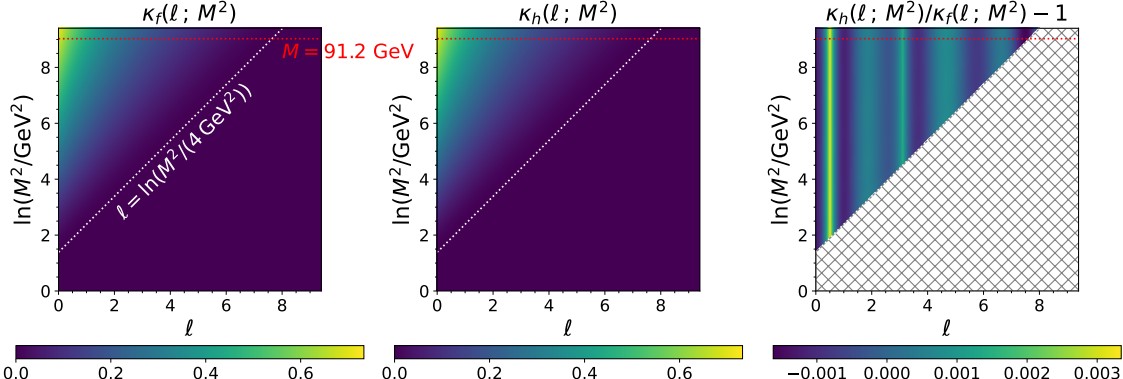

Figure 6: Heatmaps depicting the $z$-independent splitting kernel $\kappa_{f_{qq}}$ (left panel) and redistribution kernel $\kappa_{h_{qq}}$ (middle panel), as a function of $\ell$ and $M$, for $2\,\text{GeV} \leq M \leq 110\,\text{GeV}$ and $0 \leq \ell < \ln(M^2/(4\,\text{GeV}^2))$. The right panel shows the signed relative error in $\kappa_{h_{qq}}$ with respect to $\kappa_{f_{qq}}$. For convenience, the subscript $qq$ has been dropped from the plot labels and legends. The kernels $\kappa_{f_{qq}}$ and $\kappa_{h_{qq}}$ are both zero in the hatched region of the left panel.

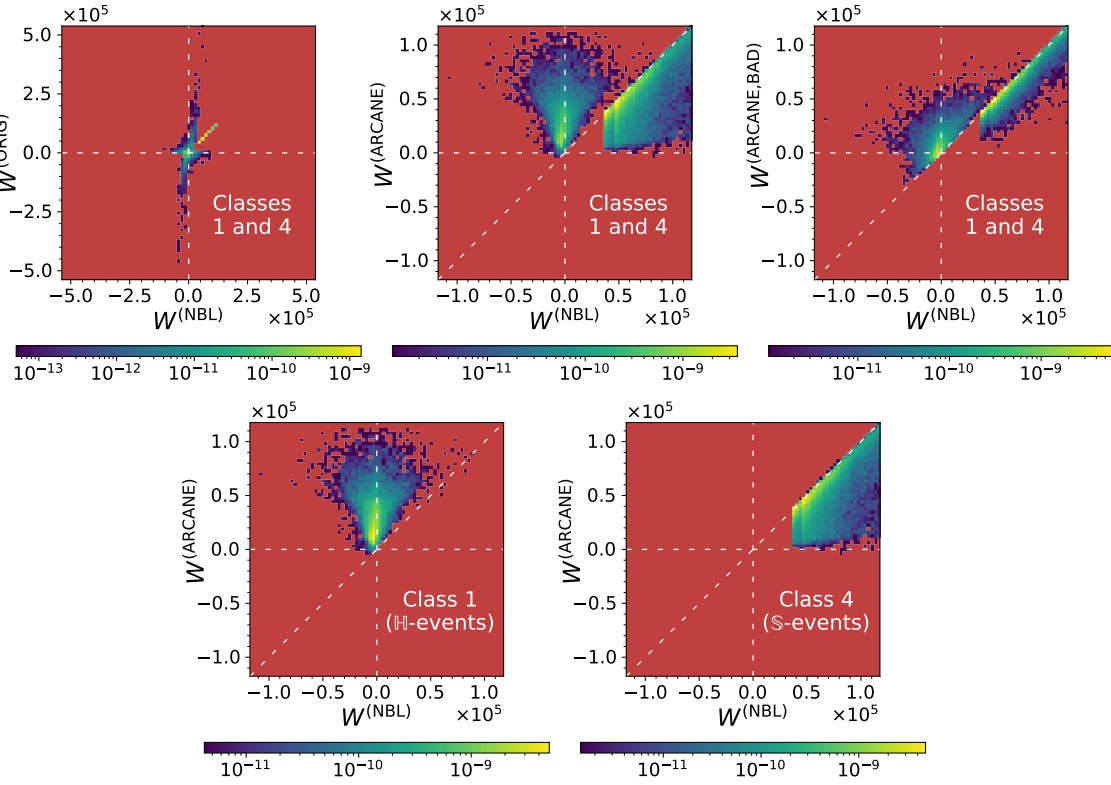

Figure 7: Two-dimensional unit-normalized histograms of different pairs of event-weight functions, for events from the medium dataset. The top row shows histograms for events in classes 1 and 4 combined. The bottom row shows histograms of $W^{(\text{NBL})}$ and $W^{(\text{ARCANE})}$ for events in class 1 only (bottom-left panel) and class 4 only (bottom-right panel). The solid maroon background in the heatmaps correspond to empty bins. The event weights are in units of picobarn in every panel in this figure.

scheme; this is also listed in Table 2. Just among the class-1 $\mathbb{H}$-events, the negative weights fraction (assuming an unweighting step is performed after the event-generation) increased from around 63.7% under the "ORIG" weighting scheme to around 69.3% under the "NBL" weighting scheme. This is consistent with a slight reduction in the sign problem from merging the two $\mathbb{H}$-histories.

The top-middle and top-right panels of Figure 7 depict the joint-distribution of the AR-CANE weights and $W^{(\text{NBL})}$; the top-middle panel uses $W^{(\text{ARCANE})}$ derived using the polynomial-fitting-based redistribution kernel and the top-right panel uses $W^{(\text{ARCANE,BAD})}$ derived using the "bad", constant redistribution kernel. Due to the redistribution of contributions under ARCANE reweighting, the weights of some events increase while the weights of others decrease. Importantly, note from the top-middle panel that, while $W^{(\text{NBL})}$ can take large negative values, $W^{(\text{ARCANE})}$ is almost completely non-negative. Only a small fraction of events have $W^{(\text{ARCANE})} < 0$, and even for those events, the magnitudes of the negative weights are relatively small; these remaining negative-weighted events will be discussed briefly later. Due to its sub-par nature, a larger fraction of events have negative values for $W^{(\text{ARCANE,BAD})}$.

The bottom two panels of Figure 7 depict $W^{(\text{NBL})}$ vs $W^{(\text{ARCANE})}$ for class 1 and class 4 events, separately. It can be seen that, for the most part, the weights of $\mathbb{H}$-events have increased under the ARCANE reweighting while the weights of $\mathbb{S}$ events have decreased. There are two main reasons for this. First, for the process under consideration, the $\mathbb{S}$-contributions (prior to AR-CANE reweighting) is strictly non-negative, with the negative contributions coming only from $\mathbb{H}$-events. So, it is reasonable that when redistributing contributions, the weights of $\mathbb{H}$-events increase at the expense of $\mathbb{S}$-event-weights. Secondly, for the process under consideration, if we split the quasi density of visible attributes as

$$F^{(\text{ORIG/NBL/ARCANE})}(V) = F^{(\text{NBL})}(V, \mathbb{H}) + F^{(\text{NBL})}(V, \mathbb{S}), \tag{110}$$

then the ratio of $|F^{(\text{NBL})}(V, \mathbb{S})|$ to $|F^{(\text{NBL})}(V)|$ is much larger than $P(\mathbb{S}) = 0.75$, for a large fraction of $V$-values. The $\mathbb{H}$-events only contribute a minor modification to the contribution from the $\mathbb{S}$-events, for the process under consideration. Consequently, in the original event generator, the $\mathbb{H}$- and $\mathbb{S}$-events are disproportionately oversampled and undersampled, respectively. ARCANE reweighting addresses this disparity by moving contributions from $\mathbb{S}$- to $\mathbb{H}$- histories, leading to the observed mostly upward and mostly downward changes in the weights of $\mathbb{H}$- and $\mathbb{S}$-events, respectively.

**Some performance Metrics.** A few different performance metrics discussed in Ref. [12] will be used here, with slightly different notations. The overall variance of weights in a dataset can be decomposed as follows:

$$\mathbb{CV}^2[W] \;\equiv\; \frac{\text{Var}[W]}{\mathbb{E}^2[W]} \;=\; \frac{\text{Var}\big[|W|\big]}{\mathbb{E}^2[W]} \;+\; \mathbb{SP}, \tag{111}$$

where $\mathbb{SP}$ captures the contribution of the sign problem to the weight-variance,[35] and is given by

$$\mathbb{SP} \;\equiv\; \frac{\mathbb{E}^2\big[|W|\big] - \mathbb{E}^2[W]}{\mathbb{E}^2[W]} \;=\; \frac{4\,f_{\text{neg}}\big(1 - f_{\text{neg}}\big)}{\big(1 - 2f_{\text{neg}}\big)^2} \;=\; \frac{1}{\big(1 - 2f_{\text{neg}}\big)^2} - 1, \tag{112}$$

where $f_{\text{neg}}$ in the fraction of events that would have negative weights, **if the events were passed through an additional unweighting step** that makes the absolute value of the event-

---

[35]$\mathbb{SP}$ in this paper corresponds to $\mathbb{SP}[V, H]$, in the notation of Ref. [12].

Table 2: Various performance metrics estimated using the medium dataset, for the different weighting schemes. For $\mathbb{CV}^2[W]$, $\mathrm{Var}[|W|]/\mathbb{E}^2[W]$, and $\mathbb{SP}$ lower values indicate better performance. For $f_{\mathrm{neg}}$, being farther away from 0.5 is better. For $f_{\mathrm{effective}}^{(\mathrm{post\text{-}unwgtng})}$, higher is better.

| | $\mathbb{CV}^2[W]$ | $\dfrac{\mathrm{Var}[|W|]}{\mathbb{E}^2[W]}$ | $\mathbb{SP}$ | $f_{\mathrm{neg}}$ | $f_{\mathrm{effective}}^{(\mathrm{post\text{-}unwgtng})}$ |
|---|---|---|---|---|---|
| ORIG | 0.550 | 0.429 | 0.121 | 0.0278 | 0.892 |
| NBL | 0.507 | 0.411 | 0.0964 | 0.0225 | 0.912 |
| ARCANE,BAD | 0.371 | 0.352 | 0.0192 | 0.00473 | 0.981 |
| ARCANE | 0.221 | 0.221 | $2.31 \times 10^{-5}$ | $5.77 \times 10^{-6}$ | 1.00 |

weights a constant. The impact of the sign problem can be alternatively captured by a post-unweighting-effective-event-fraction defined as

$$f_{\mathrm{effective}}^{(\mathrm{post\text{-}unwgtng})} \equiv \frac{\mathbb{E}^2[W]}{\mathbb{E}^2[|W|]} = \left(1 - 2f_{\mathrm{neg}}\right)^2 = \frac{1}{1 + \mathbb{SP}}. \tag{113}$$

The different performance metrics in (111), (112), and (113) were estimated[36] using the medium dataset, for each weighting scheme, as follows:

$$\mathrm{Estimate}\left[\mathbb{CV}^2[W]\right] \equiv \frac{\mathrm{SampleVariance}[W]}{\left(\mathrm{SampleMean}[W]\right)^2}, \tag{114a}$$

$$\mathrm{Estimate}\left[\frac{\mathrm{Var}[|W|]}{\mathbb{E}^2[W]}\right] \equiv \frac{\mathrm{SampleVariance}[|W|]}{\left(\mathrm{SampleMean}[W]\right)^2}, \tag{114b}$$

$$\mathrm{Estimate}\left[\mathbb{SP}\right] \equiv \mathrm{Estimate}\left[\mathbb{CV}^2[W]\right] - \mathrm{Estimate}\left[\frac{\mathrm{Var}[|W|]}{\mathbb{E}^2[W]}\right], \tag{114c}$$

$$\mathrm{Estimate}\left[f_{\mathrm{neg}}\right] \equiv \frac{1}{2}\frac{\mathrm{SampleMean}[|W| - W]}{\mathrm{SampleMean}[|W|]}, \tag{114d}$$

$$\mathrm{Estimate}\left[f_{\mathrm{effective}}^{(\mathrm{post\text{-}unwgtng})}\right] \equiv \frac{\left(\mathrm{SampleMean}[W]\right)^2}{\left(\mathrm{SampleMean}[|W|]\right)^2}, \tag{114e}$$

where SampleMean and SampleVariance are the standard unbiased estimators for mean and variance, respectively. The results are presented in Table 2. It can be seen that under each of the performance metrics, the weighting schemes follow the same order of performance, namely, from best to worst, (ARCANE) ≻ (ARCANE,BAD) ≻ (NBL) ≻ (ORIG). Note that in principle, one can construct a poor implementation of ARCANE that does worsen performance, even though (ARCANE,BAD) does not, for the present example.

It is important to note that the first two performance metrics, namely $\mathbb{CV}^2[W]$ and $\frac{\mathrm{Var}[|W|]}{\mathbb{E}^2[W]}$, can also be improved using standard alternative methods like (a) improving the implementation of importance sampling and (b) rejection-reweighting or unweighting. In fact, these

---

[36]The uncertainties (e.g., bias and variance) in the estimates of the performance metrics were not explicitly estimated. These uncertainties are expected to be sufficiently small to not affect the conclusions drawn in this paper, from the performance-metric-estimates.

alternative techniques could be used to make $\dfrac{\mathrm{Var}[|W|]}{\mathbb{E}^2[W]}$ identically zero. So, in order to claim a performance improvement from using ARCANE *in terms of these two metrics*, one will also have to consider the additional computational costs involved in performing the relevant techniques, and demonstrate that ARCANE offers a better cost–benefit tradeoff. Such nuanced comparisons are beyond the scope of this work.

On the other hand, there are no known alternative methods to improve the sign-problem-related performance metrics, namely $\mathbb{SP}$, $f_{\mathrm{neg}}$, and $f_{\mathrm{effective}}^{(\mathrm{post\text{-}unwgtng})}$, without also (a) introducing additional biases and/or dependence between datapoints or (b) changing the underlying physics formalism (and correspondingly changing the differential cross-section of the visible attributes). ARCANE reweighting has reduced the fraction of negative events (after unweighting), from around 2.25% under the NBL weighting scheme, down to comparatively negligible levels. If the detector simulation stage is significantly more computationally expensive than event-generation (even after incorporating ARCANE reweighting and unweighting), then AR-CANE (with unweighting) would roughly reduce the overall simulation cost by around

$$\frac{{}^{1}\!/_{0.912} - 1}{{}^{1}\!/_{0.912}} \quad \times \quad 100\% \quad \approx \quad 9\%,$$

when compared to using the NBL weights (with unweighting). This is a simplistic estimate, which ignores the facts that (a) the negative weights fraction is typically different in different regions of the phase space of $V$, (b) different regions in the phase space of $V$ could be important to a given physics analysis to different degrees and have different degrees of uncertainties from other sources, (c) one could preferentially oversample different regions of the phase space of $V$ to suit the needs of the final analyses, etc.

**Distributions of event-attributes.** The weighted distributions of a few visible event-attributes, under the different weighting schemes, are presented as one-dimensional histograms in Figure 8. Each row depicts a single event variable. In order, from top to bottom, the event variables plotted are (i) $M_{\mathrm{q\bar{q}}}/E_{\mathrm{cms}}$, (ii) $M_{\mathrm{qg}}/E_{\mathrm{cms}}$, (iii) $M_{\mathrm{\bar{q}g}}/E_{\mathrm{cms}}$, (iv) $1 - \sqrt{1 - \mathrm{Thrust}}$, and (v) $\sqrt{\mathrm{Spherocity}}$. Here $M_{\mathrm{p_1 p_2}}$ is the invariant mass of the system comprising the final state particles $\mathrm{p_1}$ and $\mathrm{p_2}$, and

$$\mathrm{Thrust} \equiv \frac{\max\limits_{\hat{n}}\left[\sum\limits_{i=1}^{N_{\mathrm{final\text{-}state}}} \left|\vec{p}_i \cdot \hat{n}\right|\right]}{\sum\limits_{i=1}^{N_{\mathrm{final\text{-}state}}} \left|\vec{p}_i\right|} = \frac{\max\limits_{\vec{s}\in\{-1,1\}^{N_{\mathrm{final\text{-}state}}}}\left[\left|\sum\limits_{i=1}^{N_{\mathrm{final\text{-}state}}} s_i\,\vec{p}_i\right|\right]}{\sum\limits_{i=1}^{N_{\mathrm{final\text{-}state}}} \left|\vec{p}_i\right|}, \qquad (115)$$

$$\mathrm{Spherocity} \equiv \frac{16}{\pi^2}\left(\frac{\min\limits_{\hat{n}}\left[\sum\limits_{i=1}^{N_{\mathrm{final\text{-}state}}} \left|\vec{p}_i \times \hat{n}\right|\right]}{\sum\limits_{i=1}^{N_{\mathrm{final\text{-}state}}} \left|\vec{p}_i\right|}\right)^2, \qquad (116)$$

where $N_{\mathrm{final\text{-}state}} = 3$ is the number of final state particles, $\vec{p}_i$ is the three-momentum of the $i$-th final state particle, and $\hat{n}$ represents a three-dimensional unit vector [34–37]. For events with three massless particles $\mathrm{q}$, $\mathrm{\bar{q}}$, and $\mathrm{g}$ in the final state, in the center-of-momentum frame of the system, spherocity can be computed in terms of the energies of the particles and their sum $E_{\mathrm{cms}}$ as follows [35]

$$\mathrm{Spherocity} = \frac{16}{\pi^2}\left(1 - \frac{2E_{\mathrm{q}}}{E_{\mathrm{cms}}}\right)\left(1 - \frac{2E_{\mathrm{\bar{q}}}}{E_{\mathrm{cms}}}\right)\left(1 - \frac{2E_{\mathrm{g}}}{E_{\mathrm{cms}}}\right)\left(\max\left[\frac{E_{\mathrm{q}}}{E_{\mathrm{cms}}}, \frac{E_{\mathrm{\bar{q}}}}{E_{\mathrm{cms}}}, \frac{E_{\mathrm{g}}}{E_{\mathrm{cms}}}\right]\right)^{-2}. \quad (117)$$

For each event variable, the small dataset is first used to construct 20 quantile bins within the relevant intervals: $[0, 1)$ for the three $M_{\mathrm{p_1 p_2}}/E_{\mathrm{cms}}$ variables, $\left[1 - \sqrt{1/3}, 1\right)$ for $1 - \sqrt{1 - \mathrm{Thrust}}$, and $\left[0, \sqrt{16/(3\,\pi^2)}\right)$ for $\sqrt{\mathrm{Spherocity}}$.[37] The weights of the events are ignored in the construction of the quantile bins using the small dataset. The specific transformation of thrust and spherocity were chosen to improve the visibility of the quantile bins in the plots. Next, the medium dataset is used to construct histograms of the event variables, with the quantile bins. Given some selection criteria for including an event in a histogram, the histogram height for a bin $b$ and the corresponding error estimate are given by

$$\hat{F}_b^{(\text{superscript})} \equiv \frac{1}{V_b\,N_{\text{tot}}} \sum_{i=1}^{N_{\text{tot}}} W_i^{(\text{superscript})} \quad \mathbb{I}\begin{pmatrix} \text{event-}i \text{ passes the} \\ \text{selection criteria} \\ \text{and falls in bin-}b \end{pmatrix}, \tag{118}$$

$$\hat{\sigma}_b^{(\text{superscript})} = \frac{1}{V_b\,N_{\text{tot}}} \sqrt{\sum_{i=1}^{N_{\text{tot}}} \left[W_i^{(\text{superscript})}\right]^2 \quad \mathbb{I}\begin{pmatrix} \text{event-}i \text{ passes the} \\ \text{selection criteria} \\ \text{and falls in bin-}b \end{pmatrix}}, \tag{119}$$

respectively, where $N_{\text{tot}}$ is the total number of events in the dataset, $V_b$ is the "volume" or width of the bin-$b$,[38] and $\mathbb{I}(\texttt{condition})$ is a function that evaluates to 1 when "$\texttt{condition}$" is true and 0 otherwise. Note that for the medium dataset, $N_{\text{tot}} = 10^5$, regardless of the number of events that actually pass the selection criteria for each histogram. In each row of plots in Figure 8:

1. The left panel depicts the weighted distributions of the relevant event variable, for class 1 events and class 4 events separately, under the NBL weighting scheme (solid red histograms) and the ARCANE weighting scheme (dotted green histograms).

2. The middle panel depicts the weighted distributions of the relevant event variable, for class 1 and class 4 events combined, under the same two weighting schemes. The histogram heights in the middle panel are just the sum of the relevant histogram heights in the left panel.

3. The right panel depicts two local efficiency factors (for classes 1 and 4, combined) estimated as follows:

$$\begin{matrix} \text{Local efficiency factor} \\ \text{(without unweighting)} \end{matrix} \equiv \frac{\text{SampleMean}\left[\left(W^{(\text{NBL})}\right)^2 \,\middle|\, \text{selection criteria, bin-}b\right]}{\text{SampleMean}\left[\left(W^{(\text{ARCANE})}\right)^2 \,\middle|\, \text{selection criteria, bin-}b\right]}, \tag{120}$$

$$\begin{matrix} \text{Local efficiency factor} \\ \text{(with unweighting)} \end{matrix} \equiv \left(\frac{\text{SampleMean}\left[\left|W^{(\text{NBL})}\right| \,\middle|\, \text{selection criteria, bin-}b\right]}{\text{SampleMean}\left[\left|W^{(\text{ARCANE})}\right| \,\middle|\, \text{selection criteria, bin-}b\right]}\right)^2. \tag{121}$$

The local efficiency factor for bin-$b$ without unweighting (dashed line in the right panels of Figure 8) can roughly be thought of the multiplicative factor by which the number of events sampled in bin-$b$ must be increased in order to achieve, using the NBL weighting scheme, the same uncertainty achieved in bin-$b$ by the ARCANE weighting scheme with the given dataset. The local efficiency factor for bin-$b$ with unweighting (solid line in the right panels of Figure 8) can roughly be interpreted the same way, assuming one incorporates an unweighting step at

---

[37]These intervals are based on the relevant minimum and maximum values of the event variables, for a three-massless-particles final state, in the center-of-momentum frame of the particles.

[38]The bin-widths for the histograms were computed in the transformed parameterization adopted in the respective $x$-axes.

the end of the event-generation-pipeline. Note that such an unweighting was not actually performed in order to estimate this efficiency factor. An unweighting step would change the relative frequency of the different bins—bins with more severe local sign problem will be oversampled post-unweighting [12]. Note that this effect is not taken into account in the definition of the "local efficiency factors with unweighting".

It can be seen that ARCANE reweighting modifies the separate weighted distributions of class 1 and class 4 events, but appears to preserve the combined distributions of class 1 and class 4 events. Based on the right panels of Figure 8, the efficiency of the data is amplified by different degrees in different regions of the phase space of the visible attributes. Since the uncertainties on the estimated local efficiency factors have not been estimated, no quantitative conclusions are drawn from them here. As an aside, the dips near the right- and left- ends of the plots in the histograms of $\sqrt{1 - \sqrt{1 - \text{Thrust}}}$ and $\sqrt{\text{Spherocity}}$, respectively, are caused by the Jacobian factors corresponding to the transformations from thrust and spherocity, respectively.

**Validating the reweighting technique.** The next task is to validate the ARCANE reweighting technique to verify that it does not introduce any biases in the weighted distributions of the events. To this end, the weighted distributions for different event variables predicted by (i) the "ARCANE,BAD" weighting scheme using the medium dataset and (ii) the "NBL" weighting scheme using the large dataset are compared against each other. The "ARCANE,BAD" reweighting scheme is used here to demonstrate that no biases are introduced, regardless of (a) the quality of the approximations in $\kappa_{h_{qq}}$, in particular, and (b) the quality of the redistribution function $G$, in general, as long as they satisfy the necessary well-defined conditions. The large dataset is used to get independent and more accurate reference predictions from the "NBL" weighting scheme. Such comparisons are shown in Figure 9 using one-dimensional histograms, for classes 1 and 4 combined. The same event-variables used in Figure 8 are used in Figure 9 as well, with the same histogram bins.

In each row of Figure 9, the left and right panels show, on a linear-scale and a log-scale $y$-axis, respectively, the weighted histograms under (i) the NBL weighting scheme with the large dataset as a solid red line and (ii) the "ARCANE,BAD" weighting scheme with the medium dataset as a dotted green line. In each plot, the goodness-of-fit measures $\chi^2/n_{\text{bins}}$ and $\frac{\chi^2 - n_{\text{bins}}}{\sqrt{2\,n_{\text{bins}}}}$ are shown, with $\chi^2$ computed as

$$\chi^2 \equiv \sum_{b=1}^{n_{\text{bins}}} \frac{\left[ \hat{F}_b^{(\text{ARCANE,BAD,MEDIUM})} - \hat{F}_b^{(\text{NBL,LARGE})} \right]^2}{\left[ \hat{\sigma}_b^{(\text{ARCANE,BAD,MEDIUM})} \right]^2 + \left[ \hat{\sigma}_b^{(\text{NBL,LARGE})} \right]^2} \,. \tag{122}$$

Under the hypothesis that there are no biases introduced by ARCANE reweighting, $\chi^2$ should *roughly*[39] be expected to follow the $\chi^2$-distribution with $n_{\text{bins}}$ degrees of freedom and $\frac{\chi^2 - n_{\text{bins}}}{\sqrt{2\,n_{\text{bins}}}}$ should *roughly* be expected to follow the standard normal distribution. Note that, since the same two datasets are used in all the panels of Figure 9, the different $\chi^2$-values are not independent of each other.

Taking into account the concrete theory behind the ARCANE reweighting technique, the roughly 3-$\sigma$ discrepancy in the distribution of $M_{\text{qg}}$ is attributed to statistical fluctuations. It is concluded here that the goodness-of-fit tests in Figure 9 support the claim that no biases are introduced by the reweighting. As a reminder, the claim in this paper is that no biases are

---

[39]Various factors like (i) non-normality of $\hat{F}_b$-s, (ii) non-exactness of uncertainty estimates, (iii) $\hat{\sigma}_b^2$-s being positively biased estimates of bin-height variances, (iv) the total number of events being fixed (even though the total weight is not), etc. contribute to $\chi^2$ not exactly following the $\chi^2$-distribution with $n_{\text{bins}}$ degrees of freedom.

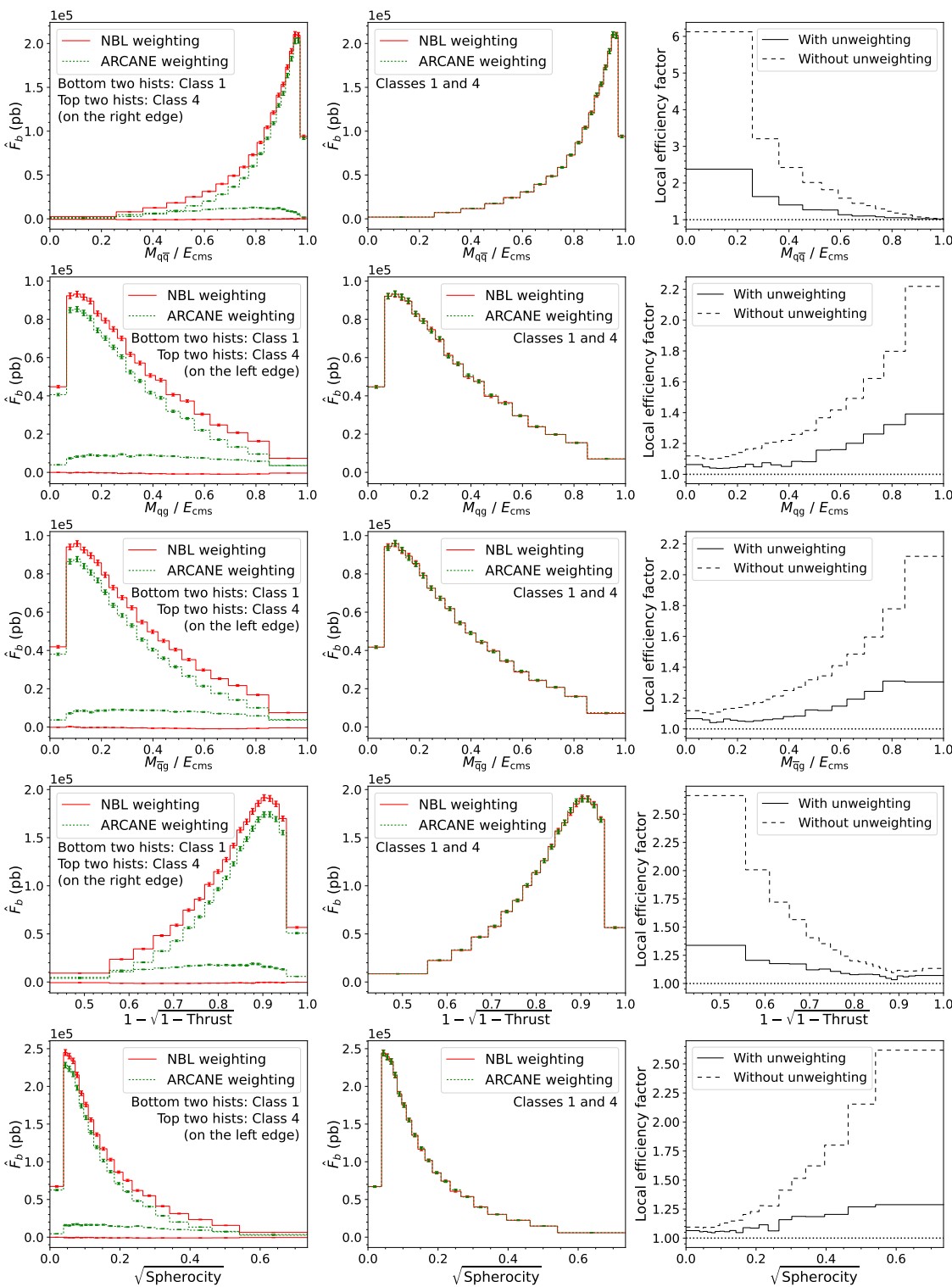

Figure 8: Each row depicts the one-dimensional weighted distributions of a different event variable, under the NBL weighting scheme (solid red) and ARCANE weighting scheme (dotted green). The left panel of each row depicts the separate weighted distributions under class 1 and class 4. The middle panel of each row depicts the combined weighted distributions under classes 1 and 4. The right panel of each row depicts the corresponding local efficiency factors defined in (120) and (121).

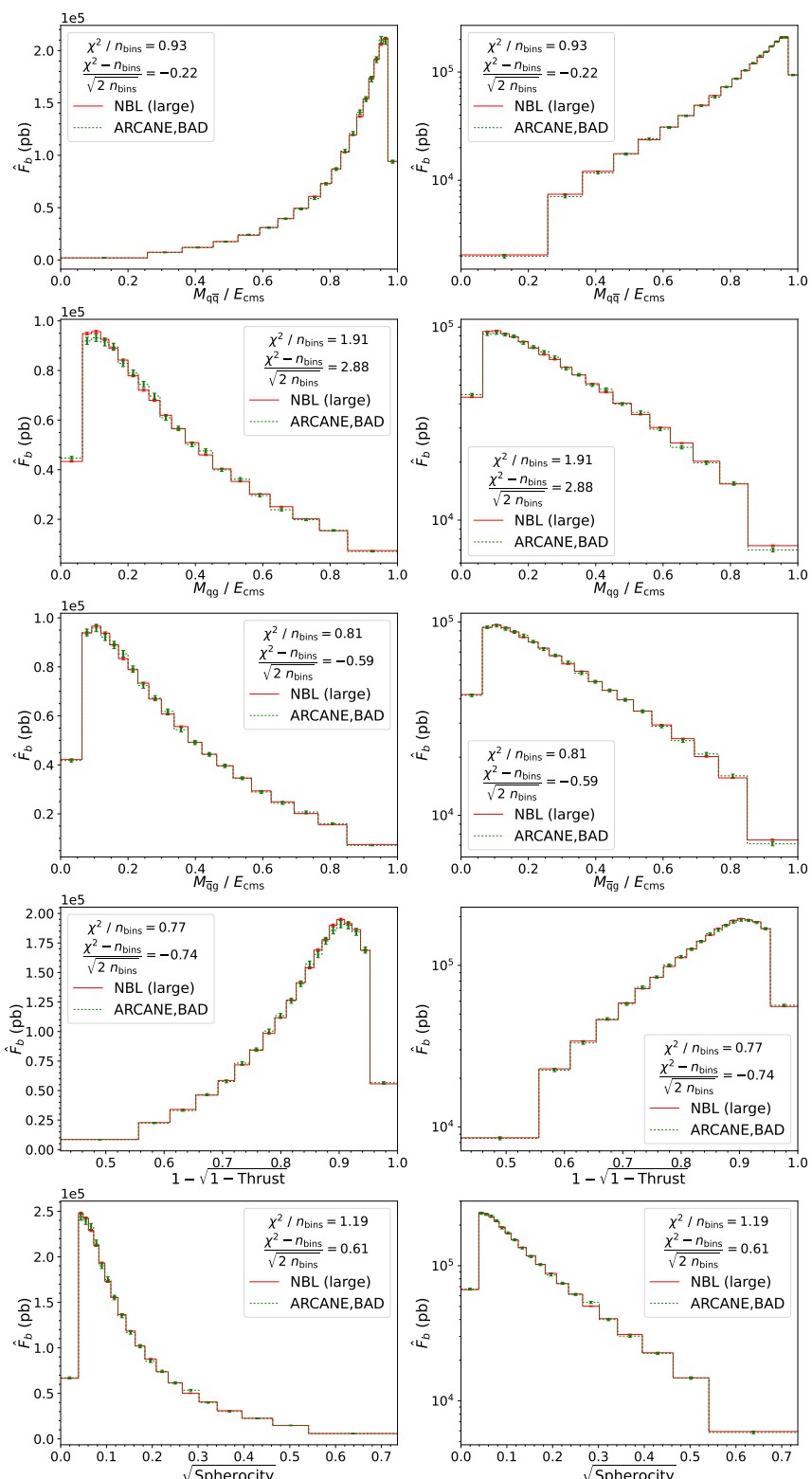

Figure 9: Each row depicts the one-dimensional weighted distributions (for events of classes 1 and 4 combined) of a different event variable, under the NBL weighting scheme using the large dataset (solid red) and under the (ARCANE,BAD) weighting scheme using the medium dataset (dotted green). The left and right panels of each row use a linear-scale and a log-scale $y$-axis, respectively. Goodness-of-fit metrics are shown in each plot, with $\chi^2$ computed using (122).

introduced in the weighted distribution of any combination of visible event-attributes (ignoring errors arising from finite-precision computations). This is claimed to be true for the joint-distributions of any set of "visible" event-variables (IR-safe or otherwise), for any choice of event selection cuts based only on the information in $V$ (including quark flavor), etc. This, of course, cannot be explicitly verified using Monte Carlo experiments.

Note that in quantifying the histogram-uncertainties, objects like the redistribution kernel $\kappa_{h_{qq}}$, the redistribution function $G$, etc. are taken to be fixed (i.e., non-random) in the event generation pipeline. So, the generated events, under the ARCANE weighting scheme, can be taken to be independent and identically sampled. As discussed in Ref. [12], this is not true for positive or neural resampling, and other related techniques in Refs. [6–9, 11, 38, 39]. Uncertainty quantification/estimation and goodness-of-fit testing that validates the uncertainty-estimation-procedure will both have to be more sophisticated, than they are in this paper, when using dependent datasamples produced by such techniques.

**Remaining negatively weighted events.** Even after performing ARCANE reweighting, a total of 22 negatively weighted events still remained in the medium dataset, in classes 1 and 4. The weights of these events were of relatively small magnitudes compared to the typical weight-magnitudes in the dataset. Although the post-unweighting negative weight fraction is practically negligible, the remaining negative weights may be interesting academically.

Broadly, there are two reasons for the ARCANE weight of an event to be negative. The first is suboptimal implementation of ARCANE. If $F^{(\mathrm{ARCANE})}(V) \equiv F^{(\mathrm{ORIG})}(V)$ is strictly non-negative, then performing ARCANE redistribution using the optimal redistribution function $G^*(V, L)$, which satisfies

$$F^{(\mathrm{ARCANE}),*}(V, L) \equiv F^{(\mathrm{ORIG})}(V, L) + G^*(V, L) = F^{(\mathrm{ORIG})}(V)\, P(L \mid V), \qquad (123)$$

is guaranteed to completely eliminate the negative weights. The same is also true for a connected set of redistribution functions $G^\dagger$, which satisfy

$$\mathrm{sgn}\Big(F^{(\mathrm{ARCANE}),\dagger}(V, L)\Big) \equiv \mathrm{sgn}\Big(F^{(\mathrm{ORIG})}(V, L) + G^\dagger(V, L)\Big) \in \Big\{0\,,\ \mathrm{sgn}\Big(F^{(\mathrm{ORIG})}(V)\Big)\Big\}, \qquad (124)$$

where sgn is the sign function. However, if the redistribution function $G$ does not satisfy these conditions, then one can have leftover remaining weights, even if $F^{(\mathrm{ORIG})}(V)$ is non-negative.

The second reason for $W^{(\mathrm{ARCANE})}$ to be negative is $F^{(\mathrm{ORIG})}(V)$ itself being negative for some values of $V$. This could be acceptable for a few reasons, discussed briefly in Ref. [12]. To investigate the source of the remaining negative weights, a scatter plot of $W^{*,\mathrm{lb}}$ vs $W^{*,\mathrm{ub}}$, zoomed in on the negatively weighted events, is shown in the left panel of Figure 10. Recall that for each event, $W^{*,\mathrm{lb}} \leq W^* \leq W^{*,\mathrm{ub}}$ and $W^{*,\mathrm{lb}} \leq W^{(\mathrm{ARCANE})} \leq W^{*,\mathrm{ub}}$, where $W^*$ is the weight of the event under optimal ARCANE reweighting. This plot shows that each of 22 negatively weighted events also have $W^* < 0$, indicating that $F^{(\mathrm{ORIG})}(V) < 0$ for all these events.[40] However, it is likely that there is a small probability for events to have $W^{(\mathrm{ARCANE})} < 0$ but $W^* \geq 0$, even though no such events were present in the medium dataset.

With some trial-and-error, the remaining negatively weighted events were found to occupy a small corner in the phase space of $V$, with low values for the emission parameters $t$ and $z$ (both corresponding to the emission channel which sets the next emission start scale). A scatter plot of $z$ vs $t$ in this corner of the phase space is shown in the right panel of Figure 10. There could potentially be some other regions in the phase space of $(V, L)$ satisfying $W^{(\mathrm{ARCANE})} < 0$, outside this scatter plot; such regions were not represented in the medium dataset. In addition to the events discussed so far, some $\mathbb{H}$-events of class 2 also have negative weights. Recall that,

---

[40]These negative $F^{(\mathrm{ORIG})}(V)$ values cannot be attributed to errors from finite-precision computations.

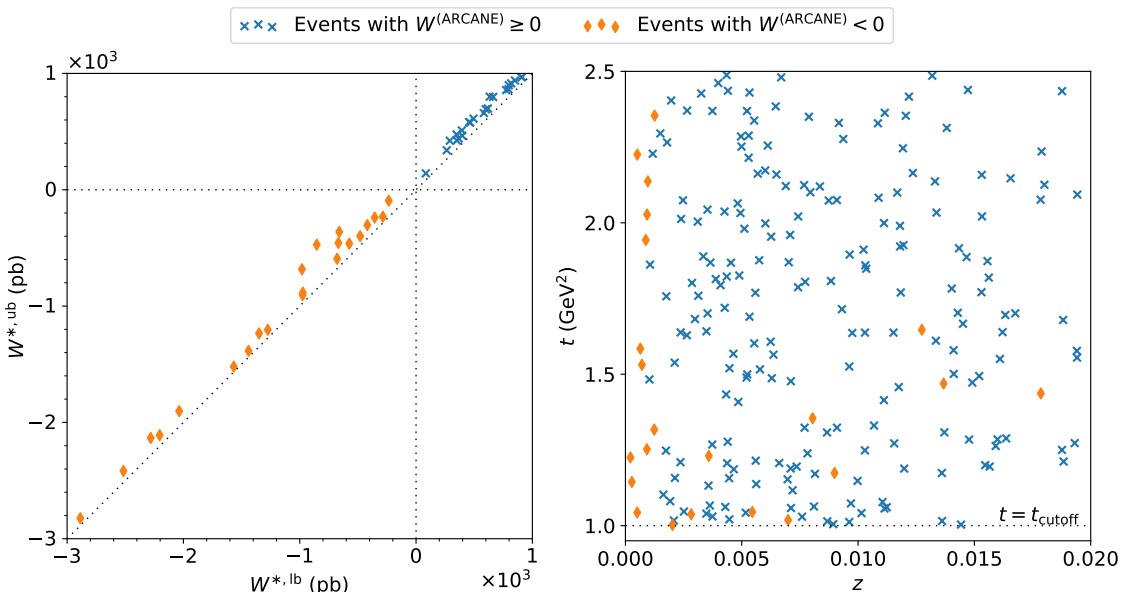

Figure 10: The left panel shows a scatter plot of $W^{*,\mathrm{lb}}$ and $W^{*,\mathrm{ub}}$ for events from classes 1 and 4 with $W^{(\mathrm{ARCANE})} < 0$ in the medium dataset. There are 22 such events in total, marked by orange diamonds. Events from the same classes with $W^{(\mathrm{ARCANE})} \geq 0$, which lie within the zoomed-in region shown in the plot (23 is total), are marked by blue ×-marks. The right panel shows a scatter plot of $z$ and $t \equiv e^{\ell}\,\mathrm{GeV}^2$ for the same 22 events with $W^{(\mathrm{ARCANE})} < 0$ (orange diamond markers). There are a total of 199 events with $W^{(\mathrm{ARCANE})} \geq 0$ (from classes 1 and 4 in the medium dataset) that lie within the zoomed-in region shown in the right panel; they are marked by blue ×-marks. For context, there are a total of 95 154 events in classes 1 and 4 in the medium dataset. So, the zoomed in region filters out a large fraction of events with $W^{(\mathrm{ARCANE})} \geq 0$, while retaining every observed event in these classes with $W^{(\mathrm{ARCANE})} < 0$.

due to the emission scale cutoff, these events have no overlap in distribution with $\mathbb{S}$-events and so their weights were not modified by ARCANE reweighting. Some of the remaining negative weights could be potentially be removed by further redistributing the contributions of different pathways to the same final event, say, after the entire parton showering stage or after the hadronization stage, i.e., by choosing a different stopping criterion for performing the ARCANE reweighting at. This is discussed further in Appendix A.1.

# 5   An Alternative Implementation: "Hard Remainder Spreading"

This section presents an alternative approach to implementing ARCANE reweighting for the process under consideration. As an extreme limit of the original event generator, consider the case where $P(\mathbb{H}) = 0$ and $P(\mathbb{S}) = 1$. In this case, the original generator will not have proper coverage for the contribution of the $\mathbb{H}$-pathways. This is indicated by the weight function

$$W^{(\mathrm{NBL})}(V, \mathbb{H}) \equiv \frac{W_{\mathbb{H}}^{(\mathrm{NBL})}(V)}{P(\mathbb{H})}, \tag{125}$$

becoming undefined when $P(\mathbb{H}) = 0$. However, the expression for $W^{(\mathrm{ARCANE})}(V, \mathbb{H})$ in (80) will be real-valued even if $P(\mathbb{H}) = 0$, as long as $P_{\mathbb{S}}(V) > 0$. This is because, even if $P(\mathbb{H})$ is set to 0, the original contribution of the $\mathbb{H}$-pathways can be carried by the $\mathbb{S}$-pathways under ARCANE

reweighting. This suggests the following alternative implementation of the event generation pipeline: i) Set $P(\mathbb{H})$ to 0 and only sample $\mathbb{S}$-events in the generator. ii) Weight them as per (91), with $P(\mathbb{H}) = 0$ and $P(\mathbb{S}) = 1$. Let this pipeline be referred to as the "$\mathbb{H}$-spreading generator." Similar to Born spreading in Ref. [10], here the contribution of the hard remainder term is being spread over the $\mathbb{S}$-generation pathways.

A problem here is that $\mathbb{H}$-events from class 2, which have non-zero weights, have no overlap with $\mathbb{S}$-events, so their contributions (under the original generator) will be missed by the $\mathbb{H}$-spreading generator. This can be rectified by sampling class 2 events selectively, with a small probability, say $P(\text{cls-2})$, and weighting them appropriately to capture their contribution. With the remaining probability $1 - P(\text{cls-2})$, one would sample events from the $\mathbb{H}$-spreading generator, and correspondingly scale the weights up by a factor of $1/(1 - P(\text{cls-2}))$. For completeness, the probability density and weights of the not-class-2 events under this approach, in terms of the NBL weight functions, will be given by

$$P(V, \text{not cls-2}, \texttt{rej-list}) = \Big(1 - P(\text{cls-2})\Big) P_{\mathbb{S}}(V, \texttt{rej-list}), \qquad (126)$$

$$W(V, \text{not cls-2}, \texttt{rej-list}) = \frac{W_{\mathbb{S}}^{(\text{NBL})}(V_{\text{lo}}, \varphi)}{1 - P(\text{cls-2})}$$
$$+ \left[ \frac{P_{\mathbb{H}}(V) W_{\mathbb{H}}^{(\text{NBL})}(V)}{\Big(1 - P(\text{cls-2})\Big) \widehat{P}_{\mathbb{S}}^{\{h\}}(V)} \right] \left[ \prod_{i=1}^{N_{\text{rej}}} \left( \frac{\kappa_{\{g\}}(\ell_i') - \kappa_{\{h\}}(\ell_i')}{\kappa_{\{g\}}(\ell_i') - \kappa_{\{f\}}(\ell_i')} \right) \right]. \qquad (127)$$

This alternative approach is presented here simply to demonstrate that there could be many different ways of incorporating a deferred additive reweighting into a Monte Carlo pipeline.

# 6   Summary and Conclusions, and Outlook

ARCANE reweighting is a new Monte Carlo technique, introduced in Ref. [12], for tackling the negative weights problem in collider event generation. The present work successfully demonstrates the technique for the generation of $(e^+ e^- \longrightarrow q\bar{q} + 1\,\texttt{jet})$ events under the MC@NLO formalism. After applying ARCANE reweighting, the negative-weights-fraction (in the phase space region with overlapping contributions from both $\mathbb{H}$- and $\mathbb{S}$-pathways) has been reduced to nearly negligible levels, without any changes to the weighted distributions of event-attributes that will be used by the subsequent event production and analysis stages.

Under ARCANE reweighting, the weight of each event is modified with an additive term, which averages out to zero, over the different event generator pathways that lead to the exact same final event. In this way, the additive weight term redistributes the contributions of these generator pathways. A crucial aspect of implementing ARCANE reweighting is the construction of a good "ARCANE redistribution function" for the generation task at hand. The claim in Ref. [12] is that constructing such redistribution functions will be relatively straightforward, especially when compared to constructing alternative (or modified) theory formalisms, which (a) can be sampled straightforwardly using the "Forward Chain Monte Carlo" (FCMC) paradigm as defined in Ref. [12] and (b) have a reduced negative weights problem. This paper demonstrates the construction of such a redistribution function, for the problem at hand, based on just two one-dimensional polynomial fits.

The general difficulty in moving contributions across, say, the $\mathbb{H}$- and $\mathbb{S}$-pathways, within the FCMC paradigm can be understood as follows. Let us say one modifies some aspect of an existing NLO-accurate FCMC event generation pipeline, e.g., by adding a correction term in the $\mathbb{S}$-pathways to (a) the Born differential cross-section function and/or (b) the splitting kernel functions. Even if one has a closed-form expression for the changes made to these func-

tions, the corresponding change in the weighted density of the $\mathbb{S}$-events will likely be hard to compute analytically. So, one likely cannot easily compute the corresponding counterterm to be added to the $\mathbb{H}$-pathways to cancel out the change made on the $\mathbb{S}$-side. More generally, when one modifies an individual step within a FCMC-pipeline, the weighted distribution of the final visible events will be affected in a complicated manner. So, orchestrating the changes to a generation pipeline, within the FCMC-paradigm, such that (a) their effects on the distribution of the visible attributes cancel out sufficient accurately and b) the degree of the negative weights problem reduces to sufficiently low levels can be difficult. In a sense, this is the main reason why the negative weights problem has been difficult to solve. ARCANE sidesteps this difficulty entirely by not conforming to the FCMC paradigm. By computing the reweighting term after the entire event $(V, L)$ has been generated, ARCANE reweighting can isolate the pathways corresponding to a specific $V$, and redistribute their contributions in a controlled manner. For the example process considered in this paper, ARCANE reweighting allows one to, quite literally, take some contribution from the $\mathbb{H}$-pathway leading to a visible event $V$ and redistribute it within the $\mathbb{S}$-pathways leading to the same event $V$.

For the implementation of ARCANE reweighting in this paper, the ARCANE redistribution function $G(V, L)$ depends not only on the original quasi density function $F^{(\mathrm{ORIG})}(V, L)$, but also on the sampling probability function $P(V, L)$. This is necessary for a nearly optimal implementation of ARCANE reweighting. However, if the goal is only to reduce the negative weights problem, then $G(V, L)$ could be defined independent of $P(V, L)$. In other words, $G(V, L)$ could be defined a priori, based only on the theory formalism in $F^{(\mathrm{ORIG})}(V, L)$, independent of the actual event generation pipeline. This approach is arguably conceptually cleaner, and may be preferable in some situations. More generally, the ability to incorporate the contributions of different terms to an event with a specific value of $V$, via a deferred additive (ARCANE) reweighting could facilitate new physics theory formalisms, e.g., for interfacing higher order matrix element calculations with parton showers.

## Thoughts on Tackling Other Situations and Processes of Interest

The implementation of ARCANE reweighting in this paper is just one example of how ARCANE could be used in an event generation pipeline; this implementation alone is not expected to solve the negative weights problem in every situation of interest. When applying the technique to other situations, e.g., hadron collisions or electron-ion collisions, one will likely encounter different challenges—different latent pathways to redistribute contributions across, different functions to approximate, etc.—than the ones tackled in this paper. ARCANE reweighting has been proposed in the companion paper, Ref. [12], as a general framework to tackle these challenges systematically. Some additional challenges and complications one might encounter when tackling different processes and scenarios are as follows:

- In addition to the $f_{qq}$ splitting kernel relevant for the splitting channels tackled in this paper, one will also have to handle the $f_{gq}$ and $f_{gg}$ splitting kernels in hadron collisions. Furthermore, different event generators may be using different choices of splitting kernels than the spin-averaged Catani–Seymour kernels used in this paper. These alternative splitting kernels could possibly (i) involve different shower-evolution-variables, (ii) have azimuthal dependence, etc.

  The implementation in this paper can be extended to such situations as well. Some analytical results which may be useful for approximating the spin-averaged Catani–Seymour $f_{gq}$ and $f_{gg}$ splitting kernels are provided in Appendix A.3.

- Parton showers in hadron collisions involve not just the final state particles, but also the initial state particles. Initial state radiation, final state radiation, and multiparton interactions could be competing in the original event generator [40]. This, combined with the fact

that there are multiple options for initial-state parton-flavors in hadron collisions, increases the complexity of (a) the event generation flowchart and (b) the miscellany of Monte Carlo histories corresponding to the same visible event. The redistribution across the latent generator pathways will also need to be accordingly more sophisticated. Another latent variable that arises under the MC@NLO formalism for hadron collision is the radiative phase space variable in the generation of the unresolved $\mathbb{S}$-events [10]. This particular latent variable is not present in the example tackled in this paper. However, it could be handled with ARCANE reweighting, as well as other previously known techniques like folded integration [41] and Born reweighting [10].

A useful pointer here is that one does not always need to approximately optimally redistribute the contributions across **all** the latent generator pathways of a given visible event. Instead, one can perform the redistribution just within different subsets of pathways, with the goal being to sufficiently reduce the negative weights problem [12].

- In addition to matrix elements (modified by subtraction schemes) and splitting kernels, the quasi density of events will also depend on parton distribution functions (PDFs) in hadron collisions. While this may not add a new latent variable per se, it can still make the implementation of ARCANE reweighting more complicated.

- In the reference event generator [16] used in this paper, the event kinematics (for the unresolved event as well as the first emission) were sampled directly, using simple distributions over analytic parametrizations of the phase space. However, in production quality event generators, event kinematics are often sampled using complicated heuristic parameterizations of the phase space, constructed using importance sampling techniques like VEGAS [42], normalizing flows [42–48], etc., possibly using multichannel techniques [29,30, 46–48]. Tracking the Jacobian factors required to transform to a common parameterization of the visible event $V$ across different latent pathways, in order to perform ARCANE reweighting consistently, will be more complicated in such situations.

- PDFs and splitting kernels of indefinite sign [49] could be another source of negative weights. The implementation of ARCANE reweighting to tackle these situations could be qualitatively different from the implementations to handle the negative weights arising from oversubtractions in the shower matching (which is the problem tackled in this paper).

  The original event generator might involve the weighted veto algorithm in some situations, especially if splitting kernels of indefinite sign are used. The implementation of ARCANE reweighting in this paper was restricted to the unweighted veto algorithm case. However, it can be extended to accommodate the weighted veto algorithm straightforwardly, as discussed in Appendix A.2. Some additional complexities associated with splitting kernels of indefinite sign are also discussed briefly in Appendix A.2.

- ARCANE reweighting could potentially be used with beyond next-to-leading-order event generation as well. For next-to-next-to-leading order generation, one may have to perform ARCANE reweighting after two parton-shower-emissions of the $\mathbb{S}$-events, in order to tackle the corresponding sign problem.

- If the original event generator uses truncate showers [50], then $\mathbb{H}$- and $\mathbb{S}$-events (with one parton shower emission) may not be treated identically, even if their final-state momenta and subsequent starting scales are identical. In such cases, one may have to choose a later stopping point to perform ARCANE reweighting. In general, sufficient care must be taken to make sure that ARCANE reweighting only redistributes the contributions of **latent** generator pathways that lead to practically identical visible events.

The different generalizations of ARCANE reweighting listed in Ref. [12], and the extensions of the implementation in this paper discussed in Appendix A could be useful in tackling some of these situations. A meta-algorithm for implementing ARCANE reweighting for a general event generation task is as follows:

1. Choose a suitable stopping point or criterion within the event generator at which to perform ARCANE reweighting. Identify the visible and latent event-attributes as this point.

2. Identify the set of pathways that can lead to the same visible event. Write down a redistribution function $G(V, L)$ that implements a strategy for exactly redistributing the contributions of these pathways, with enough flexibility to sufficiently reduce the negative weights problem. At this stage $G(V, L)$ could be characterized by yet-unspecified constituent functions, like $\Lambda$ and $\Phi_{\text{un}}$ in this paper.

3. Make the redistribution function concrete, by fixing the constituent functions. This step could be performed by comparing the expression for $G(V, L)$ with the possibly intractable expression for the optimal redistribution function $G^*(V, L)$, and making pragmatic approximations where needed. Machine learning techniques could also be useful here.

The choice of stopping criterion where ARCANE reweighting is performed limits the extent to which the negative weights problem can be reduced by the technique. In practice, one would want to strike a balance between the costs involved in implementing and performing ARCANE reweighting (at different stopping points) and the corresponding value one gets from it. In addition to these steps, the event generator will also have to be modified to track the necessary hidden event-attributes that were not tracked originally. This meta-algorithm can be implemented for classes of collider processes simultaneously, instead of one at a time, similar to how various other aspects of event generation are automated in software [40, 51–56].

## Code and Data Availability

The code and data that support the findings of this study are openly available at the following URL: https://gitlab.com/prasanthcakewalk/arcane-reweighting-demo.

## Acknowledgements

The author thanks Stefan Höche for facilitating this work by suggesting the example collision process considered in this paper and providing a reference event generator implementation [16]. The author thanks Stefan Höche, Stephen Mrenna, Kevin Pedro, Nicholas Smith, and Manuel Szewc for useful discussions and/or feedback on this manuscript. The following open source software were used *directly* in performing this research and generating images and plots: Python [57], NumPy [58], SageMath [31], Maxima [32], SymPy [33], Matplotlib [59], Jupyterlab [60], tqdm [61], Schemdraw [62].

**Funding information** This document was prepared using the resources of the Fermi National Accelerator Laboratory (Fermilab), a U.S. Department of Energy, Office of Science, Office of High Energy Physics HEP User Facility. Fermilab is managed by Fermi Forward Discovery Group, LLC, acting under Contract No. 89243024CSC000002. The author is supported by the U.S. Department of Energy, Office of Science, Office of High Energy Physics QuantISED program under the grants "HEP Machine Learning and Optimization Go Quantum", Award

Number 0000240323, and "DOE QuantiSED Consortium QCCFP-QMLQCF", Award Number DE-SC0019219.

# A  Some Extensions of the ARCANE Implementation in This Paper

## A.1  More Pathways to Redistribute Across

As discussed in Section 4, even an optimal implementation of ARCANE reweighting, at the stopping point chosen in this paper, would leave some residual negative weights. Some of these could potentially be eliminated, if needed, by performing an additional redistribution, as discussed below.

For events of classes 1 or 4, the same $q\bar{q}g$ final state momenta can be reached via either of the two emission channels, indicated by $c_{\text{for-scale}} = 1$ and $c_{\text{for-scale}} = 2$, respectively. However, the two channels lead to different starting scales, say $\ell_{\text{start},1}$ and $\ell_{\text{start},2}$, respectively, for the subsequent parton shower emission. So, the two channels do not lead to the same visible event, and hence their contributions were not redistributed in this paper. Let $\ell_{\text{start},i} \geq \ell_{\text{start},3-i}$ for some $i \in \{1,2\}$. If one continues the evolution of the event from channel $i$ through the generator, there is a non-zero probability of no emissions occurring until $\ell = \ell_{\text{start},3-i}$. At this point, the event from channel $i$ will practically be the same as the event from channel $3-i$, so their contributions can be redistributed. The stopping criterion needs to be carefully defined to perform this redistribution properly. One option is to delay the stopping criterion only for the channel leading to the larger of the $\ell_{\text{start}}$-s, until either another successful emission occurs or the smaller of the $\ell_{\text{start}}$-s is reached. Implementation-wise, such a redistribution can be performed simultaneously with or subsequent to the redistribution performed in this paper.

One can also undertake a more complicated task, by showering all the events up to some fixed scale, leading to different numbers of successful emissions for different events, and performing ARCANE reweighting at this point. This fixed scale could be greater than the actual cutoff scale $\ell_{\text{cutoff}}$ of the generator.

## A.2  Weighted Veto Algorithm and Splitting Kernels of Indefinite Sign

Let the only change to the original event generator be that the weighted veto algorithm, as discussed in Section 2.6, is used instead of the unweighted veto algorithm. The optimal redistribution function in this case is given by

$$G^*(V, \mathbb{H}) \equiv -\Lambda_1^*(V) - \Lambda_2^*(V), \tag{A.1}$$

$$G^*(V, \mathbb{S}, \texttt{rej-list}) \equiv \Lambda_1^*(V)\, \Phi_{\{f\}}^*(\texttt{rej-list}\,;\, V) + \Lambda_2^*(V)\, \Phi_{\{\alpha g\}}^*(\texttt{rej-list}\,;\, V), \tag{A.2}$$

where $\Lambda_1^*$ and $\Lambda_2^*$ are given by

$$\Lambda_1^*(V) \equiv -F^{(\text{NBL})}(V, \mathbb{S}), \tag{A.3}$$

$$\Lambda_2^*(V) \equiv \frac{F^{(\text{NBL})}(V, \mathbb{H}) + F^{(\text{NBL})}(V, \mathbb{S})}{P(V, \mathbb{H}) + P(V, \mathbb{S})}\, P(V, \mathbb{S}), \tag{A.4}$$

and the unit-normalized (not necessarily non-negative) quasi densities $\Phi_{\{f\}}^*(\texttt{rej-list}\,;\, V)$ and $\Phi_{\{\alpha g\}}^*(\texttt{rej-list}\,;\, V)$ are given by

$$\Phi_{\{f\}}^*(\texttt{rej-list}\,;\, V)$$
$$= \frac{\Theta(\ell_{N_{\text{rej}}}' - \ell)\, \Delta_{\{g\}}(\ell, \ell_{N_{\text{rej}}}')}{\Theta(\ell_{\text{start}} - \ell)\, \Delta_{\{f\}}(\ell, \ell_{\text{start}})}\ \prod_{i=1}^{N_{\text{rej}}} \left[ \Theta(\ell_{i-1}' - \ell_i')\, \Delta_{\{g\}}(\ell_i', \ell_{i-1}') \left( g_{c_i'}(\ell_i', z_i') - f_{c_i'}(\ell_i', z_i') \right) \right], \tag{A.5}$$

$$\Phi^*_{\{\alpha g\}}(\texttt{rej-list}\,;\,V)$$

$$= \frac{\Theta(\ell'_{N_{\text{rej}}} - \ell)\,\Delta_{\{g\}}(\ell, \ell'_{N_{\text{rej}}})}{\Theta(\ell_{\text{start}} - \ell)\,\Delta_{\{\alpha g\}}(\ell, \ell_{\text{start}})}\,\prod_{i=1}^{N_{\text{rej}}}\left[\Theta(\ell'_{i-1} - \ell'_i)\,\Delta_{\{g\}}(\ell'_i, \ell'_{i-1})\left(g_{c'_i}(\ell'_i, z'_i) - \alpha_{c'_i}(\ell'_i, z'_i)\,g_{c'_i}(\ell'_i, z'_i)\right)\right],$$

$$\text{(A.6)}$$

In (A.2), the first additive term on the right-hand-side subtracts away the original weight of the $\mathbb{S}$-events and the second additive term adds the optimal weight $W^*(V)$. $\Delta_{\{f\}}$ and/or $\Delta_{\{\alpha g\}}$ may have to be appropriately approximated in order to construct a redistribution function $G$ which approximately matches $G^*$. For the unweighted veto algorithm $\{\alpha g\} = \{f\}$; plugging this in the previous equations leads to the optimal redistribution function in Section 3.4.

Weighted veto algorithms become necessary if the splitting kernels used in the parton showers are of indefinite sign. Such splitting kernels introduce another difficulty. Even if one (almost completely) eliminates the negative weights at some point within the parton shower pipeline, subsequent emissions can lead to more negative weights. If this is a significant source of negative weights for some generation task, then one will have to choose a later stopping point within the generation pipeline, at which to perform ARCANE reweighting, in order to sufficiently reduce the negative weights problem.

## A.3 Some Additional Analytic Integrals of Catani–Seymour Splitting Kernels

In the notations of this paper, the spin-averaged $z$-dependent Catani–Seymour splitting kernels $f_{gq}$ and $f_{gg}$, for the $g \longrightarrow q\bar{q}$ and $g \longrightarrow gg$ splittings, respectively are given by [19]:

$$f_{gq}(\ell, z\,;\,M^2) \equiv \frac{\alpha_s(t)}{2\,\pi}\,\frac{T_R}{2}\,(1-y)\left(1 - 2z(1-z)\right)\Theta(z - z_-)\,\Theta(z_+ - z)\,\Theta\!\left(\frac{M^2}{4} - t\right), \quad \text{(A.7)}$$

$$f_{gg}(\ell, z\,;\,M^2) \equiv \frac{\alpha_s(t)}{2\,\pi}\,\frac{C_A}{2}\,(1-y)\left(\frac{2}{1 - z(1-y)} - 2 + z(1-z)\right)\Theta(z - z_-)\,\Theta(z_+ - z)$$

$$\times\,\Theta\!\left(\frac{M^2}{4} - t\right), \quad \text{(A.8)}$$

where $T_R$ and $C_A$ are appropriate color-factors. These functions were integrated with respect to $z$, over the relevant domain, using SageMath to get

$$\kappa_{f_{gq}}(\ell\,;\,M^2) \equiv \int_{\mathbb{R}} \mathrm{d}z\, f_{gq}(\ell, z\,;\,M^2)$$

$$= \frac{\alpha_s(t)}{2\,\pi}\,\frac{T_R}{2}\,\Theta\!\left(\frac{M^2}{4} - t\right) \times \left(-z^2 + \frac{2z^3}{3} + \rho\,(1 + 2\rho) + \rho\,\ln\!\left(\frac{1-z}{z}\right)\right)\Bigg|_{z=z_-}^{z=z_+}, \quad \text{(A.9)}$$

$$\kappa_{f_{gg}}(\ell\,;\,M^2) \equiv \int_{\mathbb{R}} \mathrm{d}z\, f_{gg}(\ell, z\,;\,M^2)$$

$$= \frac{\alpha_s(t)}{2\,\pi}\,\frac{C_A}{2}\,\Theta\!\left(\frac{M^2}{4} - t\right) \times \left(\frac{z^2}{2} - \frac{z^3}{3} - z\,(2 + \rho) - 2\rho\,\ln(1-z) + \frac{2\rho^2\,\ln(z)}{1+\rho}\right.$$

$$\left.- \frac{\ln\!\left(\rho + (1-z)^2\right)}{1+\rho} + \frac{2\,\sqrt{\rho}\,\arctan\!\left(\frac{1-z}{\sqrt{\rho}}\right)}{1+\rho}\right)\Bigg|_{z=z_-}^{z=z_+}, \quad \text{(A.10)}$$

where $\rho \equiv t/M^2$. The following limiting behavior of $\kappa_{f_{gq}}$ and $\kappa_{f_{qq}}$, guessed numerically and subsequently verified using SymPy, may be useful for constructing approximations for them:

$$\lim_{\ell \to \ln(M^2/(4\,\text{GeV}^2))^-} \kappa_{f_{gq}}(\ell \; ; \; M^2) \left(\ln\left(\frac{M^2}{4\,\text{GeV}^2}\right) - \ell\right)^{-3/2} = \frac{\alpha_s(t)}{2\,\pi} \frac{T_R}{6}, \qquad \text{(A.11)}$$

$$\lim_{\ell \to \ln(M^2/(4\,\text{GeV}^2))^-} \kappa_{f_{gg}}(\ell \; ; \; M^2) \left(\ln\left(\frac{M^2}{4\,\text{GeV}^2}\right) - \ell\right)^{-3/2} = \frac{\alpha_s(t)}{2\,\pi} \frac{C_A}{12}. \qquad \text{(A.12)}$$

The equivalent result for $\kappa_{f_{qq}}$, provided in a different form in (105), is

$$\lim_{\ell \to \ln(M^2/(4\,\text{GeV}^2))^-} \kappa_{f_{qq}}(\ell \; ; \; M^2) \left(\ln\left(\frac{M^2}{4\,\text{GeV}^2}\right) - \ell\right)^{-3/2} = \frac{\alpha_s(t)}{2\,\pi} \frac{C_F}{3}. \qquad \text{(A.13)}$$

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
