# Peer review of "A Demonstration of ARCANE Reweighting: Reducing the Sign Problem in the MC@NLO Generation of e+eqˉq+1jet Events"

_SciPost Physics_

## Round 1 · Referee Report · Anonymous (Referee 1) · 2025-5-2

Report

This manuscript presents an example implementation of a novel method
for reducing the number of negative weights in event generation, as
introduced in a companion paper. The method itself, as discussed in
that companion work, is sound and potentially valuable for the
field. The present paper attempts to showcase its application in a
specific context.

However, I have serious reservations about the implementation
presented in this work. The most critical issue lies in the choice of
the shower starting scale lstart for the H-events. In
this work, it is assumed that the value of this scale is typically the
same as for S-events that have similar kinematics after one shower
emission. Based on this assumption, the author applies the ARCANE
reweighting after a single shower emission in S-events and no
emissions in H-events.

In a realistic setup, however, the shower scale for H-events generally
does not correspond to the shower scale for the S-events in the
relevant regions of phase space. This was already recognized in the
original MC@NLO publication, where the discussion around the `EMSCA'
scale (page 36 of Ref.~[13]) supports choosing EMSCA=s2pT, a value quite different from the shower scale of the
corresponding one-emission S-event. A more recent discussion of this
issue can also be found in \texttt{arXiv:2409.16417}, particularly
around equation~(4.6).

The underlying reason is that to preserve the cancellation of
soft/collinear singularities between the virtual and real-emission
matrix elements, one must ensure that the parton shower acts
consistently across these contributions. This requires the shower
starting scale for H-events to be large---close to
s---especially when the additional parton in the
real-emission matrix element is soft or collinear. Choosing a starting
scale based on the transverse momentum of the emitted parton, as in
this paper, undermines this consistency.

Since the ARCANE reweighting implementation critically depends on a
simple identification between S-events (after one shower emission) and
H-events (with none), using an inconsistent shower starting scale for
H-events compromises the validity of the reweighting. In principle,
ARCANE could be applied after the full shower is completed, but that
would be significantly more involved.

Given this, the author's setup might still be acceptable in a
controlled toy model. However, in general, it is not appropriate for
any realistic physics scenario. Unfortunately, the paper does not
clearly state its intended scope---whether it aims to serve as a toy
model validation or a realistic application.

If the intention was to demonstrate the method in a simplified toy
setting, the implementation is unnecessarily elaborate. In that case,
this study would have been better presented as an extended example
within the main method paper rather than a stand-alone publication.

Conversely, if the goal was to showcase a realistic application of the
method, the current implementation falls short. A realistic study
would require a better-justified choice of parameters---most
importantly, a physically motivated shower starting scale.

Due to this fundamental ambiguity in the paper’s goals, and the
inadequate justification of a critical technical choice, I do not
believe the manuscript in its current form meets the standards for
publication. A major revision would be necessary, including a clear
statement of the paper’s aim (toy model vs.\ realistic setup), a
reworking of the event generation to match that aim, and a more
defensible treatment of the shower starting scale.

Recommendation

Ask for major revision

---

## Round 1 · Referee Report · Anonymous (Referee 2) · 2025-5-11

Strengths

  1. The author presents the application of a novel method to significantly reduce the number of negative weights in Monte Carlo events. Negative events increase the computational resources needed to generate and process Monte Carlo events by a significant factor. An almost complete removal of negative events from a sample, like presented in the paper, albeit for a simple case, is therefore highly desirable.
  2. The author identifies and discusses challenges that need to be overcome for adapting the method to hadronic processes, and the complexities of state-of-the-art calculations.

Weaknesses

  1. The stopping point identified (i.e. after the H event, or after the S event with an extra emission), would not work for more complex calculations, that e.g. include starting scale dependent on whether the event is of H- or S-type, or for truncated showers. In these cases, one might need to defer the starting point until after the shower has evolved further down to a scale that is reachable by both events, in order to be able to call them identical in terms of visible parameters, such that the condition for weight redistribution is indeed given. However, this is not a serious weakness, because the formalism allows for it in principle, and the author identifies and discusses this challenge himself. Also, the author repeatedly acknowledges that this is not an application to a production-quality generator (e.g. page 21 + 30), thus not claiming that this is not yet at the level of being a solution for a state-of-the-art simulation.

Report

The journal's acceptance criteria are met. Given the identified weakness, I would argue that "Present a breakthrough on a previously-identified and long-standing research stumbling block" is not yet fulfilled at this point. However, I agree with the author that the expectation "Open a new pathway in an existing or a new research direction, with clear potential for multi-pronged follow-up work" is indeed fulfilled. The follow-up work as I see it would be to adapt the method to hadronic collisions and multi-jet merged calculations at NLO and show that the general formalism can indeed be practically used.

Also all general acceptance criteria are fulfilled. In particular, the author has made sure to describe his work in a pedagogical and detailed way that allows for a straightforward reproduction of his results.

Requested changes

  1. Please make sure to prominently discuss the current limitation that the implementation as given assumes the equality of the starting scales of H and S events. While this is not a limitation of the formalism itself, and it is mentioned in the paper in several places, it would be useful to guide the user's expectation with respect to the "readiness" of the approach early in the paper. I would suggest to discuss this in the introduction. The last point on p. 45 should also be adapted to acknowledge that the subsequent starting scales might not be identical.
  2. Is there a word missing for the title of App. A.1? It would seem that the title should end with "across different event classes", or something like that.

Recommendation

Ask for minor revision

---

## Round 1 · Referee Report · Anonymous (Referee 3) · 2025-5-20

Strengths

1) Demonstration of a novel method to reduce the number of negative weights in MC@NLO matching samples 2) Has a clear path towards full event generator setup

Weaknesses

1) Does not provide sufficient detail as a stand-alone paper 2) Unclear scope of the paper: toy model vs. realistic simulation

Report

The article presents an example implementation of the "ARCANE reweighting" strategy introduced by the same author in an accompanying paper. The focus on the current article is on demonstrating the reduction of negative weights in the MC@NLO matching of e+e- -> 3jet events. The method is interesting and promising to solve this problem at least partially.

While there are novel aspects in this work, I can not recommend to publish this article in SciPost Physics for the following reasons:

1) A large part of the paper (section 2) is simply summarizing known facts about parton shower monte carlo programs in an unconventional notation.

2) The author pays extreme attention towards defining and deriving all necessary probability densities and weights. However, these expositions are hard to understand without the accompanying paper. Therefore, this article does not provide sufficient details for this paper to stand by itself.

3) Somehow the scope of the paper changed from the introduction to the conclusions: p.3 "From the simple description above of ARCANE reweighting, it may not be clear how or if the technique can be used to tackle the negative weights problem in realistic event generation scenarios; this paper is intended to bridge this gap."

p.45 "However, in production quality event generators, ..."

So it is not clear if this paper is supposed to be a simplified demonstration of the technique or a full fledged analysis of a new event generator. As the generator used in this article is itself just a toy model implementation I can only assume that the current article is at best a toy model demonstration of Arcane reweighting.

This toy model aspect is also supported by some fundamental misconceptions. For instance, it has been repeatedly mentioned that the event-type is chosen randomly from the set {S,H} with probabilities P(H) = 1/4 and P(S) = 3/4. However, these probabilities should be proportional to the NLO cross section contributions.

Overall, this article reads like a -- very elaborate -- toy model implementation that should better be part of the original publication. In its current form I can not recommend it for publication in SciPost Physics.

Nonetheless, I want to encourage the author to continue his work on this topic, as it indeed has many possible follow up directions and could contribute to lowering the necessary computing budget for the upcoming HL-LHC phase.

Recommendation

Reject

---

## Editorial Decision

awaiting_resubmission